# Constrained Langevin Algorithms with L-mixing External Random Variables

**Yuping Zheng**
Department of Electrical and Computer Engineering
University of Minnesota, Twin Cities
Minneapolis, MN 55455
zhen0348@umn.edu

**Andrew Lamperski**
Department of Electrical and Computer Engineering
University of Minnesota, Twin Cities
Minneapolis, MN 55455
alampers@umn.edu

## Abstract

Langevin algorithms are gradient descent methods augmented with additive noise, and are widely used in Markov Chain Monte Carlo (MCMC) sampling, optimization, and machine learning. In recent years, the non-asymptotic analysis of Langevin algorithms for non-convex learning has been extensively explored. For constrained problems with non-convex losses over a compact convex domain with IID data variables, the projected Langevin algorithm achieves a deviation of $O(T^{-1/4}(\log T)^{1/2})$ from its target distribution [27] in 1-Wasserstein distance. In this paper, we obtain a deviation of $O(T^{-1/2} \log T)$ in 1-Wasserstein distance for non-convex losses with $L$-mixing data variables and polyhedral constraints (which are not necessarily bounded). This improves on the previous bound for constrained problems and matches the best-known bound for unconstrained problems.

## 1 Introduction

Langevin algorithms can be viewed as the simulation of Langevin dynamics from statistical physics [14]. They have been widely studied for Markov Chain Monte Carlo (MCMC) sampling [37], non-convex optimization [5, 23] and machine learning [43]. In the statistical community, Langevin methods are used to resolve the difficulty of exact sampling from a high dimensional distribution. For non-convex optimization, the additive noise assists the algorithms to escape from local minima and saddles. Since many modern technical challenges can be cast as sampling and optimization problems, Langevin algorithms are a potential choice for the areas of adaptive control, deep neural networks, reinforcement learning, time series analysis, image processing and so on [4, 10, 29].

**Related Work.** In recent years, the non-asymptotic analysis of Langevin algorithms has been extensively studied. The discussion below reviews theoretical studies of Langevin algorithms for MCMC sampling, optimization, and learning.

The non-asymptotic analysis of Langevin algorithms for approximate sampling (Langevin Monte Carlo, or LMC) began with [16, 17], with more recent relevant work given in [3, 4, 10, 13, 19, 22, 28, 30, 32–34, 42, 46]. Most works on LMC consider log-concave target distributions, though there exists some work relaxing log-concavity [10, 13, 33, 34, 42] and smoothness of the target distribution [13, 28, 34]. Most LMC work focuses on the unconstrained case.

36th Conference on Neural Information Processing Systems (NeurIPS 2022).

Constrained problems are less studied, but a variety of works have begun to address constraints in recent years. The work [8, 9] analyzes the case of log-concave distributions with samples constrained to a convex, compact set. Other methods derived from optimization have been introduced to handle constraints, such as mirror descent [1, 25, 26, 45] and proximal methods [7].

Pioneering work on non-asymptotic analysis of Langevin algorithms for unconstrained non-convex optimization with IID external data variables was given in [35], which was motivated by machine learning applications [43]. Since then, numerous improvements and variations on unconstrained Langevin algorithms for non-convex optimization have been reported [10–12, 21, 44].

The work [41] examines the Unadjusted Langevin Algorithms without convexity assumption of the objective function and achieves a convergence guarantee in Kullback-Leibler (KL) divergence assuming that the target distribution satisfies a log-Sobolev inequlity. However, KL divergence is infinite with the deterministic initialization. To mitigate this pitfall, our work measures the convergence bound in 1-Wasserstein distance, which allows the initial condition to be deterministic.

The first analysis of Langevin algorithms for non-convex optimization with IID external variables constrained to compact convex sets is given in [27], and builds upon [8, 9]. However, the convergence rate derived in [27] is rather slow since it uses a loose result on Skorokhod problems in [40]. Recent work of [39] obtains $\epsilon$-suboptimality guarantees in $\tilde{O}(\epsilon^{-1/3})$. However, some extra work would be required to give a direct comparison with the current work, as the results in [39] depend additionally on the spectral gap, which is not computed here.

Most convergence analyses for constrained non-convex optimization require no constraints or bounded constraint sets and IID external random variables or no external variables. In practice, the boundedness of constraint sets and the dependence of external variables do not always hold. The work [10] gives non-asymptotic bounds with L-mixing external variables and non-convex losses, which achieves tight performance guarantees in the unconstrained case. In contrast, our work gets a tight convergence bound (up to logarithmic factors) with L-mixing data streams and applies to arbitrary polyhedral constraints, which may be unbounded.

**Contributions.** This paper focuses on the non-asymptotic analysis of constrained Langevin algorithms for a non-convex problem with L-mixing external random variables and polyhedral constraints. We show the algorithm can achieve a deviation of $O(T^{-1/2} \log T)$ from its target distribution in 1-Wasserstein distance in the polyhedral constraint and with dependent variables. The result from [10] on unconstrained Langevin algorithms with L-mixing external random variables gives a deviation of $O(T^{-1/2}(\log T)^{1/2})$, and so we see that our results match, up to a factor of $(\log T)^{1/2}$. For constrained problems, our general polyhedral assumption is not directly comparable to related work of [27], which examines compact convex constraints, and [39], which examines bounded non-convex constraints. In the cases where the domains and random variable assumptions match (i.e. bounded polyhedra with IID external random variables or no external random variables), our paper gives the tightest bounds. In particular, this improves on the bound from [27], which gives a deviation of $O(T^{-1/4}(\log T)^{1/2})$ with respect to 1-Wasserstein distance.

A key enabling result in this paper is a new quantitative bound on the deviation between Skorokhod problem solutions over polyhedra, which gives a more explicit variation of an earlier non-constructive result from [18]. Additionally, we derive a relatively simple approach to averaging out the effect of L-mixing random variables on algorithms.

## 2 Problem Setup

### 2.1 Notation and terminology

$\mathbb{R}$ denotes the set of real numbers while $\mathbb{N}$ denotes the set of non-negative integers. The Euclidean norm over $\mathbb{R}^n$ is denoted by $\| \cdot \|$.

Random variables will be denoted in bold. If $\mathbf{x}$ is a random variable, then $\mathbb{E}[\mathbf{x}]$ denotes its expected value and $\mathcal{L}(\mathbf{x})$ denotes its law. IID stands for independent, identically distributed. The indicator function is denoted by $\mathbb{1}$. If $P$ and $Q$ are two probability measures over $\mathbb{R}^n$, then the 1-Wasserstein distance between them with respect to the Euclidean norm is denoted by $W_1(P, Q)$.

The 1-Wasserstein distance is defined as:

$$W_1(P, Q) = \inf_{\Gamma \in \mathfrak{C}(P,Q)} \int_{\mathcal{K} \times \mathcal{K}} \|x - y\| d\Gamma(x, y)$$

where $\mathfrak{C}$ is the couplings between $P$ and $Q$.

Let $\mathcal{K}$ be a convex set. (In this paper, we will assume that $\mathcal{K}$ is polyhedral with 0 in its interior.) The boundary of $\mathcal{K}$ is denoted by $\partial \mathcal{K}$. The normal cone of $\mathcal{K}$ at a point $x$ is denoted by $N_{\mathcal{K}}(x)$. The convex projection onto $\mathcal{K}$ is denoted by $\Pi_{\mathcal{K}}$.

Let $\mathcal{Z}$ denote the domain of the external random variables $\mathbf{z}_k$.

If $\mathcal{F}$ and $\mathcal{G}$ are $\sigma$-algebras, let $\mathcal{F} \vee \mathcal{G}$ denote the $\sigma$-algebra generated by the union of $\mathcal{F}$ and $\mathcal{G}$.

## 2.2 Constrained Langevin algorithm

For integers $k$ let $\hat{\mathbf{w}}_k \sim \mathcal{N}(0, I)$ be IID Gaussian random variables and let $\mathbf{z}_k$ be an L-mixing process whose properties will be described later. Assume that $\mathbf{z}_i$ is independent of $\hat{\mathbf{w}}_j$ for all $i, j \in \mathbb{N}$.

Assume that the initial value of $\mathbf{x}_0 \in \mathcal{K}$ is independent of $\mathbf{z}_i$ and $\hat{\mathbf{w}}_j$. Then the constrained Langevin algorithm has the form:

$$\mathbf{x}_{k+1} = \Pi_{\mathcal{K}} \left( \mathbf{x}_k - \eta \nabla_x f(\mathbf{x}_k, \mathbf{z}_k) + \sqrt{\frac{2\eta}{\beta}} \hat{\mathbf{w}}_k \right), \tag{1}$$

with $k$ an integer. Here $\eta > 0$ is the step size parameter and $\beta > 0$ is the inverse temperature parameter. In the learning context, $f(\mathbf{x}, \mathbf{z})$ is the objective function where $\mathbf{x}$ are the parameters we aim to learn and $\mathbf{z}$ is a training data point.

## 2.3 L-mixing processes

In this paper, we assume that $\mathbf{z}_k$ is a sequence of external data variables. The class of $L$-mixing processes was introduced in [24] for applications in system identification and time-series analysis, and gives a means to quantitatively measure how the dependencies between the $\mathbf{z}_k$ decay over time. Formally, $L$-mixing requires two components: 1) M-boundedness, which specifies a global bound on the moments and 2) a measure of the decay of influence over time.

A discrete-time stochastic processes $\mathbf{z}_k$ is M-bounded if for all $m \geq 1$

$$\mathcal{M}_m(\mathbf{z}) = \sup_{k \geq 0} \mathbb{E}^{1/m} \left[ \|\mathbf{z}_k\|^m \right] < \infty. \tag{2}$$

Let $\mathcal{F}_k$ be an increasing family of $\sigma$-algebras such that $\mathbf{z}_k$ is $\mathcal{F}_k$-measurable and $\mathcal{F}_k^+$ be a decreasing family of $\sigma$-algebras such that $\mathcal{F}_k$ and $\mathcal{F}_k^+$ are independent for all $k \geq 0$. Then, the process $\mathbf{z}_k$ is L-mixing with respect to $\left( (\mathcal{F}_k), (\mathcal{F}_k^+) \right)$ if it is M-bounded and

$$\Psi_m(\mathbf{z}) = \sum_{\tau=0}^{\infty} \psi_m(\tau, \mathbf{z}) < \infty \tag{3a}$$

with

$$\psi_m(\tau, \mathbf{z}) = \sup_{k \geq \tau} \mathbb{E}^{1/m} \left[ \left\| \mathbf{z}_k - \mathbb{E} \left[ \mathbf{z}_k | \mathcal{F}_{k-\tau}^+ \right] \right\|^m \right]. \tag{3b}$$

For a concrete example, consider the order-1 autoregressive model:

$$\mathbf{z}_{k+1} = \alpha \mathbf{z}_k + \boldsymbol{\xi}_{k+1} \tag{4}$$

where $\alpha$ is a constant with $|\alpha| < 1$ and for all $k \in \mathbb{Z}$, $\boldsymbol{\xi}_k$ are IID standard Gaussian random variables and $\mathbf{z}_k \in \mathcal{Z}$, where $\mathcal{Z} = \mathbb{R}$ in this case. It can be observed from (4) that

$$\mathbf{z}_k = \sum_{j=0}^{\infty} \alpha^j \boldsymbol{\xi}_{k-j}. \tag{5}$$

Then, if we specify $\mathcal{F}_k = \sigma\{\boldsymbol{\xi}_i : i \leq k\}$ and $\mathcal{F}_k^+ = \sigma\{\boldsymbol{\xi}_i : i > k\}$, it can be verified that $\mathbf{z}_k$ satisfies (2) and (3) and so is an L-mixing process.

## 2.4 Assumptions

We assume that $\nabla_x f(x, z)$ is $\ell$-Lipschitz in both $x$ and $z$. In particular, this implies that $\|\nabla_x f(x_1, z) - \nabla_x f(x_2, z)\| \leq \ell\|x_1 - x_2\|$ and $\|\nabla_x f(x, z_1) - \nabla f(x, z_2)\| \leq \ell\|z_1 - z_2\|$.

We assume that $\mathbf{z}_t$ is a stationary $L$-mixing process, and let $\bar{f}(x) = \mathbb{E}[f(x, \mathbf{z}_t)]$ denote the function which averages $f(x, \mathbf{z}_t)$ with respect to $\mathbf{z}_t$.

Further, we assume that $\bar{f}(x)$ is $\mu$-strongly convex outside a ball of radius $R > 0$, i.e. $(x_1 - x_2)^\top \left( \nabla \bar{f}(x_1) - \nabla \bar{f}(x_2) \right) \geq \mu \|x_1 - x_2\|^2$ for all $x_1, x_2 \in \mathcal{K}$ such that $\|x_1 - x_2\| \geq R$.

We assume that the initial second moment is bounded above as $\mathbb{E}[\|\mathbf{x}_0\|^2] \leq \varsigma < \infty$.

Throughout the paper, $\mathcal{K}$ will denote a polyhedral subset of $\mathbb{R}^n$ with $0$ in its interior.

## 3 Main results

### 3.1 Convergence of the law of the iterates

For $\bar{f}$ defined above, the associated Gibbs measure is defined by:

$$\pi_{\beta\bar{f}}(A) = \frac{\int_{A \cap \mathcal{K}} e^{-\beta\bar{f}(x)} dx}{\int_{\mathcal{K}} e^{-\beta\bar{f}(x)} dx}. \tag{6}$$

The main result of this paper is stated next:

**Theorem 1.** *Assume that $\eta \leq \min\left\{\frac{1}{4}, \frac{\mu}{4\ell^2}\right\}$, $\mathcal{K}$ is a polyhedron with $0$ in its interior, $\mathbf{x}_0 \in \mathcal{K}$, and $\mathbb{E}[\|\mathbf{x}_0\|^2] \leq \varsigma$. There are constants $a$, $c_1$, $c_2$, $c_3$, and $c_4$ such that the following bound holds for all integers $k \geq 4$:*

$$W_1(\mathcal{L}(\mathbf{x}_k), \pi_{\beta\bar{f}}) \leq (c_1 + c_2\sqrt{\varsigma})e^{-\eta ak} + (c_3 + c_4\sqrt{\varsigma})\sqrt{\eta \log(\eta^{-1})}.$$

*In particular, if $\eta = \frac{\log T}{2aT}$, $T \geq 4$ and $T \geq e^{2a}$, then*

$$W_1(\mathcal{L}(\mathbf{x}_T), \pi_{\beta\bar{f}}) \leq \left(c_1 + c_2\sqrt{\varsigma} + \frac{c_3 + c_4\sqrt{\varsigma}}{(2a)^{1/2}}\right) T^{-1/2} \log T.$$

*Furthermore, the constants, $c_1, c_2, c_3$, and $c_4$ are $O(n)$ with respect to the dimension of $\mathbf{x}_k$, and $O(e^{\ell\beta R^2/2})$ with respect to the inverse temperature, $\beta$. And for all $\beta > 0$, $a \geq \frac{2}{\frac{\beta R^2}{2} + \frac{16}{\mu}} e^{-\frac{\beta\ell R^2}{4}}$.*

The constants depend on the dimension of $\mathbf{x}_k$, $n$, the noise parameter, $\beta$, the Lipschitz constant, $\ell$, the strong convexity constant $\mu$, the variance bound of the initial states, $\varsigma$, and some geometric properties of the polyhedron, $\mathcal{K}$.

The constants shown in Theorem 1 are described explicitly in Appendix H.

### 3.2 Auxiliary processes for convergence analysis

Similar to the previous analyses of Langevin methods, e.g. [9, 10, 27, 35], the proof of Theorem 1 uses a collection of auxiliary processes fitting between the algorithms iterates from (1) and a stationary distribution given by (6).

The algorithm and a variation in which the $\mathbf{z}_t$ variables are averaged out are respectively given by:

$$\mathbf{x}_{t+1}^A = \Pi_{\mathcal{K}} \left( \mathbf{x}_t^A - \eta \nabla_x f(\mathbf{x}_t^A, \mathbf{z}_t) + \sqrt{\frac{2\eta}{\beta}} \hat{\mathbf{w}}_t \right) \tag{7a}$$

$$\mathbf{x}_{t+1}^M = \Pi_{\mathcal{K}} \left( \mathbf{x}_t^M - \eta \nabla_x \bar{f}(\mathbf{x}_t^M) + \sqrt{\frac{2\eta}{\beta}} \hat{\mathbf{w}}_t \right). \tag{7b}$$

Here $\mathbf{x}_t^A$ represents the Algorithm, while $\mathbf{x}_t^M$ represents a corresponding Mean process.

We embed the mean process in continuous time by setting $\mathbf{x}_t^M = \mathbf{x}_{\lfloor t \rfloor}^M$, where $\lfloor t \rfloor$ indicates floor function. The Gaussian noise $\hat{\mathbf{w}}_k$ can be realized as $\hat{\mathbf{w}}_k = \mathbf{w}_{k+1} - \mathbf{w}_k$ where $\mathbf{w}_t$ is a Brownian motion.

Let $\mathbf{x}_t^C$ denote a Continuous-time approximation of $\mathbf{x}_t^M$ defined by the following reflected stochastic differential equation (RSDE):

$$d\mathbf{x}_t^C = -\eta \nabla_x \bar{f}(\mathbf{x}_t^C)dt + \sqrt{\frac{2\eta}{\beta}}d\mathbf{w}_t - \mathbf{v}_t^C d\boldsymbol{\mu}^C(t). \tag{8}$$

Here $-\int_0^t \mathbf{v}_s^C d\boldsymbol{\mu}^C(s)$ is a bounded variation reflection process that ensures that $\mathbf{x}_t^C \in \mathcal{K}$ for all $t \geq 0$, as long as $\mathbf{x}_0^C \in \mathcal{K}$. In particular, the measure $\boldsymbol{\mu}^C$ is such that $\boldsymbol{\mu}^C([0,t])$ is finite, $\boldsymbol{\mu}^C$ supported on $\{s | \mathbf{x}_s^C \in \partial \mathcal{K}\}$, and $\mathbf{v}_s^C \in N_{\mathcal{K}}(\mathbf{x}_s^C)$ where $N_{\mathcal{K}}(x)$ is the normal cone of $\mathcal{K}$ at $x$. Lemma 10 in Appendix A shows that the reflection process is uniquely defined and $\mathbf{x}^C$ is the unique solution to the Skorokhod problem for the process defined by:

$$\mathbf{y}_t^C = \mathbf{x}_0^C + \sqrt{\frac{2\eta}{\beta}}\mathbf{w}_t - \eta \int_0^t \nabla_x \bar{f}(\mathbf{x}_s^C)ds. \tag{9}$$

See Appendix A for more details on the Skorokhod problem.

For compact notation, we denote the Skorokhod solution for a given trajectory, $\mathbf{y}$, by $\mathcal{S}(\mathbf{y})$. So, the fact that $\mathbf{x}^C$ is the solution to the Skorokhod problem for $\mathbf{y}^C$ will be denoted succinctly by $\mathbf{x}^C = \mathcal{S}(\mathbf{y}^C)$.

The basic idea behind the proof is to utilize the triangle inequality:

$$W_1(\mathcal{L}(\mathbf{x}_k^A), \pi_{\beta f}) \leq W_1(\mathcal{L}(\mathbf{x}_k^A), \mathcal{L}(\mathbf{x}_k^C)) + W_1(\mathcal{L}(\mathbf{x}_k^C), \pi_{\beta \bar{f}}). \tag{10}$$

and then bound each of the terms separately.

The second term is bounded by the following lemma:

**Lemma 2.** *Assume that $\mathbf{x}_0 \in \mathcal{K}$ and $\mathbb{E}[\|\mathbf{x}_0^C\|^2] \leq \varsigma$. There are positive constants $a$, $c_1$ and $c_2$ such that for all $t \geq 0$*

$$W_1(\mathcal{L}(\mathbf{x}_t^C), \pi_{\beta \bar{f}}) \leq (c_1 + c_2\sqrt{\varsigma})e^{-\eta a t}.$$

This result is based on an extension of the contraction results from Corollary 2 of [20] for SDEs to the case of the reflected SDEs. Appendix D steps through the methodology from [20] in order to derive $a$, $c_1$ and $c_2$ for our particular problem.

Most of the novel work in the paper focuses on deriving the following bound on $W_1(\mathcal{L}(\mathbf{x}_k^A), \mathcal{L}(\mathbf{x}_k^C))$:

**Lemma 3.** *Assume that $\mathbf{x}_0^A = \mathbf{x}_0^C \in \mathcal{K}$, $\mathbb{E}[\|\mathbf{x}_0^C\|^2] \leq \varsigma$, and $\eta \leq \min\{\frac{1}{4}, \frac{\mu}{8\ell^2}\}$. Then there are positive constants $c_3$ and $c_4$ such that for all integers $k \geq 0$:*

$$W_1(\mathcal{L}(\mathbf{x}_k^A), \mathcal{L}(\mathbf{x}_k^C)) \leq (c_3 + c_4\sqrt{\varsigma})\sqrt{\eta \log(\eta^{-1})}.$$

**Proof of Theorem 1** Plugging the results of Lemmas 2 and 3 into the triangle inequality bound from (10) proves the first result of the theorem. Specifically, let $\eta = \frac{\log T}{2aT}$, then

$$W_1(\mathcal{L}(\mathbf{x}_T), \pi_{\beta \bar{f}}) \leq (c_1 + c_2\sqrt{\varsigma})T^{-1/2} + (c_3 + c_4\sqrt{\varsigma})\sqrt{\frac{\log T}{2aT}\log(\frac{2aT}{\log T})}$$

$$\leq (c_1 + c_2\sqrt{\varsigma})T^{-1/2}\log T + \frac{c_3 + c_4\sqrt{\varsigma}}{(2a)^{1/2}}T^{-1/2}\log T.$$

This gives the specific bound in the theorem. The last inequality utilizes the fact that $\log T > 1$ for all $T \geq 4$ and $\frac{2aT}{\log T} \leq T$ when $T \geq e^{2a}$.

Furthermore, we examine the bounds of the constants $c_1$, $c_2$, $c_3$, $c_4$ and $a$ in Appendix H, where the dependencies of the convergence guarantee on state dimension $n$ and the inverse temperature parameter, $\beta$ can be observed directly. ∎

The rest of the paper focuses on proving Lemma 3.

### 3.3 Proof overview for Lemma 3

This subsection describes the main ideas in the proof of Lemma 3. The results highlighted here, and proved in the appendix, cover the main novel aspects of the current work. The first novelty, captured in Lemmas 4 and 5, is a new way to bound stochastic gradient Langevin schemes with L-mixing data from a Langevin method with the data variables averaged out. The key idea is a method for examining a collection of partially averaged processes. The second novelty is a tight quantitative bound on the deviation of discretized Langevin algorithms from their continuous-time counterparts when constrained to a polyhedron. This result is based on a new quantitative bound on Skorokhod solutions over polyhedra.

First we derive time-dependent bounds (i.e. bounds that depend on $k$) for $W_1(\mathcal{L}(\mathbf{x}_k^A), \mathcal{L}(\mathbf{x}_k^C))$. This is achieved by introducing a collection of intermediate processes and bounding their differences. Time-uniform bounds are then achieved by exploiting contractivity properties of $\mathbf{x}_t^C$.

To bound $W_1(\mathcal{L}(\mathbf{x}_k^A), \mathcal{L}(\mathbf{x}_k^C))$, we first use the triangle inequality:

$$W_1(\mathcal{L}(\mathbf{x}_k^A), \mathcal{L}(\mathbf{x}_k^C)) \leq W_1(\mathcal{L}(\mathbf{x}_k^A), \mathcal{L}(\mathbf{x}_k^M)) + W_1(\mathcal{L}(\mathbf{x}_k^M), \mathcal{L}(\mathbf{x}_k^C)). \tag{11}$$

We bound $W_1(\mathcal{L}(\mathbf{x}_k^A), \mathcal{L}(\mathbf{x}_k^M))$ via a collection of auxiliary processes in which the effect of $\mathbf{z}_k$ is partially averaged out. We bound $W_1(\mathcal{L}(\mathbf{x}_k^M), \mathcal{L}(\mathbf{x}_k^C))$ via a specialized discrete-time approximation of $\mathbf{x}_t^C$.

Now we construct the collection of partially averaged processes. Recall that $\mathbf{z}_k \in \mathcal{Z}$ is a stationary L-mixing process with respect to the $\sigma$-algebras $\mathcal{F}_k$ and $\mathcal{F}_k^+$. For $k < 0$, we set $\mathcal{F}_k = \{\emptyset, \mathcal{Z}\}$, i.e. the trivial $\sigma$-algebra. Let $\mathcal{G}_t$ be the filtration generated by the Brownian motion, $\mathbf{w}_t$.

Recall that for $k \in \mathbb{N}$, we set $\hat{\mathbf{w}}_k = \mathbf{w}_{k+1} - \mathbf{w}_k$. Define the following discrete-time processes:

$$\mathbf{x}_{k+1}^{M,s} = \Pi_{\mathcal{K}} \left( \mathbf{x}_k^{M,s} - \eta \mathbb{E}[\nabla_x f(\mathbf{x}_k^{M,s}, \mathbf{z}_k) | \mathcal{F}_{k-s} \vee \mathcal{G}_k] + \sqrt{\frac{2\eta}{\beta}} \hat{\mathbf{w}}_k \right) \tag{12a}$$

$$\mathbf{x}_{k+1}^{B,s} = \Pi_{\mathcal{K}} \left( \mathbf{x}_k^{B,s} - \eta \mathbb{E}[\nabla_x f(\mathbf{x}_k^{M,s}, \mathbf{z}_k) | \mathcal{F}_{k-s-1} \vee \mathcal{G}_k] + \sqrt{\frac{2\eta}{\beta}} \hat{\mathbf{w}}_k \right). \tag{12b}$$

Assume that all initial conditions are equal. In other words, $\mathbf{x}_0^A = \mathbf{x}_0^M = \mathbf{x}_0^{M,s} = \mathbf{x}_0^{B,s}$, for all $s \geq 0$. The iterations from (12a) define a family of algorithms in which the data variables are partially averaged, while $\mathbf{x}_k^{B,s}$ from (12b) corresponds to an auxiliary process that fits between $\mathbf{x}_k^{M,s}$ and $\mathbf{x}_k^{M,s+1}$. (Here "A" stands for algorithm, "M" stands for mean, and "B" stands for between.)

Note for $s = 0$, we have that $\mathbf{x}_k^{M,0} = \mathbf{x}_k^A$ and for $s > k$, we have that $\mathbf{x}_k^{M,s} = \mathbf{x}_k^M$. So, in order to bound $W_1(\mathcal{L}(\mathbf{x}_k^A), \mathcal{L}(\mathbf{x}_k^M))$, it suffices to bound $W_1(\mathcal{L}(\mathbf{x}_k^{M,s}), \mathcal{L}(\mathbf{x}_k^{B,s}))$ and $W_1(\mathcal{L}(\mathbf{x}_k^{B,s}), \mathcal{L}(\mathbf{x}_k^{M,s+1}))$ for all $s \geq 0$. These bounds are achieved in the following lemmas, which are proved in Appendix E.

**Lemma 4.** *For all $s \geq 0$ and all $k \geq 0$, the following bound holds:*

$$W_1\left( \mathcal{L}(\mathbf{x}_k^{M,s}), \mathcal{L}(\mathbf{x}_k^{B,s}) \right) \leq \mathbb{E}[\|\mathbf{x}_k^{M,s} - \mathbf{x}_k^{B,s}\|] \leq 2\ell\psi_2(s, \mathbf{z})\eta\sqrt{k}.$$

**Lemma 5.** *For all $s \geq 0$ and all $k \geq 0$, the following bound holds*

$$W_1\left( \mathcal{L}(\mathbf{x}_k^{B,s}), \mathcal{L}(\mathbf{x}_k^{M,s+1}) \right) \leq \mathbb{E}[\|\mathbf{x}_k^{B,s} - \mathbf{x}_k^{M,s+1}\|] \leq 2\ell\psi_2(s, \mathbf{z})\eta\sqrt{k} \left( e^{\eta k\ell} - 1 \right).$$

Now we define the discretized approximation of $\mathbf{x}_t^C$. For any initial $\mathbf{x}_0^D \in \mathcal{K}$, we define the following iteration on the integers:

$$\mathbf{x}_{k+1}^D = \Pi_{\mathcal{K}}(\mathbf{x}_k^D + \mathbf{y}_{k+1}^C - \mathbf{y}_k^C) = \Pi_{\mathcal{K}} \left( \mathbf{x}_k^D + \int_k^{k+1} \nabla\bar{f}(\mathbf{x}_s^C)ds + \sqrt{\frac{2\eta}{\beta}} \hat{\mathbf{w}}_k \right).$$

Recall that the process $\mathbf{y}^C$ is defined by (9).

Provided that $\mathbf{x}_0^D = \mathbf{x}_0^C$, we have that $\mathbf{x}^D = \mathcal{S}(\mathbf{y}^D) = \mathcal{S}(\mathcal{D}(\mathbf{y}^C))$, where $\mathcal{D}$ is the discretization operator that sets $\mathcal{D}(x)_t = x_{\lfloor t \rfloor}$ for any continuous-time trajectory $x_t$. Recall that $\mathcal{S}$ is the Skorokhod solution operator.

The approximation, $\mathbf{x}^D$, was utilized in [9, 27] to bound discretization errors. The next lemmas show how to bound $W_1(\mathcal{L}(\mathbf{x}_k^C), \mathcal{L}(\mathbf{x}_k^D))$ and $W_1(\mathcal{L}(\mathbf{x}_k^M), \mathcal{L}(\mathbf{x}_k^D))$, respectively. In particular, Lemma 6 is analogous to Propositions 2.4 and 3.6 of [9] and Lemma 9 of [27]. These earlier works end up with bounds of $O(\eta^{3/4}k^{1/2} + \sqrt{\eta \log k})$. It is shown in [27] that such bounds can be translated into time-uniform bounds of the form $\tilde{O}(\eta^{1/4})$. The bound from Lemma 6 is of the form $O(\eta k^{1/2} + \sqrt{\eta \log k})$, and we will see in the next subsection that this leads to a time-uniform bound of the form $\tilde{O}(\eta^{1/2})$.

**Lemma 6.** *Assume that $\mathcal{K}$ is a polyhedron with $0$ in its interior. Assume that $\mathbf{x}_0^C = \mathbf{x}_0^D \in \mathcal{K}$ and that $\mathbb{E}[\|\mathbf{x}_0^C\|] \leq \varsigma$. There are constants, $c_5$, $c_6$ and $c_7$ such that for all integers $k \geq 0$, the following bound holds:*

$$W_1\left(\mathcal{L}(\mathbf{x}_k^C), \mathcal{L}(\mathbf{x}_k^D)\right) \leq \mathbb{E}\left[\|\mathbf{x}_k^C - \mathbf{x}_k^D\|\right] \leq (c_5 + c_6\sqrt{\varsigma})\eta\sqrt{k} + c_7\sqrt{\eta \log(4k)}.$$

**Lemma 7.** *Assume that $\mathcal{K}$ is a polyhedron with $0$ in its interior. Assume that $\mathbf{x}_0^C = \mathbf{x}_0^D = \mathbf{x}_0^M \in \mathcal{K}$ and that $\mathbb{E}[\|\mathbf{x}_0^C\|] \leq \varsigma$. Then for all integers $k \geq 0$, the following bound holds*

$$W_1\left(\mathcal{L}(\mathbf{x}_k^M), \mathcal{L}(\mathbf{x}_k^D)\right) \leq \mathbb{E}\left[\|\mathbf{x}_k^M - \mathbf{x}_k^D\|\right] \leq \left((c_5 + c_6\sqrt{\varsigma})\eta\sqrt{k} + c_7\sqrt{\eta \log(4k)}\right)\left(e^{\eta \ell k} - 1\right).$$

We highlight that Lemma 6 utilizes the rather tight bounds on solutions to Skorokhod problems over a polyhedral domain shown in Theorem 9. The derivation of such tight bounds is one of the novelties of our work. More details will be discussed in Section 4 and Appendix A.

With all of the auxiliary processes defined and their differences, we have the following lemma, which gives a time-dependent bound on $W_1(\mathcal{L}(\mathbf{x}_k^A), \mathcal{L}(\mathbf{x}_k^C))$:

**Lemma 8.** *Assume that $\mathcal{K}$ is a polyhedron with $0$ in its interior. Assume that $\mathbf{x}_0^A = \mathbf{x}_0^C \in \mathcal{K}$ and that $\mathbb{E}[\|\mathbf{x}_0^A\|] \leq \varsigma$. There are constants, $c_6$, $c_7$ and $c_8$, such that for all $k \geq 0$, the following bound holds:*

$$W_1(\mathcal{L}(\mathbf{x}_k^A), \mathcal{L}(\mathbf{x}_k^C)) \leq \left((c_8 + c_6\sqrt{\varsigma})\eta\sqrt{k} + c_7\sqrt{\eta \log(4k)}\right)e^{\eta \ell k}.$$

**Proof of Lemma 8**   Recalling that $\mathbf{x}_k^{M,0} = \mathbf{x}_k^A$ and $\mathbf{x}_k^{M,k+1} = \mathbf{x}_k^M$ and using the triangle inequality gives:

$$W_1(\mathcal{L}(\mathbf{x}_k^A), \mathcal{L}(\mathbf{x}_k^M))$$
$$\leq \sum_{s=0}^{k} W_1(\mathcal{L}(\mathbf{x}_k^{M,s}), \mathcal{L}(\mathbf{x}_k^{M,s+1}))$$
$$\leq \sum_{s=0}^{k} \left(W_1(\mathcal{L}(\mathbf{x}_k^{M,s}), \mathcal{L}(\mathbf{x}_k^{B,s})) + W_1(\mathcal{L}(\mathbf{x}_k^{B,s}), \mathcal{L}(\mathbf{x}_k^{M,s+1}))\right)$$
$$\overset{\text{Lemmas 4 \& 5}}{\leq} \sum_{s=0}^{k} 2\ell\psi_2(s, \mathbf{z})\eta\sqrt{k}e^{\eta \ell k}$$
$$\leq 2\ell\Psi_2(\mathbf{z})\eta\sqrt{k}e^{\eta \ell k}. \tag{13}$$

Here $\psi_2(s, \mathbf{z})$ and $\Psi_2(\mathbf{z})$ are the terms that bound the decay of probabilistic dependence between the $\mathbf{z}_k$ variables, as defined in (3).

Similarly, we bound

$$W_1(\mathcal{L}(\mathbf{x}_k^M), \mathcal{L}(\mathbf{x}_k^C))$$
$$\leq W_1(\mathcal{L}(\mathbf{x}_k^M), \mathcal{L}(\mathbf{x}_k^D)) + W_1(\mathcal{L}(\mathbf{x}_k^D), \mathcal{L}(\mathbf{x}_k^C))$$
$$\overset{\text{Lemmas 6 \& 7}}{\leq} \left((c_5 + c_6\sqrt{\varsigma})\eta\sqrt{k} + c_7\sqrt{\eta \log(4k)}\right)e^{\eta \ell k}. \tag{14}$$

Plugging the bounds from (13) and (14) into (11) proves the lemma, with $c_8 = c_5 + 2\ell\Psi_2(\mathbf{z})$. ∎

The proof of Lemma 3 is completed by showing how the time-dependent bound from Lemma 8 can be turned into a bound that is independent of $k$. The technique used for this step is based on ideas from [10], and is shown in Appendix G.

# 4 Quantitative bounds on Skorokhod solutions over polyhedra

In this section, we present a result that enables our new bound between the continuous-time process $\mathbf{x}_t^C$ and the discretized process $\mathbf{x}_t^M$ when constrained to the set $\mathcal{K}$ defined by:

$$\mathcal{K} = \{x | a_i^\top x \le b_i \text{ for } i = 1, \ldots, m\}, \tag{15}$$

where $a_i$ are unit vectors.

As discussed in Section 3.3, the bound in Lemma 6 improves upon the corresponding results in earlier works [9, 27]. The improvement arises from the use of Theorem 9 below, which utilizes the explicit polyhedral structure of $\mathcal{K}$ to achieve a tighter bound than could be obtained for general convex constraint sets. It is a variation on an earlier result from [18]. The main distinction is that the proof in [18] is non-constructive, and so there is no way to calculate the constants, whereas the proof in Appendix A is fully constructive and the constants can be computed explicitly.

**Theorem 9.** *There are constants $c_9$ and $\alpha \in (0, 1/2]$ such that if $x = \mathcal{S}(y)$ and $x' = \mathcal{S}(y')$ are Skorokhod solutions on the polyhedral set $\mathcal{K}$ defined by (15), then for all $t \ge 0$, the following bound holds:*

$$\sup_{0 \le s \le t} \|x_s - x'_s\| \le (c_9 + 1) \sup_{0 \le s \le t} \|y_s - y'_s\|.$$

*Here*

$$c_9 = 6 \left(\frac{1}{\alpha}\right)^{\text{rank}(A)/2}$$

*and $A = \begin{bmatrix} a_1 & \cdots a_m \end{bmatrix}^\top$ whose rows are the $a_i^\top$ vectors.*

# 5 Limitations

Our current work is restricted to polyhedral sets. In particular, Theorem 9 requires the polyhedral assumption, and it is unclear if Skorokhod problems satisfy similar bounds on any more general classes of constraint sets. As a result, it is unclear if our main results on projected Langevin algorithms can be extended beyond polyhedra. We also only considered constant step sizes, but in many cases decreasing or adaptive step sizes are used in practice. Finally, the dependence of the external data variables is limited to the class of L-mixing processes, which does not include all the real-world dependent data streams. Furthermore, it can be difficult to check that a data stream is L-mixing without requiring strong assumptions or knowledge about how it is generated.

# 6 Conclusions and future work

In this paper, we derived non-asymptotic bounds in 1-Wasserstein distance for a constrained Langevin algorithm applied to non-convex functions with dependent data streams satisfying L-mixing assumptions. Our convergence bounds match the best known bounds of the unconstrained case up to logarithmic factors, and improve on all existing bounds from the constrained case. The tighter bounds are enabled by a constructive and explicit bound on Skorokhod solutions, which builds upon an earlier non-constructive bound from [18]. The analysis of L-mixing variables followed by a comparatively simple averaging method. Future work will examine extensions beyond polyhedral domains, higher-order Langevin algorithms, alternative approaches to handling constraints, such as mirror descent, and more sophisticated step size rules. More specifically, future work will examine whether the projection step, and thus Skorkhod problems, can be circumvented by utilizing different algorithms, such as those based on proximal LMC [7]. Additionally, applications to real-world problems such as time-series analysis and adaptive control will be studied.

# 7 Acknowledgments

This work was supported in part by NSF CMMI-2122856. The authors thank the reviewers for helpful suggestions for improving the paper.

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
