# A  Background and results on Skorokhod problems

In this section, we will show that when the domain is a polyhedron, rather tight bounds on solutions to Skorokhod problems can be obtained.

## A.1  Background on Skorokhod problems

Let $\mathcal{K}$ be a convex subset of $\mathbb{R}^n$ with non-empty interior. Let $y : [0, \infty) \to \mathbb{R}^n$ be a trajectory which is right-continuous with left limits and has $y_0 \in \mathcal{K}$. For each $x \in \mathbb{R}^n$, let $N_{\mathcal{K}}(x)$ be the normal cone at $x$. Then the functions $x_t$ and $\phi_t$ solve the *Skorokhod problem* for $y_t$ if the following conditions hold:

- $x_t = y_t + \phi_t \in \mathcal{K}$ for all $t \in [0, T)$.

- The function $\phi$ has the form $\phi(t) = -\int_0^t v_s d\mu(s)$, where $\|v_s\| \in \{0, 1\}$ and $v_s \in N_{\mathcal{K}}(x_s)$ for all $s \in [0, T)$, while the measure, $\mu$, satisfies $\mu([0, T)) < \infty$ for any $T > 0$.

It can be shown that if a solution exists, it is unique. See [40]. However, existence of solutions typically relies on extra requirements beyond just convexity. For example, [40] showed the existence of solutions in the case that $y$ is continuous and $\mathcal{K}$ is compact. Below, we will utilize results from [2] to prove existence in the case that $\mathcal{K}$ is a polyhedron. Whenever solutions are guaranteed to exist, uniqueness implies that we may view the Skorokhod solution as a mapping: $x = \mathcal{S}(y)$.

## A.2  Existence of solutions over polyhedra

The following is a consequence of Theorem 4 from [2].

**Lemma 10.** *Let $\mathcal{K}$ be a polyhedron with non-empty interior. If $y_t$ is a trajectory in $\mathbb{R}^n$ which is right-continuous with left-limits, then $x = \mathcal{S}(y)$ exists, is unique, and is right-continuous with left-limits.*

**Proof**  To verify the conditions of Theorem 4 from [2], we just need to show that $\mathcal{K}$ satisfies condition $\beta$ of that paper, which states that there exist constants $\epsilon > 0$ and $\bar{\delta} > 0$ such that for all $x \in \partial \mathcal{K}$, there exist $x_0 \in \mathcal{K}$ such that $\|x - x_0\| \leq \bar{\delta}$ and $\{y | \|y - x_0\| < \epsilon\} \subset \mathcal{K}$. We will show how to construct $\epsilon, \bar{\delta}$, and we will see that a suitable vector, $x_0$, exists for any $x \in \mathcal{K}$.

Note that since $\mathcal{K}$ is a polyhedron, there are vectors $u_1, \ldots, u_p$ such that $x \in \mathcal{K}$ if and only if it can be expressed as

$$x = \sum_{i=1}^{k} \lambda_i u_i + \sum_{i=k+1}^{p} \lambda_i u_i$$

with $\lambda_i \geq 0$ for $i = 1, \ldots, p$ and $\sum_{i=1}^{k} \lambda_i = 1$. See [38]. (If $p = k$, then $\mathcal{K}$ is a compact polytope, while if $k = 0$, then $\mathcal{K}$ is a convex cone.)

Let $x^\star$ be an arbitrary point in the interior of $\mathcal{K}$ and let $\epsilon > 0$ be such that $\{y | \|y - x^\star\| < \epsilon\} \subset \mathcal{K}$. Pick $\bar{\delta}$ such that $\|u_i - x^\star\| \leq \bar{\delta}$ for $i = 1, \ldots, k$.

For any $x = \sum_{i=1}^{p} \lambda_i u_i \in \mathcal{K}$, let $x_0 = x^\star + \sum_{i=k+1}^{p} \lambda_i u_i$. It follows that

$$
\begin{aligned}
\|x - x_0\| &= \left\| \sum_{i=1}^{k} \lambda_i u_i - x^\star \right\| \\
&= \left\| \sum_{i=1}^{k} \lambda_i (u_i - x^\star) \right\| \\
&\leq \sum_{i=1}^{k} \lambda_i \|u_i - x^\star\| \\
&\leq \bar{\delta}.
\end{aligned}
$$

Also, if $y \in \{y | \|y - x_0\| < \epsilon\}$, then there is a vector, $v$, with $\|v\| < \epsilon$, such that

$$y = x_0 + v = (x^\star + v) + \sum_{i=k+1}^{p} \lambda_i u_i.$$

Now note that $x^\star + v \in \mathcal{K}$, so there must be numbers $\lambda_i' \geq 0$ such that $\sum_{i=1}^{k} \lambda_i' = 1$ and $x^\star + v = \sum_{i=1}^{p} \lambda_i' u_i$. It follows that

$$y = \sum_{i=1}^{p} \lambda_i' u_i + \sum_{i=k+1}^{p} \lambda_i u_i = \sum_{i=1}^{k} \lambda_i' u_i + \sum_{i=k+1}^{p} (\lambda_i + \lambda_i') u_i \in \mathcal{K}.$$

∎

### A.3  Proof of Theorem 9

In this subsection, we provide a short proof of Theorem 9. A supporting Lemma is firstly presented to complete the proof.

The technical work in this subsection relies on some notation about the vectors defining $\mathcal{K}$ from (15). Let $A = [a_1 \ \cdots a_m]^\top$ be the matrix whose rows are the $a_i^\top$ vectors. For $\mathcal{I} \subset \{1, \ldots, m\}$ let $A_\mathcal{I}$ be the matrix whose rows are $a_i^\top$ for $i \in \mathcal{I}$. Let $[W_\mathcal{I} \ \ V_\mathcal{I}]$ be an orthogonal matrix such that $\mathcal{N}(A_\mathcal{I}) = \mathcal{R}(W_\mathcal{I})$. Here $\mathcal{N}(A_\mathcal{I})$ denotes the null space of $A_\mathcal{I}$ and $\mathcal{R}(W_\mathcal{I})$ denotes the range space of $W_\mathcal{I}$. Let $P_\mathcal{I} = W_\mathcal{I} W_\mathcal{I}^\top$, which is the orthogonal projection onto $\mathcal{N}(A_\mathcal{I})$. We will use the convention that $A_\emptyset$ is a $1 \times n$ matrix of zeros, so that $\mathcal{N}(A_\emptyset) = \mathbb{R}^n$, and thus $P_\emptyset = I$.

The following lemma is a quantitative and explicit version of Theorem 2.1 of [18]:

**Lemma 11.** *If $\mathcal{K}$ is a polyhedron defined by (15), then there is a compact, convex set $\mathcal{B}$ with $0 \in \mathrm{int}(\mathcal{B})$ such that if $z \in \partial \mathcal{B}$, $v \in N_\mathcal{B}(z)$, and $a_j$ is a unit vector from (15) with $a_j^\top v \neq 0$, then*

*1. $|a_j^\top z| \geq 1$*

*2. $\mathrm{sign}(a_j^\top z) = \mathrm{sign}(a_j^\top v)$.*

*Furthermore, the diameter of $\mathcal{B}$ is at most $c_9$, defined by*

$$c_9 = 6 \left(\frac{1}{\alpha}\right)^{\mathrm{rank}(A)/2}$$

*where*

$$\alpha = \frac{1}{2} \min \left\{\|P_\mathcal{I} a_j\|^2 \big| P_\mathcal{I} a_j \neq 0, \ \mathcal{I} \subset \{1, \ldots, m\}, \ j \in \{1, \ldots, m\}\right\},$$

*and $\alpha \in (0, 1/2]$.*

A non-constructive proof of the existence of $\mathcal{B}$ was given in [18]. While that paper shows that $\mathcal{B}$ is compact, it does not quantitatively bound its diameter. The diameter of $\mathcal{B}$ is precisely the quantity that is used to bound the difference between Skorokhod solutions.

**Proof of Theorem 9.**  Theorem 2.2 of [18] shows that if a compact convex set with $0 \in \mathrm{int}(B)$ satisfying conditions 1 and 2 exists, then

$$\sup_{0 \leq s \leq t} \|x_s - x_s'\| \leq (\mathrm{diameter}(\mathcal{B}) + 1) \sup_{0 \leq s \leq t} \|y_s - y_s'\|.$$

The result now follows since $c_9$ is an upper bound on the diameter of the set $\mathcal{B}$ constructed in Lemma 11. ∎

**Proof of Lemma 11.**  We will focus on constructing a compact, convex $\mathcal{B}$ with $0 \in \mathrm{int}(\mathcal{B})$ which satisfies condition 1. Lemma 2.1 of [18] shows that condition 2 must also hold. (Note that the sign is opposite of what appears in [18], because that paper examines inward normal vectors, while we are examining outward normal vectors.)

We will find numbers $\epsilon \in (0,1)$ and $r_{\mathcal{I}} \in (0,1)$ for $\mathcal{I} \subset \{1, \ldots, m\}$ such that

$$\mathcal{B} = \{x \mid \|P_{\mathcal{I}}x\| \leq \epsilon^{-1}r_{\mathcal{I}} \text{ for } \mathcal{I} \in \{1, \ldots, m\}\}$$

has the desired properties. By construction, $\mathcal{B}$ is compact and convex, $0 \in \mathrm{int}(\mathcal{B})$, and the diameter is at most $2\epsilon^{-1}r_{\emptyset} < 2\epsilon^{-1}$, since every $x \in \mathcal{B}$ satisfies $\|P_{\emptyset}x\| = \|x\| \leq \epsilon^{-1}r_{\emptyset}$. Furthermore, $\mathcal{B} = \epsilon^{-1}\hat{\mathcal{B}}$, where

$$\hat{\mathcal{B}} = \{x \mid \|P_{\mathcal{I}}x\| \leq r_{\mathcal{I}} \text{ for } \mathcal{I} \subset \{1, \ldots, m\}\}.$$

A similar construction for $\mathcal{B}$ was utilized in [18]. The main distinction is that this proof will give an explicit procedure for determining the values of $\epsilon$ and $r_{\mathcal{I}}$.

Note that $z \in \hat{\mathcal{B}}$ if and only if $\epsilon^{-1}z \in \mathcal{B}$, $z \in \partial\hat{\mathcal{B}}$ if and only if $\epsilon^{-1}z \in \partial\mathcal{B}$, and $N_{\hat{\mathcal{B}}}(z) = N_{\mathcal{B}}(\epsilon^{-1}z)$. Thus, Condition 1 holds for $\mathcal{B}$ if and only if

$$z \in \partial\hat{\mathcal{B}}, \ v \in N_{\hat{\mathcal{B}}}(z), \ \text{ and } a_j^{\top}v \neq 0 \implies |a_j^{\top}z| \geq \epsilon > 0. \tag{16}$$

Note that if $x \in \partial\hat{\mathcal{B}}$, then

$$N_{\hat{\mathcal{B}}}(x) = \mathrm{cone}\{P_{\mathcal{I}}x \mid \|P_{\mathcal{I}}x\| = r_{\mathcal{I}}\}$$

$$= \left\{ \sum_{\{\mathcal{I} \mid \|P_{\mathcal{I}}x\| = r_{\mathcal{I}}\}} \lambda_{\mathcal{I}}P_{\mathcal{I}}x \,\middle|\, \lambda_{\mathcal{I}} \geq 0 \right\}. \tag{17}$$

See Corollary 23.8.1 of [38].

The representation in (17) implies that if $x \in \partial\hat{\mathcal{B}}$, $v \in N_{\mathcal{B}}(x)$, and $a_j^{\top}v \neq 0$, then there must be a set $\mathcal{I}$ such that, $\|P_{\mathcal{I}}x\| = r_{\mathcal{I}}$, $\lambda_{\mathcal{I}} > 0$, and $a_j^{\top}P_{\mathcal{I}} \neq 0$. We will choose $\epsilon$ such that for all $\mathcal{I}$ and $j$ with $P_{\mathcal{I}}a_j \neq 0$, $\epsilon$ is a lower bound on the optimal value of the following (non-convex) optimization problem:

$$\min_{x} \qquad |a_j^{\top}x| \tag{18a}$$

$$\text{subject to} \qquad \|P_{\mathcal{I}}x\| \geq r_{\mathcal{I}} \tag{18b}$$

$$\|P_{\mathcal{I} \cup \{j\}}x\| \leq r_{\mathcal{I} \cup \{j\}} \tag{18c}$$

$$\|x\| \leq 1. \tag{18d}$$

By construction, if $x \in \partial\hat{\mathcal{B}}$, $v \in N_{\mathcal{B}}(x)$, and $a_j^{\top}v \neq 0$, there must be some $\mathcal{I}$ such that $x$ is feasible for (18). As a result, we must have that $|a_j^{\top}x| \geq \epsilon$. Thus, the implication from (16) will hold, provided that the values of $r_{\mathcal{I}}$ can be chosen so that all of the problems of the form (18) have strictly positive optimal values.

The rest of the proof proceeds as follows. First we derive conditions on $r_{\mathcal{I}}$ that ensure that the problems from (18) always have positive optimal values. Next, we compute specific values of $r_{\mathcal{I}}$ that satisfy these conditions. Finally, we use those values of $r_{\mathcal{I}}$ to compute $\epsilon$, the desired lower bound on the optimal value of (18).

We now assume that $r_{\mathcal{I}}, r_{\mathcal{I} \cup \{j\}} \in (0,1)$ and derive sufficient conditions to make the optimal value in (18) strictly positive.

To derive the optimal value of (18), we need a few basic facts:

- If $\mathcal{I} \subset \mathcal{J}$, then $P_{\mathcal{J}}P_{\mathcal{I}} = P_{\mathcal{J}}$ and $P_{\mathcal{I}}P_{\mathcal{J}} = P_{\mathcal{J}}$.
- The matrix $\left[ W_{\mathcal{I} \cup \{j\}} \quad \frac{P_{\mathcal{I}}a_j}{\|P_{\mathcal{I}}a_j\|} \quad V_{\mathcal{I}} \right]$ is orthogonal.

First we show that $\mathcal{I} \subset \mathcal{J}$ implies that $P_{\mathcal{J}}P_{\mathcal{I}} = P_{\mathcal{J}}$. Symmetry of the projection matrices would then imply that $P_{\mathcal{I}}P_{\mathcal{J}} = P_{\mathcal{J}}$. Note that $P_{\mathcal{I}} = I - V_{\mathcal{I}}V_{\mathcal{I}}^{\top}$, where

$$\mathcal{R}(V_{\mathcal{I}}) = \mathcal{R}(P_{\mathcal{I}})^{\perp} = \mathcal{N}(A_{\mathcal{I}})^{\perp} \subset \mathcal{N}(A_{\mathcal{J}})^{\perp} = \mathcal{R}(P_{\mathcal{J}})^{\perp}.$$

It follows that $P_{\mathcal{J}}V_{\mathcal{I}} = 0$ and thus $P_{\mathcal{J}}P_{\mathcal{I}} = P_{\mathcal{J}}$.

Now we will show that $P_{\mathcal{I}}a_j \in \mathcal{R}(P_{\mathcal{I}})\backslash\mathcal{R}(P_{\mathcal{I}\cup\{j\}})$. By construction, $P_{\mathcal{I}}a_j \in \mathcal{R}(P_{\mathcal{I}})$. Also, we have that $P_{\mathcal{I}\cup\{j\}}P_{\mathcal{I}}a_j = P_{\mathcal{I}\cup\{j\}}a_j = 0$, where the second equality follows because $a_j \in \mathcal{N}(A_{\mathcal{I}\cup\{j\}})^{\perp} = \mathcal{R}(P_{\mathcal{I}\cup\{j\}})^{\perp}$. Thus, we have that $P_{\mathcal{I}}a_j \neq P_{\mathcal{I}\cup\{j\}}P_{\mathcal{I}}a_j$. Now, if $P_{\mathcal{I}}a_j \in \mathcal{R}(P_{\mathcal{I}\cup\{j\}})$, then $P_{\mathcal{I}}a_j = P_{\mathcal{I}\cup\{j\}}z$ for some vector $z$. But then $P_{\mathcal{I}\cup\{j\}}^2 = P_{\mathcal{I}\cup\{j\}}$ would imply that $P_{\mathcal{I}\cup\{j\}}P_{\mathcal{I}}a_j = P_{\mathcal{I}\cup\{j\}}z = P_{\mathcal{I}}a_j$, which gives a contradiction. Thus, $P_{\mathcal{I}}a_j \notin \mathcal{R}(P_{\mathcal{I}\cup\{j\}})$.

Now the rank nullity theorem implies that

$$\mathrm{rank}(A_{\mathcal{I}}) = n - \dim(\mathcal{N}(A_{\mathcal{I}}))$$
$$\mathrm{rank}(A_{\mathcal{I}\cup\{j\}}) = n - \dim(\mathcal{N}(A_{\mathcal{I}\cup\{j\}})).$$

Now since $A_{\mathcal{I}\cup\{j\}}$ has only one more row than $A_{\mathcal{I}}$, we must have that $\mathrm{rank}(A_{\mathcal{I}}) \leq \mathrm{rank}(A_{\mathcal{I}\cup\{j\}}) \leq \mathrm{rank}(A_{\mathcal{I}}) + 1$. Also, $\mathcal{N}(A_{\mathcal{I}\cup\{j\}}) \subset \mathcal{N}(A_{\mathcal{I}})$ by construction, and we just saw that $P_{\mathcal{I}}a_j \in \mathcal{N}(A_{\mathcal{I}})\backslash\mathcal{N}(A_{\mathcal{I}\cup\{j\}})$, so the inclusion is strict. It follows that

$$\begin{aligned}
\dim(\mathcal{N}(A_{\mathcal{I}})) &= \dim(\mathcal{R}(P_{\mathcal{I}})) \\
&= \dim(\mathcal{N}(A_{\mathcal{I}\cup\{j\}})) + 1 \\
&= \dim(\mathcal{R}(P_{\mathcal{I}\cup\{j\}})) + 1.
\end{aligned} \tag{19}$$

Now, since $\mathcal{R}(W_{\mathcal{I}\cup\{j\}}) = \mathcal{N}(A_{\mathcal{I}\cup\{j\}})$, we must have that

$$\mathcal{R}\left(\begin{bmatrix} W_{\mathcal{I}\cup\{j\}} & \frac{P_{\mathcal{I}}a_j}{\|P_{\mathcal{I}}a_j\|} \end{bmatrix}\right) = \mathcal{N}(A_{\mathcal{I}}).$$

Furthermore, since $\mathcal{R}(W_{\mathcal{I}\cup\{j\}}) = \mathcal{R}(P_{\mathcal{I}\cup\{j\}})$ and $P_{\mathcal{I}\cup\{j\}}P_{\mathcal{I}}a_j = 0$, we must have that

$$\begin{bmatrix} W_{\mathcal{I}\cup\{j\}}^{\top} \\ \frac{(P_{\mathcal{I}}a_j)^{\top}}{\|P_{\mathcal{I}}a_j\|} \\ V_{\mathcal{I}}^{\top} \end{bmatrix} \begin{bmatrix} W_{\mathcal{I}\cup\{j\}} & \frac{P_{\mathcal{I}}a_j}{\|P_{\mathcal{I}}a_j\|} & V_{\mathcal{I}} \end{bmatrix} = I$$

Now we use this orthogonal matrix to perform a change of coordinates. In particular, let $y_1$, $y_2$, and $y_3$ be such that

$$x = W_{\mathcal{I}\cup\{j\}}y_1 + \frac{P_{\mathcal{I}}a_j}{\|P_{\mathcal{I}}a_j\|}y_2 + V_{\mathcal{I}}y_3.$$

In these new coordinates, (18) is equivalent to

$$\min_{y} \qquad \|\|P_{\mathcal{I}}a_j\|y_2 + a_j^{\top}V_{\mathcal{I}}y_3\| \tag{20a}$$

$$\text{subject to} \qquad \|y_1\|^2 + y_2^2 \geq r_{\mathcal{I}}^2 \tag{20b}$$

$$\|y_1\| \leq r_{\mathcal{I}\cup\{j\}} \tag{20c}$$

$$\|y_1\|^2 + y_2^2 + \|y_3\|^2 \leq 1. \tag{20d}$$

The equivalence arises because

$$a_j^{\top}x = \|P_{\mathcal{I}}a_j\|y_2 + a_j^{\top}V_{\mathcal{I}}y_3$$
$$P_{\mathcal{I}}x = W_{\mathcal{I}\cup\{j\}}y_1 + \frac{P_{\mathcal{I}}a_j}{\|P_{\mathcal{I}}a_j\|}y_2$$
$$P_{\mathcal{I}\cup\{j\}}x = W_{\mathcal{I}\cup\{j\}}y_1$$

along with orthogonality of the corresponding transformation from $y$ to $x$.

If we choose $r_{\mathcal{I}} > r_{\mathcal{I}\cup\{j\}}$, then we must have

$$y_2^2 \geq r_{\mathcal{I}}^2 - \|y_1\|^2 \geq r_{\mathcal{I}}^2 - r_{\mathcal{I}\cup\{j\}}^2 > 0.$$

Now, if $y$ is feasible, $-y$ is also feasible, and they have the same objective value in (20). So, without loss of generality, we may assume that $y_2 > 0$.

The Cauchy-Schwartz inequality, combined with (20d), implies that

$$\|P_{\mathcal{I}}a_j\|y_2 + a_j^\top V_{\mathcal{I}}y_3 \geq \|P_{\mathcal{I}}a_j\|y_2 - \|V_{\mathcal{I}}^\top a_j\|\sqrt{1 - \|y_1\|^2 - y_2^2}. \tag{21}$$

Note that this bound is achieved by setting $y_3 = -\frac{V_{\mathcal{I}}^\top a_j}{\|V_{\mathcal{I}}^\top a_j\|}\sqrt{1 - \|y_1\|^2 - y_2^2}$.

The right side of (21) is monotonically increasing in $y_2$. So, (20b) implies that it is minimized over $y_2$ by setting $y_2 = \sqrt{r_{\mathcal{I}}^2 - \|y_1\|^2}$. This leads to a lower bound of the form:

$$\|P_{\mathcal{I}}a_j\|y_2 - \|V_{\mathcal{I}}^\top a_j\|\sqrt{1 - \|y_1\|^2 - y_2^2} \geq \|P_{\mathcal{I}}a_j\|\sqrt{r_{\mathcal{I}}^2 - \|y_1\|^2} - \|V_{\mathcal{I}}^\top a_j\|\sqrt{1 - r_{\mathcal{I}}^2}.$$

The right side is now monotonically decreasing with respect to $\|y_1\|$, and so it is minimized by setting $\|y_1\| = r_{\mathcal{I}\cup\{j\}}$. This leads to the characterization:

Optimal Value of (20)

$$= \|P_{\mathcal{I}}a_j\|\sqrt{r_{\mathcal{I}}^2 - r_{\mathcal{I}\cup\{j\}}^2} - \|V_{\mathcal{I}}^\top a_j\|\sqrt{1 - r_{\mathcal{I}}^2}$$

$$= \|P_{\mathcal{I}}a_j\|\sqrt{r_{\mathcal{I}}^2 - r_{\mathcal{I}\cup\{j\}}^2} - \sqrt{1 - \|P_{\mathcal{I}}^\top a_j\|^2}\sqrt{1 - r_{\mathcal{I}}^2}. \tag{22}$$

The second equality follows because

$$\|V_{\mathcal{I}}^\top a_j\|^2 = a_j^\top V_{\mathcal{I}}V_{\mathcal{I}}^\top a_j = a_j^\top(I - P_{\mathcal{I}})a_j = 1 - \|P_{\mathcal{I}}a_j\|^2.$$

Now, we have that the right side of (22) is positive if and only if:

$$\|P_{\mathcal{I}}a_j\|^2\left(r_{\mathcal{I}}^2 - r_{\mathcal{I}\cup\{j\}}^2\right) > \left(1 - \|P_{\mathcal{I}}a_j\|^2\right)\left(1 - r_{\mathcal{I}}^2\right) \tag{23a}$$

$$\iff r_{\mathcal{I}}^2 > 1 - \|P_{\mathcal{I}}a_j\|^2 + \|P_{\mathcal{I}}a_j\|^2 r_{\mathcal{I}\cup\{j\}}^2 \tag{23b}$$

$$\iff r_{\mathcal{I}}^2 > 1 - \|P_{\mathcal{I}}a_j\|^2(1 - r_{\mathcal{I}\cup\{j\}}^2) \tag{23c}$$

$$\iff r_{\mathcal{I}}^2 > r_{\mathcal{I}\cup\{j\}}^2 + (1 - \|P_{\mathcal{I}}a_j\|^2)(1 - r_{\mathcal{I}\cup\{j\}}^2). \tag{23d}$$

Note that (23d) implies that $r_{\mathcal{I}} > r_{\mathcal{I}\cup\{j\}}$ holds.

Also note that any collection of $r_{\mathcal{I}}$ values in $(0,1)$ that satisfy (23) will ensure that the corresponding set, $\hat{\mathcal{B}}$, satisfies the implication from (16). In that case, we have that $\mathcal{B}$ has the desired properties.

Now we seek a simpler, more explicit formula for the $r_{\mathcal{I}}$ values which satisfy (23). Note that (23c) implies that the right side is monotonically decreasing with respect to $\|P_{\mathcal{I}}a_j\|^2$. So, if $\alpha > 0$ is a number such that $\alpha \leq \frac{1}{2}\|P_{\mathcal{I}}a_j\|^2$ for all $\mathcal{I}$ and $j$ with $P_{\mathcal{I}}a_j \neq 0$ we obtain a sufficient condition for (23):

$$r_{\mathcal{I}}^2 = 1 - \alpha(1 - r_{\mathcal{I}\cup\{j\}}^2) \tag{24a}$$

$$r_{\mathcal{I}}^2 = (1 - \alpha) + \alpha r_{\mathcal{I}\cup\{j\}}^2. \tag{24b}$$

Now we use (24) to derive the desired formula for $r_{\mathcal{I}}$. In particular, consider the recursion

$$x_{k+1} = (1 - \alpha) + \alpha x_k.$$

This has an explicit solution given by

$$x_k = \alpha^k x_0 + 1 - \alpha^k = 1 - \alpha^k(1 - x_0).$$

In particular, if $x_0 \in (0,1)$, we have that $x_k \in (0,1)$ for all $k \geq 0$.

We define $r_{\mathcal{I}}$ by fixing a value $x_0 \in (0,1)$, which will be defined explicitly later, and setting $r_{\mathcal{I}}^2 = x_k = 1 - \alpha^k(1 - x_0)$ if $\text{rank}(A) - \text{rank}(A_{\mathcal{I}}) = k$.

To see that this definition satisfies (24), first note that $r_{\mathcal{I}}^2 = x_0$ for all $\mathcal{I}$ with $\text{rank}(A) = \text{rank}(A_{\mathcal{I}})$. Now, recall that if $P_{\mathcal{I}}a_j \neq 0$, then (19) implies that $\text{rank}(A_{\mathcal{I}\cup\{j\}}) = \text{rank}(A_{\mathcal{I}}) + 1$. The converse is also true: If $\text{rank}(A_{\mathcal{I}\cup\{j\}}) = \text{rank}(A_{\mathcal{I}}) + 1$, then we must have that $a_j \notin \mathcal{R}(A_{\mathcal{I}}^\top) = \mathcal{R}(V_{\mathcal{I}}) =$

$\mathcal{R}(P_{\mathcal{I}})^{\perp}$. It follows that $P_{\mathcal{I}}a_j \neq 0$. Thus, if $\operatorname{rank}(A) - \operatorname{rank}(A_{\mathcal{I}\cup\{j\}}) = k \geq 0$, we have that $P_{\mathcal{I}}a_j \neq 0$ precisely when $\operatorname{rank}(A) - \operatorname{rank}(A_{\mathcal{I}}) = k+1$. So we see that setting $r_{\mathcal{I}}^2 = x_{k+1} = 1 - \alpha(1 - x_k)$ gives the same value as specified in (24).

The final step in the proof requires finding a lower bound, $\epsilon$, for the optimal value from (22). Let $r_{\mathcal{I}}^2 = x_k$ and $r_{\mathcal{I}\cup\{j\}}^2 = x_{k-1}$. Then we have that

$$r_{\mathcal{I}}^2 - r_{\mathcal{I}\cup\{j\}}^2 = (1-\alpha)\alpha^{k-1}(1-x_0)$$
$$1 - r_{\mathcal{I}}^2 = \alpha\alpha^{k-1}(1-x_0).$$

Also note that the right side of (22) is monotonically increasing with respect to $\|P_{\mathcal{I}}a_j\|^2$ and that $\|P_{\mathcal{I}}a_j\|^2 \geq 2\alpha$ by our choice of $\alpha$. So, plugging in this lower bound gives

$$\|P_{\mathcal{I}}a_j\|\sqrt{r_{\mathcal{I}}^2 - r_{\mathcal{I}\cup\{j\}}^2} - \sqrt{1 - \|P_{\mathcal{I}}^\top a_j\|^2}\sqrt{1 - r_{\mathcal{I}}^2}$$
$$\geq \left(\sqrt{2\alpha}\sqrt{1-\alpha} - \sqrt{1-2\alpha}\sqrt{\alpha}\right)\sqrt{\alpha^{k-1}(1-x_0)}$$
$$= \left(\sqrt{2-2\alpha} - \sqrt{1-2\alpha}\right)\sqrt{\alpha^k(1-x_0)}$$
$$\geq \left(\sqrt{2}-1\right)\sqrt{\alpha^{\operatorname{rank}(A)}(1-x_0)}$$

The final inequality follows because $k \leq \operatorname{rank}(A)$ and the minimum value of $\sqrt{2-2\alpha} - \sqrt{1-2\alpha}$ over $\alpha \in [0, \|P_{\mathcal{I}}a_j\|^2/2] \subset [0, 1/2]$ occurs at $\alpha = 0$.

To simplify the final formula for $\epsilon$, note that $\sqrt{2} - 1 > 1/3$, and thus we can choose $x_0 \in (0,1)$ so that

$$(\sqrt{2}-1)\sqrt{1-x_0} = \frac{1}{3} \iff x_0 = 1 - \frac{1}{9\left(\sqrt{2}-1\right)^2} \approx 0.352.$$

Plugging in this value for $x_0$ gives the bound:

$$\text{Optimal Value of (20)} \geq \frac{1}{3}\alpha^{\frac{\operatorname{rank}(A)}{2}} =: \epsilon$$

Now recalling that the diameter of $\mathcal{B}$ is at most $2/\epsilon$ completes the proof. ∎

# B  Invariance of the Gibbs measure

**Lemma 12.** *The Gibbs measure, (6), is stationary under the dynamics of the reflected SDE from (8).*

**Proof** Before showing invariance of the Gibbs measure, we first remark that it is well-defined. In particular, we have that $\int_{\mathcal{K}} e^{-\beta\bar{f}(x)}dx < \infty$.

To see this, let $\|x\| \geq R/\theta$, where $\theta \in (0,1)$ is a number to be chosen later. Note that for $t \in [\theta, 1]$, we have that $\|\theta x\| \geq R$. So, we can use strong convexity outside a ball of radius $R$ to show

$$\bar{f}(x) \geq \bar{f}(0) + \int_0^1 \nabla\bar{f}(tx)^\top x\, dt$$
$$= \bar{f}(0) + \nabla\bar{f}(0)^\top x + \int_0^\theta \left(\nabla\bar{f}(tx) - \nabla\bar{f}(0)\right)^\top x\, dt + \int_\theta^1 \left(\nabla\bar{f}(tx) - \nabla\bar{f}(0)\right)^\top x\, dt$$
$$\geq \bar{f}(0) - \|\nabla\bar{f}(0)\|\|x\| - \ell\|x\|^2 \int_0^\theta t\, dt + \mu\|x\|^2 \int_\theta^1 t\, dt$$
$$\geq \bar{f}(0) - \|\nabla\bar{f}(0)\|\|x\| + \frac{1}{2}\|x\|^2 \left(-\ell\theta^2 + \mu(1-\theta^2)\right)$$

The coefficient $-\ell^2\theta^2 + \mu(1-\theta^2)$ is positive, as long as $\theta < \sqrt{\frac{\mu}{\mu+\ell}}$. In particular, choosing $\theta^2 = \frac{1}{2}\frac{\mu}{\mu+\ell}$ gives

$$\bar{f}(x) \geq \bar{f}(0) - \|\nabla\bar{f}(0)\|\|x\| + \frac{1}{4}\mu\|x\|^2. \tag{25}$$

It follows that $Z = \int_{\mathcal{K}} e^{-\beta \bar{f}(x)} dx < \infty$.

In [27], it was shown in that the Gibbs measure is invariant under (8) when $\mathcal{K}$ is compact. We will extend the result to non-compact $\mathcal{K}$ via a limiting argument.

Let $\mathcal{K}_i = \mathcal{K} \cap \{x \in \mathbb{R}^n | \|x\|_\infty \le i\}$. Let $Z_i = \int_{\mathcal{K}_i} e^{-\beta \bar{f}(x)} dx$. Note that $\lim_{i \to \infty} Z_i = Z$, by monotone convergence. We choose $\|x\|_\infty = \max\{|x_1|, \dots, |x_n|\} \le i$ so that $\mathcal{K}_i$ becomes a compact polyhedron for $i \ge 1$.

Let $\mathbf{x}_t^C$ be a solution to the original form of (8) and let $\mathbf{x}_t^{C,i}$ be a solution to the RSDE from (8), with $\mathcal{K}_i$ used in place of $\mathcal{K}$. Since $\mathcal{K}_i$ is polyhedral, Lemma 10 in Appendix A shows that $\mathbf{x}_t^{C,i}$ is uniquely defined. Define the diffusion operators $P$ and $P^i$ by

$$(P_t g)(x) = \mathbb{E}[g(\mathbf{x}_t^C)|\mathbf{x}_0 = x]$$
$$(P_t^i g)(x) = \mathbb{E}[g(\mathbf{x}_t^{C,i})|\mathbf{x}_0 = x]$$

Let $L_2(\mathcal{K}, \pi_{\beta \bar{f}})$ be the set of functions $g : \mathcal{K} \to \mathbb{R}$ which are square integrable with respect to the measure $\pi_{\beta \bar{f}}$. We will show that $\pi_{\beta \bar{f}}$ is invariant for (8) by showing that for all $g \in L_2(\mathcal{K}, \pi_{\beta \bar{f}})$ the following equality holds for all $t \ge 0$:

$$\frac{1}{Z} \int_{\mathcal{K}} g(x) e^{-\beta \bar{f}(x)} dx = \frac{1}{Z} \int_{\mathcal{K}} (P_t g)(x) e^{-\beta \bar{f}(x)} dx. \tag{26}$$

The subset of bounded, compactly supported functions in $L_2(\mathcal{K}, \pi_{\beta \bar{f}})$ is a dense subset. Fix an arbitary bounded, compactly supported $g \in L_2(\mathcal{K}, \pi_{\beta \bar{f}})$. It suffices to show that (26) holds for $g$.

Lemma 19 of [27] shows that for all $i \ge 1$, the following holds:

$$\frac{1}{Z_i} \int_{\mathcal{K}_i} g(x) e^{-\beta \bar{f}(x)} dx = \frac{1}{Z_i} \int_{\mathcal{K}_i} (P_t^i g)(x) e^{-\beta \bar{f}(x)} dx. \tag{27}$$

We saw earlier that $Z_i \to Z$. Furthermore, since $g$ is compactly supported, there is a number, $m$, such that $i \ge m$ implies that

$$\int_{\mathcal{K}_i} g(x) e^{-\beta \bar{f}(x)} dx = \int_{\mathcal{K}} g(x) e^{-\beta \bar{f}(x)} dx.$$

It follows that the left of (27) converges to the left of (26).

The proof will be completed if we can show that for $t > 0$,

$$\lim_{i \to \infty} \int_{\mathcal{K}_i} (P_t^i g)(x) e^{-\beta \bar{f}(x)} dx = \lim_{i \to \infty} \int_{\mathcal{K}_i} \mathbb{E}[g(\mathbf{x}_t^{C,i})|\mathbf{x}_0 = x] e^{-\beta \bar{f}(x)} dx \tag{28}$$

$$= \int_{\mathcal{K}} \mathbb{E}[g(\mathbf{x}_t^C)|\mathbf{x}_0 = x] e^{-\beta \bar{f}(x)} dx \tag{29}$$

$$= \int_{\mathcal{K}} (P_t g)(x) e^{-\beta \bar{f}(x)} dx.$$

We assumed that $g$ was bounded, and so there is a number, $b$, such that $|g(x)| \le b$ for all $x \in \mathcal{K}$. It follows from the definition of $P_t$ and $P_t^i$ that $|P_t g(x)| \le b$ and $|P_t^i g(x)| \le b$ for all $t$.

Fix any $t > 0$. The Brownian motion, $\mathbf{w}$, is continuous, and so for each $t$, $\mathbf{w}_s$ is bounded for $s \in [0, t]$. Now the form of (9) shows that $\mathbf{y}^C$, and thus $\mathbf{x}^C$ must also be continuous, and thus also bounded for $s \in [0, t]$. Thus, for each realization, we see that there is a number $m$ such that $\mathbf{x}_s^C \in \mathcal{K}_i$ for all $s \in [0, t]$ and all $i \ge m$. Thus, we see that $\mathbf{x}_s^C = \mathbf{x}_s^{C,i}$ for $s \in [0, t]$. This argument shows that the integrand on the right of (28) converges pointwise to the integrand of (29). So, the desired equality follows by the dominated convergence theorem. ∎

## C  Bounded variance of the processes

In this section, we derive variance bounds on all of the main processes, $\mathbb{E}[\|\mathbf{x}_k^A\|^2]$ and $\mathbb{E}[\|\mathbf{x}_t^C\|^2]$. The bound on $\mathbb{E}[\|\mathbf{x}_t^C\|^2]$ is used to prove bounds on the discretization error from $\mathbf{x}_t^M$ to $\mathbf{x}_t^C$. The bound on $\mathbb{E}[\|\mathbf{x}_k^A\|^2]$ is used to derive the time-uniform bounds on $W_1(\mathcal{L}(\mathbf{x}_k^A), \mathcal{L}(\mathbf{x}_k^C))$ from Lemma 3.

## C.1 Continuous-time bounds

In this section, we show that the assumption that $\bar{f}$ is strongly convex outside a ball implies that $\mathcal{V}(x) = \frac{1}{2}\|x\|^2$ can be used as a Lyapunov function for $\mathbf{x}_t^C$. In turn, we use this Lyapunov function to derive bounds on $\mathbb{E}[\|\mathbf{x}_t^C\|^2]$.

**Lemma 13.** *If $\bar{f}(x)$ is $\mu$-strongly convex outside a ball with radius $R$, then $\mathcal{V}(x) = \frac{1}{2}x^\top x$ satisfies the following the geometric drift condition:*

$$\mathcal{A}\mathcal{V}(x) \leq -2\eta\mu\mathcal{V}(x) + c_{10}\eta.$$

*Here $c_{10}$ is defined by*

$$c_{10} = (\ell + \mu)R^2 + R\|\nabla_x\bar{f}(0)\| + \frac{n}{\beta}.$$

**Proof** By Ito's formula, we have

$$
\begin{aligned}
d\mathcal{V}(\mathbf{x}_t^C) &= \nabla_x\mathcal{V}^\top d\mathbf{x}_t^C + \frac{1}{2}d(\mathbf{x}_t^C)^\top(\nabla_x^2\mathcal{V})d\mathbf{x}_t^C \\
&= (\mathbf{x}_t^C)^\top\left(-\eta\nabla_x\bar{f}(\mathbf{x}_t^C)dt + \sqrt{\frac{2\eta}{\beta}}d\mathbf{w}_t - \mathbf{v}_t d\boldsymbol{\mu}_t\right) + \frac{1}{2}d(\mathbf{x}_t^C)^\top d\mathbf{x}_t^C \\
&= -\eta(\mathbf{x}_t^C)^\top\nabla_x\bar{f}(\mathbf{x}_t^C)dt + \sqrt{\frac{2\eta}{\beta}}(\mathbf{x}_t^C)^\top d\mathbf{w}_t - (\mathbf{x}_t^C)^\top\mathbf{v}_t d\boldsymbol{\mu}_t + \frac{\eta}{\beta}\text{Tr}(d\mathbf{w}_t d\mathbf{w}_t^\top) \\
&= \left(-\eta(\mathbf{x}_t^C)^\top\nabla_x\bar{f}(\mathbf{x}_t^C) + \frac{n\eta}{\beta}\right)dt + \sqrt{\frac{2\eta}{\beta}}(\mathbf{x}_t^C)^\top d\mathbf{w}_t - (\mathbf{x}_t^C)^\top\mathbf{v}_t d\boldsymbol{\mu}_t.
\end{aligned}
$$

The third equality holds because $\int_0^t \mathbf{v}_s d\boldsymbol{\mu}_s$ has bounded variation. The last equality is based on the fact that $d\mathbf{w}_t d\mathbf{w}_t^\top = dt\, I$.

Since $\mathbf{v}_t \in N_\mathcal{K}(\mathbf{x}_t^C)$, $\boldsymbol{\mu}_t$ is a nonnegative measure, and $0 \in \mathcal{K}$, we have that $-(\mathbf{x}_t^C)^\top\mathbf{v}_t d\boldsymbol{\mu}_t \leq 0$. Thus, the generator of the Lyapunov function satisfies

$$\mathcal{A}\mathcal{V}(x) \leq -\eta x^\top\nabla_x\bar{f}(x) + \frac{n\eta}{\beta}. \tag{30}$$

If $\|x\| \geq R$, strong convexity outside a ball of radius $R$, along with the Cauchy-Schwartz inequality imply that

$$
\begin{aligned}
x^\top\nabla_x\bar{f}(x) &= (x - 0)^\top(\nabla_x\bar{f}(x) - \nabla_x\bar{f}(0)) + x^\top\nabla_x\bar{f}(0) \\
&\geq \mu\|x\|^2 - R\|\nabla_x\bar{f}(0)\| \tag{31}
\end{aligned}
$$

It follows that when $\|x\| \geq R$, we have that

$$
\begin{aligned}
\mathcal{A}\mathcal{V}(x) &\leq -\eta\mu\|x\|^2 + \eta\left(R\|\nabla_x\bar{f}(0)\| + \frac{n}{\beta}\right) \\
&= -\eta 2\mu\mathcal{V}(x) + \eta\left(R\|\nabla_x\bar{f}(0)\| + \frac{n}{\beta}\right).
\end{aligned}
$$

If $\|x\| \leq R$, then the Cauchy-Schwartz inequality and the Lipschitz continuity imply that

$$
\begin{aligned}
-x^\top\nabla_x\bar{f}(x) &= -x^\top\left(\nabla_x\bar{f}(x) - \nabla_x\bar{f}(0) + \nabla_x\bar{f}(0)\right) \\
&\leq \|x\|\|\nabla_x\bar{f}(x) - \nabla_x\bar{f}(0)\| + R\|\nabla_x\bar{f}(0)\| \\
&\leq \ell\|x\|^2 + R\|\nabla_x\bar{f}(0)\| \\
&= -\mu\|x\|^2 + (\ell + \mu)\|x\|^2 + R\|\nabla_x\bar{f}(0)\| \\
&\leq -\mu\|x\|^2 + (\ell + \mu)R^2 + R\|\nabla_x\bar{f}(0)\|. \tag{32}
\end{aligned}
$$

Note that (31) implies that (32) also holds whenever $\|x\| \geq R$. So, combining (32) with (30) shows that for all $x \in \mathcal{K}$,

$$\mathcal{A}\mathcal{V}(x) \leq \eta \left( -\mu\|x\|^2 + (\ell + \mu)R^2 + R\|\nabla_x \bar{f}(0)\| \right) + \frac{n\eta}{\beta}$$

$$= -\eta 2\mu \mathcal{V}(x) + \eta \left( (\ell + \mu)R^2 + R\|\nabla_x \bar{f}(0)\| + \frac{n}{\beta} \right)$$

∎

**Lemma 14.** *If $\mathbb{E}[\|\mathbf{x}_0^C\|^2] \leq \varsigma$, then for all $t \geq 0$, we have that*

$$\mathbb{E}[\|\mathbf{x}_t^C\|^2] \leq \varsigma + \frac{1}{\mu}c_{10},$$

*where $c_{10}$ is defined in Lemma 13.*

**Proof** Recall that Lyapunov generator $\mathcal{A}$ is defined as below

$$\mathcal{A}\mathcal{V}(x) = \lim_{t\downarrow 0} \mathbb{E}\left[ \frac{1}{t}(\mathcal{V}(\mathbf{x}_t^C) - \mathcal{V}(\mathbf{x}_0^C))|\mathbf{x}_0^C = x \right].$$

Using Dynkin's formula and Lemma 13 gives

$$\mathbb{E}\left[\mathcal{V}(\mathbf{x}_t^C) - \mathcal{V}(\mathbf{x}_0^C)\right] = \int_0^t \mathbb{E}\left[\mathcal{A}\mathcal{V}(\mathbf{x}_s^C)\right] ds$$

$$\leq -2\eta\mu \int_0^t \mathbb{E}\left[\mathcal{V}(\mathbf{x}_s^C)\right] ds + c_{10}\eta t.$$

Let $u_t = \mathbb{E}\left[\mathcal{V}(\mathbf{x}_t^C)\right]$, $u_0 = \mathbb{E}\left[\mathcal{V}(\mathbf{x}_0^C)\right]$. By Grönwall's inequality, we get

$$u_t \leq e^{-2\eta\mu t}u_0 + \eta c_{10} \int_0^t e^{-2\eta\mu s}ds$$

$$= e^{-2\eta\mu t}u_0 + \frac{c_{10}}{2\mu}\left(1 - e^{-2\eta\mu t}\right)$$

$$\leq u_0 + \frac{c_{10}}{2\mu}.$$

Recalling that $u_t = \frac{1}{2}\mathbb{E}[\|\mathbf{x}_t\|^2]$ and $\mathbb{E}[\|\mathbf{x}_0\|^2] \leq \varsigma$ completes the proof. ∎

## C.2 Discrete-time bounds

Here we derive a uniform bound on $\mathbb{E}[\|\mathbf{x}_k^A\|^2]$.

**Lemma 15.** *Assume that $\mathbb{E}[\|\mathbf{x}_0^A\|^2] \leq \varsigma$ and that $\eta \leq \min\left\{1, \frac{\mu}{4\ell^2}\right\}$. There is a constant, $c_{11}$ such that for all $k \geq 0$, we have that*

$$\mathbb{E}[\|\mathbf{x}_k^A\|^2] \leq \varsigma + c_{11}.$$

*The constant is given by*

$$c_{11} = \frac{4}{\mu}\left( \frac{n}{\beta} + (\ell + \mu)R^2 + (2 + R)\|\nabla_x \bar{f}(0)\| + \left(8\ell^2 + \frac{1}{\mu}\right)\ell^2 \mathcal{M}_2(\mathbf{z}) \right)$$

**Proof** Using non-expansiveness of the projection and then expanding the square of the norm gives:

$$\mathbb{E}\left[\|\mathbf{x}_{t+1}^A\|^2\right] = \mathbb{E}\left[ \left\| \Pi_\mathcal{K}\left( \mathbf{x}_t^A - \eta\nabla_x f(\mathbf{x}_t^A, \mathbf{z}_t) + \sqrt{\frac{2\eta}{\beta}}\hat{\mathbf{w}}_t \right) - \Pi_\mathcal{K}(0) \right\|^2 \right]$$

$$\leq \mathbb{E}\left[ \left\| \mathbf{x}_t^A - \eta\nabla_x f(\mathbf{x}_t^A, \mathbf{z}_t) + \sqrt{\frac{2\eta}{\beta}}\hat{\mathbf{w}}_t \right\|^2 \right]$$

$$= \mathbb{E}\left[ \|\mathbf{x}_t^A\|^2 + \eta^2\|\nabla_x f(\mathbf{x}_t^A, \mathbf{z}_t)\|^2 - 2\eta(\mathbf{x}_t^A)^\top \nabla_x f(\mathbf{x}_t^A, \mathbf{z}_t) \right] + \frac{2n\eta}{\beta}.$$

Now we bound the term $\mathbb{E}\left[\|\nabla_x f(\mathbf{x}_t^A, \mathbf{z}_t)\|^2\right]$. For any $x \in \mathcal{K}$, we have that

$$
\begin{aligned}
\|\nabla_x f(x, z)\|^2 &= \|\nabla_x f(x, z) - \nabla_x f(0, z) + \nabla_x f(0, z)\|^2 \\
&\leq 2\|\nabla_x f(0, z)\|^2 + 2\ell^2\|x\|^2.
\end{aligned}
\tag{33}
$$

This leads to:

$$
\begin{aligned}
\mathbb{E}\left[\|\mathbf{x}_{t+1}^A\|^2\right] \leq & \left(1 + 2\ell^2\eta^2\right)\mathbb{E}\left[\|\mathbf{x}_t^A\|^2\right] \\
& + \left(\frac{2\eta n}{\beta} + 2\eta^2\mathbb{E}[\|\nabla_x f(0, \mathbf{z}_t)\|^2]\right) - 2\eta\mathbb{E}\left[(\mathbf{x}_t^A)^\top \nabla_x f(\mathbf{x}_t^A, \mathbf{z}_t)\right].
\end{aligned}
$$

To bound the term $\mathbb{E}[\|\nabla_x f(0, \mathbf{z}_t)\|^2]$, note that $\nabla_x \bar{f}(0) = \mathbb{E}[\nabla_x f(0, \hat{\mathbf{z}}_t)]$, where $\hat{\mathbf{z}}_t$ is identically distributed to $\mathbf{z}_t$ and independent of $\mathbf{z}_t$.

$$
\begin{aligned}
\mathbb{E}[\|\nabla_x f(0, \mathbf{z}_t)\|^2] &= \mathbb{E}\left[\|\nabla_x \bar{f}(0) + \nabla_x f(0, \mathbf{z}_t) - \mathbb{E}[\nabla_x f(0, \hat{\mathbf{z}}_t)]\|^2\right] \\
&\leq 2\|\nabla_x \bar{f}(0)\|^2 + 2\mathbb{E}[\|\nabla_x f(0, \mathbf{z}_t) - \mathbb{E}[\nabla_x f(0, \hat{\mathbf{z}}_t)]\|^2] \\
&\overset{\text{Jensen}}{\leq} 2\|\nabla_x \bar{f}(0)\|^2 + 2\mathbb{E}[\|\nabla_x f(0, \mathbf{z}_t) - \nabla_x f(0, \hat{\mathbf{z}}_t)\|^2] \\
&\leq 2\|\nabla_x \bar{f}(0)\| + 2\ell^2\mathbb{E}[\|\mathbf{z}_t - \hat{\mathbf{z}}_t\|^2] \\
&\leq 2\|\nabla_x \bar{f}(0)\| + 4\ell^2\mathbb{E}[\|\mathbf{z}_t\|^2 + \|\hat{\mathbf{z}}_t\|^2] \\
&\leq 2\|\nabla_x \bar{f}(0)\| + 8\ell^2\mathcal{M}_2(\mathbf{z}),
\end{aligned}
\tag{34}
$$

where $\mathcal{M}_2(\mathbf{z})$ is a bound on $\mathbb{E}[\|\mathbf{z}_t\|^2]$ from (2).

So, we have a bound of the form

$$
\begin{aligned}
\mathbb{E}\left[\|\mathbf{x}_{t+1}^A\|^2\right] \leq & \left(1 + 2\ell^2\eta^2\right)\mathbb{E}\left[\|\mathbf{x}_t^A\|^2\right] \\
& + \left(\frac{2\eta n}{\beta} + \eta^2\left(4\|\nabla_x \bar{f}(0)\| + 16\ell^2\mathcal{M}_2(\mathbf{z})\right)\right) - 2\eta\mathbb{E}\left[(\mathbf{x}_t^A)^\top \nabla_x f(\mathbf{x}_t^A, \mathbf{z}_t)\right].
\end{aligned}
\tag{35}
$$

To bound the inner product term, note that

$$
\begin{aligned}
\mathbb{E}\left[(\mathbf{x}_t^A)^\top \nabla_x f(\mathbf{x}_t^A, \mathbf{z}_t)\right] &= \mathbb{E}\left[(\mathbf{x}_t^A)^\top \left(\nabla_x f(\mathbf{x}_t^A, \mathbf{z}_t) - \nabla_x f(\mathbf{x}_t^A, \hat{\mathbf{z}}_t)\right)\right] + \mathbb{E}\left[(\mathbf{x}_t^A)^\top \nabla_x f(\mathbf{x}_t^A, \hat{\mathbf{z}}_t)\right] \\
&= \mathbb{E}\left[(\mathbf{x}_t^A)^\top \left(\nabla_x f(\mathbf{x}_t^A, \mathbf{z}_t) - \nabla_x f(\mathbf{x}_t^A, \hat{\mathbf{z}}_t)\right)\right] + \mathbb{E}\left[(\mathbf{x}_t^A)^\top \nabla_x \bar{f}(\mathbf{x}_t^A)\right].
\end{aligned}
$$

The second equality follows because $\hat{\mathbf{z}}_t$ is independent of $\mathbf{x}_t^A$ and identically distributed to $\mathbf{z}_t$. So, we can use the Cauchy-Schwartz inequality on the first term on the right and (32) on the second term to give:

$$
\mathbb{E}\left[(\mathbf{x}_t^A)^\top \nabla_x f(\mathbf{x}_t^A, \mathbf{z}_t)\right] \geq -\ell\mathbb{E}[\|\mathbf{x}_t^A\|\|\mathbf{z}_t - \hat{\mathbf{z}}_t\|] + \mu\mathbb{E}[\|\mathbf{x}_t^A\|^2] - \left((\ell + \mu)R^2 + R\|\nabla_x \bar{f}(0)\|\right).
$$

Using a completing-the-squares argument shows that for any numbers $a$ and $b$

$$
\begin{aligned}
\frac{\mu}{2}a^2 - \ell a b &= \frac{\mu}{2}\left(a - \frac{\ell}{\mu}b\right)^2 - \frac{\ell^2}{2\mu}b^2 \\
&\geq -\frac{\ell^2}{2\mu}b^2.
\end{aligned}
$$

Setting $a = \|\mathbf{x}_t^A\|$ and $b = \|\mathbf{z}_t - \hat{\mathbf{z}}_t\|$ leads to a bound of the form

$$
\begin{aligned}
\mathbb{E}\left[(\mathbf{x}_t^A)^\top \nabla_x f(\mathbf{x}_t^A, \mathbf{z}_t)\right] &\geq \frac{\mu}{2}\mathbb{E}[\|\mathbf{x}_t^A\|^2] - \frac{\ell^2}{2\mu}\mathbb{E}[\|\mathbf{z}_t - \hat{\mathbf{z}}_t\|^2] - \left((\ell + \mu)R^2 + R\|\nabla_x \bar{f}(0)\|\right) \\
&\geq \frac{\mu}{2}\mathbb{E}[\|\mathbf{x}_t^A\|^2] - \frac{\ell^2}{\mu}\mathcal{M}_2(\mathbf{z}) - \left((\ell + \mu)R^2 + R\|\nabla_x \bar{f}(0)\|\right).
\end{aligned}
\tag{36}
$$

Plugging the new bounds into (35) gives

$$\mathbb{E}\left[\|\mathbf{x}_{t+1}^A\|^2\right] \leq \left(1 - \mu\eta + 2\ell^2\eta^2\right)\mathbb{E}\left[\|\mathbf{x}_t^A\|^2\right]$$
$$+ \left(\frac{2\eta n}{\beta} + \eta^2\left(4\|\nabla_x\bar{f}(0)\| + 16\ell^2\mathcal{M}_2(\mathbf{z})\right)\right) + 2\eta\left(\frac{\ell^2}{\mu}\mathcal{M}_2(\mathbf{z}) + \left((\ell+\mu)R^2 + R\|\nabla_x\bar{f}(0)\|\right)\right)$$

Note that if $\eta \leq \frac{\mu}{4\ell^2}$, then

$$1 - \mu\eta + 2\ell^2\eta^2 \leq 1 - \frac{\mu\eta}{2}.$$

Furthermore, if $\eta \leq 1$, we get the simplified bound:

$$\mathbb{E}\left[\|\mathbf{x}_{t+1}^A\|^2\right] \leq \left(1 - \frac{\mu\eta}{2}\right)\mathbb{E}\left[\|\mathbf{x}_t^A\|^2\right]$$
$$+ 2\eta\left(\frac{n}{\beta} + 2\|\nabla_x\bar{f}(0)\| + 8\ell^2\mathcal{M}_2(\mathbf{z}) + \frac{\ell^2}{\mu}\mathcal{M}_2(\mathbf{z}) + (\ell+\mu)R^2 + R\|\nabla_x\bar{f}(0)\|\right). \quad (37)$$

Now for any $a \in [0,1)$ and any $b \geq 0$, if $u_t \geq 0$ satisfies

$$u_{t+1} \leq au_t + b$$

then

$$u_t \leq a^t u_0 + b\sum_{k=0}^{t-1} a^k$$
$$= a^t u_0 + b\frac{1-a^t}{1-a}$$
$$\leq u_0 + \frac{b}{1-a}$$

Applying this bound to $\mathbb{E}[\|\mathbf{x}_t^A\|^2]$ and using that $\mathbb{E}[\|\mathbf{x}_0^A\|^2] \leq \varsigma$ gives

$$\mathbb{E}[\|\mathbf{x}_t^A\|^2] \leq \varsigma + \frac{4}{\mu}\left(\frac{n}{\beta} + (\ell+\mu)R^2 + (2+R)\|\nabla_x\bar{f}(0)\| + \left(8\ell^2 + \frac{1}{\mu}\right)\ell^2\mathcal{M}_2(\mathbf{z})\right).$$

∎

## D  Stochastic contraction analysis

In this Appendix, we prove Lemma 2.

### D.1  Contraction for the reflected SDEs

We extend the analysis of standard SDEs from [20] to the case of reflected SDEs. The main idea of [20] is to construct a specialized metric over $\mathbb{R}^n$ and corresponding Wasserstein distance under which contraction rates can be computed. In the context of this paper, we only use Euclidean norm to construct the metric, whereas in [20], both Euclidean and a second norm were used to construct the specilized metric. Using just one norm leads to some simplifications. Our choice of reflection term in the coupling process is also slightly different, leading to further simplifications.

In the following, we firstly examine the contractivity properties of the generalized reflected SDEs and then associate the generalized process with the original process from (1).

Let $\mathcal{K}$ be a closed convex subset of $\mathbb{R}^n$ and consider a reflected stochastic differential equations of the form:

$$d\mathbf{x}_t = H(\mathbf{x}_t)dt + Gd\mathbf{w}_t - \mathbf{v}_t d\boldsymbol{\mu}(t), \quad (38)$$

where $G$ is an invertible $n \times n$ matrix with minimum singular value $\sigma_{\min}(G)$, $\mathbf{w}_t$ is a standard Brownian motion, and $-\int_0^t \mathbf{v}_s d\boldsymbol{\mu}(s)$ is a reflection term that ensures that $\mathbf{x}_t \in \mathcal{K}$ for all $t \geq 0$. (We

are slightly abusing notation, since here $\mathbf{x}_t$ denotes the solution to a general RSDE, and is not the iterates of the original algorithm from (1).)

Following [20], we construct a function $\delta : [0, +\infty) \to \mathbb{R}$ such that $\delta(0) = 0$, $\delta'(0) = 1$, $\delta'(r) > 0$, and $\delta''(r) \leq 0$ for all $r \geq 0$. With these properties, it can be shown that $\delta(\|x - y\|)$ forms a metric over $\mathcal{K}$. The particular metric is constructed so that the dynamics are contractive with respect to the corresponding Wasserstein distance.

Assume there exists a continuous function $\kappa(r) : [0, +\infty) \to \mathbb{R}$ such that for any $x, y \in \mathbb{R}^n, x \neq y$,

$$(x - y)^\top (H(x) - H(y)) \leq \kappa(\|x - y\|)\|x - y\|^2. \tag{39}$$

Also, assume that

$$\limsup \kappa(r) < 0. \tag{40}$$

This implies that there is a positive constant, $R_0$, and a negative constant $\bar{\kappa}$, such that $\kappa(r) \leq \bar{\kappa} < 0$ for all $r > R_0$.

We choose

$$R_1 = \frac{R_0}{2} + \frac{1}{2}\sqrt{R_0^2 - \frac{16\sigma_{min}(G)^2 e^{h(R_0)}}{\bar{\kappa}}} > R_0,$$

and define $\delta$ via the following chain of definitions:

$$\delta(r) = \int_0^r \varphi(s)g(s)ds \tag{41a}$$

$$g(r) = 1 - \frac{\xi}{2}\int_0^{r \wedge R_1} \Phi(s)\varphi(s)^{-1}ds \tag{41b}$$

$$\xi^{-1} = \int_0^{R_1} \Phi(s)\varphi(s)^{-1}ds \tag{41c}$$

$$\Phi(r) = \int_0^r \varphi(s)ds \tag{41d}$$

$$\varphi(r) = e^{-h(r)} \tag{41e}$$

$$h(r) = \frac{1}{2\sigma_{\min}(G)^2}\int_0^r s(\kappa(s) \vee 0)ds. \tag{41f}$$

In the above definition, we use the shorthand notation $a \wedge b = \min\{a, b\}$ and $a \vee b = \max\{a, b\}$.

The details on the choices of $R_0$ and $R_1$ will be presented during the proof of Theorem 16 for the general reflection coupling related to (38) and Corollary 17 for the specific reflection coupling related to (8).

As discussed above, $\delta(\|x - y\|)$ is a metric. See [20] for details. The corresponding Wasserstein distance is defined by

$$W_\delta(P, Q) = \inf_{\Gamma \in \mathfrak{C}(P,Q)} \int_{\mathcal{K} \times \mathcal{K}} \delta(\|x - y\|)d\Gamma(x, y)$$

Here, $\mathfrak{C}$ is the couplings between $P$ and $Q$.

To get an explicit form of the constant factor in Lemma 2, we use the following theorem, which is analogous to Corollary 2 of [20].

**Theorem 16.** *If $\mathbf{x}_t^1$ and $\mathbf{x}_t^2$ are two solutions to (38), then for all $0 \leq s \leq t$, their laws satisfy*

$$W_\delta(\mathcal{L}(\mathbf{x}_t^1), \mathcal{L}(\mathbf{x}_t^2)) \leq e^{-\tilde{a}(t-s)}W_\delta(\mathcal{L}(\mathbf{x}_s^1), \mathcal{L}(\mathbf{x}_s^2))$$

*where $\tilde{a} = \xi\sigma_{\min}(G)^2$.*

**Proof** The proof closely follows the proof of Theorem 1 from [20] with constraints handled similar to works in [27, 29]. The key is to create an explicit coupling between $\mathbf{x}_t^1$ and $\mathbf{x}_t^2$, which is known as a *reflection coupling* [31].

To define the reflection coupling, let $\tau$ be coupling time: $\tau = \inf\left\{t | \mathbf{x}_t^1 = \mathbf{x}_t^2\right\}$. Let $\mathbf{r}_t = \|\mathbf{x}_t^1 - \mathbf{x}_t^2\|$, $\mathbf{u}_t = (\mathbf{x}_t^1 - \mathbf{x}_t^2)/\mathbf{r}_t$. Then the reflection coupling between $\mathbf{x}_t^1$ and $\mathbf{x}_t^2$ is defined by:

$$dx_t^1 = H(\mathbf{x}_t^1)dt + Gd\mathbf{w}_t - \mathbf{v}_t^1 d\boldsymbol{\mu}^1(t) \tag{42a}$$

$$dx_t^2 = H(\mathbf{x}_t^2)dt + (I - 2\mathbf{u}_t\mathbf{u}_t^\top \mathbb{1}(t < \boldsymbol{\tau}))Gd\mathbf{w}_t - \mathbf{v}_t^2 d\boldsymbol{\mu}^2(t) \tag{42b}$$

where $-\int_0^t \mathbf{v}_s^1 d\boldsymbol{\mu}^1(s)$ and $-\int_0^t \mathbf{v}_s^2 d\boldsymbol{\mu}^2(s)$ are reflection terms that ensure that $\mathbf{x}_t^1 \in \mathcal{K}$ and $\mathbf{x}_t^2 \in \mathcal{K}$ for all $t \geq 0$.

The processes from (42) define a valid coupling since $\int_0^T (I - 2\mathbf{u}_t\mathbf{u}_t^\top \mathbb{1}(t < \boldsymbol{\tau}))Gd\mathbf{w}_t$ is a Brownian motion by Lévy's characterization.

The main idea is to show that with the specially constructed metric (41), there will be a constant $\tilde{a}$ such that $e^{\tilde{a}t}\delta(\mathbf{r}_t)$ is a supermartingale. Then, the definition of $W_\delta$ and the supermartingale property shows that

$$W_\delta(\mathcal{L}(\mathbf{x}_t^1), \mathcal{L}(\mathbf{x}_t^2)) \leq \mathbb{E}\left[\delta(\mathbf{r}_t)\right] \leq e^{-\tilde{a}(t-s)}\mathbb{E}[\delta(\mathbf{r}_s)]$$

Since this bound holds for all couplings of the laws $\mathcal{L}(\mathbf{x}_s^1)$ and $\mathcal{L}(\mathbf{x}_s^2)$, it must hold for the optimal coupling, and so

$$W_\delta(\mathcal{L}(\mathbf{x}_t^1), \mathcal{L}(\mathbf{x}_t^2)) \leq e^{-\tilde{a}(t-s)}W_\delta(\mathcal{L}(\mathbf{x}_s^1), \mathcal{L}(\mathbf{x}_s^2)),$$

which is the desired conclusion.

Therefore, to complete the proof, we must show that $e^{\tilde{a}t}\delta(\mathbf{r}_t)$ is a supermartingale, which is to ensure that this process is non-increasing on average. Recall that $\boldsymbol{\tau}$ is the coupling time, so that $e^{\tilde{a}t}\delta(\mathbf{r}_t) = 0$ for $t \geq \boldsymbol{\tau}$. So we want to bound the behavior of the process for all $t < \boldsymbol{\tau}$. Specifically, it is required to show that non-martingale terms of $d\left(e^{\tilde{a}t}\delta(\mathbf{r}_t)\right)$ are non-positive. By Itô's formula, we have that

$$d\left(e^{\tilde{a}t}\delta(\mathbf{r}_t)\right) = e^{\tilde{a}t}\left(\tilde{a}\delta(r)dt + \delta'(r)d\mathbf{r}_t + \frac{1}{2}\delta''(r)(d\mathbf{r}_t)^2\right).$$

To achieve the desired differential, we have to derive the terms $d\mathbf{r}_t$ and $(d\mathbf{r}_t)^2$.

$$\begin{aligned}
d\mathbf{r}_t &= \mathbf{u}_t^\top\left(d\mathbf{x}_t^1 - d\mathbf{x}_t^2\right)\\
&= \mathbf{u}_t^\top\left(\left(H(\mathbf{x}_t^1) - H(\mathbf{x}_t^2)\right)dt + 2\mathbf{u}_t\mathbf{u}_t^\top Gd\mathbf{w}_t - \mathbf{v}_t^1 d\boldsymbol{\mu}^1(t) + \mathbf{v}_t^2 d\boldsymbol{\mu}^2(t)\right)
\end{aligned}$$

The above equation is simplified because $(d\mathbf{x}_t^1 - d\mathbf{x}_t^2)^\top(\nabla^2\mathbf{r}_t)(d\mathbf{x}_t^1 - d\mathbf{x}_t^2) = 0$.

Also, by assumption we have

$$(\mathbf{x}_t^1 - \mathbf{x}_t^2)^\top\left(H(\mathbf{x}_t^1) - H(\mathbf{x}_t^2)\right) \leq \kappa(\|\mathbf{x}_t^1 - \mathbf{x}_t^2\|)\|\mathbf{x}_t^1 - \mathbf{x}_t^2\|^2. \tag{43}$$

By the definition of $\mathbf{u}_t$ and the facts that $\mathbf{v}_t^1 \in N_\mathcal{K}(\mathbf{x}_t^1)$ and $\mathbf{v}_t^2 \in N_\mathcal{K}(\mathbf{x}_t^2)$ imply that $-\mathbf{u}_t^\top\mathbf{v}_t^1 d\boldsymbol{\mu}^1(t) \leq 0$ and $\mathbf{u}_t^\top\mathbf{v}_t^2 d\boldsymbol{\mu}^2(t) \leq 0$. It follows that and the assumption (39) gives

$$d\mathbf{r}_t \leq \kappa(r)rdt + 2\mathbf{u}_t^\top Gd\mathbf{w}_t.$$

Now, since the terms that were dropped in the inequality have bounded variation, we have that

$$(d\mathbf{r}_t)^2 = 4\mathbf{u}_t GG^\top\mathbf{u}_t dt \geq 4\sigma_{\min}(G)^2 dt.$$

By construction $\delta'(r) \geq 0$ and $\delta''(r) \leq 0$, and so Itô's formula gives

$$\begin{aligned}
d\left(e^{\tilde{a}t}\delta(\mathbf{r}_t)\right) &\leq dt e^{\tilde{a}t}\left(\tilde{a}\delta(r) + \delta'(r)\kappa(r)r + \delta''(r)2\sigma_{\min}(G)^2\right) + \mathbf{m}_t\\
&= 2\sigma_{\min}(G)^2 e^{\tilde{a}t}dt\left(\frac{\tilde{a}}{2\sigma_{\min}(G)^2}\delta(r) + \frac{k(r)r}{2\sigma_{\min}(G)^2}\delta'(r) + \delta''(r)\right) + \mathbf{m}_t,
\end{aligned}$$

where $\mathbf{m}_t$ denotes a local martingale.

So it suffices to pick certain $\tilde{a}$ and $R_1$ to ensure that for all $r \geq 0$, the following holds:

$$\frac{\tilde{a}}{2\sigma_{\min}(G)^2}\delta(r) + \frac{\kappa(r)r}{2\sigma_{\min}(G)^2}\delta'(r) + \delta''(r) \leq 0. \tag{44}$$

Recall that

$$\delta''(r) = \varphi'(r)g(r) + g'(r)\varphi(r)$$

$$= -\frac{1}{2\sigma_{\min}(G)^2}r(\kappa(r) \vee 0)\delta'(r) - \frac{\xi}{2}\Phi(r)\mathbb{1}(r < R_1).$$

So if we set $\tilde{a} = \xi\sigma_{\min}(G)^2$, then $\delta(r) \leq \Phi(r)$ implies that (44) holds for all $r < R_1$.

The remaining work is to find a sufficient condition under which (44) holds when $r \geq R_1$.

Recall that we assume that there exists $0 < R_0$, such that $k(r) < 0$ for all $r \geq R_0$. So, if we choose $R_1 > R_0$, we have for all $r \geq R_1$ that $\varphi(r) = \varphi(R_0)$. By definition, $g(r) = \frac{1}{2}$ for all $r \geq R_1$, and so we must also have $\delta'(r) = \frac{1}{2}\varphi(R_0)$.

Therefore, for $r \geq R_1$, (44) becomes

$$\frac{\tilde{a}}{2\sigma_{\min}(G)^2}\delta(r) + \frac{\kappa(r)r}{2\sigma_{\min}(G)^2}\frac{1}{2}\varphi(R_0) \leq 0.$$

So, a sufficient condition for (44) to hold when $r \geq R_1$ is given by:

$$\frac{\tilde{a}\delta(r)}{2\sigma_{\min}(G)^2} + \frac{\kappa(r)r}{2\sigma_{\min}(G)^2}\frac{1}{2}\varphi(R_0) \leq 0 \tag{46a}$$

$$\iff \tilde{a}\delta(r) + \kappa(r)r\frac{1}{2}\varphi(R_0) \leq 0 \tag{46b}$$

$$\iff \kappa(r)r\frac{1}{2}\varphi(R_0) \leq -\tilde{a}\delta(r) \tag{46c}$$

$$\iff \kappa(r)r\frac{1}{2}\varphi(R_0) \leq -\xi\sigma_{min}(G)^2\delta(r) \tag{46d}$$

$$\iff \kappa(r)r\frac{1}{2}\varphi(R_0) \leq -\frac{\sigma_{min}(G)^2}{\int_0^{R_1}\Phi(s)\varphi(s)^{-1}ds}\delta(r) \tag{46e}$$

$$\Longleftarrow \kappa(r)r\frac{1}{2}\varphi(R_0) \leq -\frac{\sigma_{min}(G)^2}{(R_1 - R_0)\Phi(R_1)\varphi(R_0)^{-1}/2}\delta(r) \tag{46f}$$

$$\Longleftarrow \kappa(r)r\frac{1}{2}\varphi(R_0) \leq -\frac{\sigma_{min}(G)^2}{(R_1 - R_0)\Phi(R_1)\varphi(R_0)^{-1}/2}r \tag{46g}$$

$$\iff \kappa(r) \leq -\frac{4\sigma_{min}(G)^2}{(R_1 - R_0)\Phi(R_1)} \tag{46h}$$

$$\iff (R_1 - R_0)\Phi(R_1) \geq -\frac{4\sigma_{min}(G)^2}{\kappa(r)} \tag{46i}$$

$$\Longleftarrow (R_1 - R_0)R_1e^{-h(R_0)} \geq -\frac{4\sigma_{min}(G)^2}{\kappa(r)} \tag{46j}$$

$$\Longleftarrow (R_1 - R_0)R_1e^{-h(R_0)} \geq -\frac{4\sigma_{min}(G)^2}{\bar{\kappa}} \tag{46k}$$

Note (46e) is implied by (46f) because for $r > R_0$, $\varphi(r) = \varphi(R_0)$, therefore, $\Phi(r) = \Phi(R_0) + \varphi(R_0)(r - R_0)$ which gives

$$\int_0^{R_1}\Phi(s)\varphi(s)^{-1}ds \geq \int_{R_0}^{R_1}\Phi(s)\varphi(s)^{-1}ds$$

$$= \int_{R_0}^{R_1}\left(\Phi(R_0) + \varphi(R_0)(s - R_0)\right)\varphi(R_0)^{-1}ds$$

$$= \Phi(R_0)\varphi(R_0)^{-1}(R_1 - R_0) + \frac{(R_1 - R_0)^2}{2}$$

$$\geq \frac{\Phi(R_0)\varphi(R_0)^{-1}(R_1 - R_0)}{2} + \frac{(R_1 - R_0)^2}{2}$$

$$= (R_1 - R_0)\left(\Phi(R_0) + (R_1 - R_0)\varphi(R_0)\right)\varphi(R_0)^{-1}/2$$

$$= (R_1 - R_0)\Phi(R_1)\varphi(R_0)^{-1}/2. \tag{47}$$

Also, (46f) is implied by (46g) because $\delta(r) < r$.

From (46i) to (46j), we use:

$$
\begin{aligned}
\Phi(R_1) &= \int_0^{R_1} \varphi(s)ds \\
&= \int_0^{R_1} e^{-h(s)}ds \\
&\geq \int_0^{R_1} e^{-h(R_0)}ds \\
&= R_1 e^{-h(R_0)}.
\end{aligned}
\tag{48}
$$

The implication (46k) $\implies$ (46j) arises because of the assumption that $\kappa(r) \leq \bar{\kappa} < 0$ for all $r > R_0$. Therefore, (44) will hold all $r \geq R_1$, as long as $R_1$ satisfies (46k). The smallest such $R_1$ is given by

$$
R_1 = \frac{R_0}{2} + \frac{1}{2}\sqrt{R_0^2 - \frac{16\sigma_{min}(G)^2 e^{h(R_0)}}{\bar{\kappa}}} > R_0.
\tag{49}
$$

$\blacksquare$

We choose our reflection term as $(I - 2\mathbf{u}_t\mathbf{u}_t^\top \mathbb{1}(t < \boldsymbol{\tau}))Gd\mathbf{w}_t$, while [20] uses $G(I - 2\mathbf{e}_t\mathbf{e}_t^\top \mathbb{1}(t < \boldsymbol{\tau}))d\mathbf{w}_t$, with $\mathbf{e}_t = \frac{G^{-1}(\mathbf{x}_t^1 - \mathbf{x}_t^2)}{\|G^{-1}(\mathbf{x}_t^1 - \mathbf{x}_t^2)\|}$. Our form of the reflection term leads to mild simplification of some formulas.

Now we specialize the result from the previous theorem to the specific case of this paper:

**Corollary 17.** *If $\mathbf{x}_t^1$ and $\mathbf{x}_t^2$ are two solutions to (8), then for all $0 \leq s \leq t$, their laws satisfy*

$$
W_\delta(\mathcal{L}(\mathbf{x}_t^1), \mathcal{L}(\mathbf{x}_t^2)) \leq e^{-\tilde{a}(t-s)} W_\delta(\mathcal{L}(\mathbf{x}_s^1), \mathcal{L}(\mathbf{x}_s^2))
$$

*where $\tilde{a} = \xi\frac{2\eta}{\beta}$, $R_0 = R$, and $R_1 = \frac{R}{2} + \frac{1}{2}\sqrt{R^2 + \frac{32}{\mu\beta}e^{\frac{\beta\ell R^2}{8}}}$ in the construction of $\delta$.*

**Proof** We can see that (8) is a special case of (38) with

$$
\begin{aligned}
H(x) &= -\eta\nabla_x \bar{f}(x) \\
G &= \sqrt{\frac{2\eta}{\beta}}I.
\end{aligned}
$$

Since we assume that $\bar{f}$ is $\ell$-Lipschitz and convex outside a ball with radius $R$, we have that (39) holds with $\kappa(s) = \eta\ell$ for $0 \leq s < R$ and $\kappa(s) = -\eta\mu$ for $s \geq R$. Therefore, we can pick $R_0 = R$ to construct the metric (41).

Now, $\sigma_{\min}(G)^2 = \frac{2\eta}{\beta}$ implies that $\tilde{a} = \xi\frac{2\eta}{\beta}$. Furthermore, the choice of $\kappa(r)$ implies that $h(R_0) = h(R) = \frac{\beta\ell R^2}{8}$.

The choice of $\kappa(r)$ also implies that $\bar{\kappa} = -\eta\mu$. Thus, the form of $R_1$ is given by plugging terms into (49). $\blacksquare$

**Corollary 18.** *If $\mathbf{x}_t^1$ and $\mathbf{x}_t^2$ are two solutions to (8), then for all $0 \leq s \leq t$, their laws satisfy*

$$
W_1(\mathcal{L}(\mathbf{x}_t^1), \mathcal{L}(\mathbf{x}_t^2)) \leq 2\varphi(R)^{-1}e^{-\tilde{a}(t-s)} W_1(\mathcal{L}(\mathbf{x}_s^1), \mathcal{L}(\mathbf{x}_s^2)).
$$

**Proof** From the special constructed of $\delta$, we that $\delta'(r)$ is monotonically decreasing, and also $\delta'(r) = \delta(R_1)$ for all $r \geq R_1$. Furthermore, $\delta(r) = \int_0^r \delta'(s)ds \geq \delta'(r)\int_0^r ds = r\delta'(r)$. Thus, for all $r \geq 0$, the following bounds hold:

$$
\delta'(R_1)r \leq \delta'(r)r \leq \delta(r) \leq r
$$

These bounds are now used to relate the $W_\delta$ and $W_1$ distances:

$$
\delta'(R_1)W_1(\mathcal{L}(\mathbf{x}_t^1), \mathcal{L}(\mathbf{x}_t^2)) \leq W_\delta(\mathcal{L}(\mathbf{x}_t^1), \mathcal{L}(\mathbf{x}_t^2)) \leq W_1(\mathcal{L}(\mathbf{x}_t), \mathcal{L}(\mathbf{y}_t)).
\tag{50}
$$

In particular,

$$\delta'(R_1) = \varphi(R_1)g(R_1) = \frac{1}{2}\varphi(R). \tag{51}$$

Plugging (51) into the first inequality of (50) gives

$$W_1(\mathcal{L}(\mathbf{x}_t^1), \mathcal{L}(\mathbf{x}_t^2)) \le 2\varphi(R)^{-1}W_\delta(\mathcal{L}(\mathbf{x}_t^1), \mathcal{L}(\mathbf{x}_t^2)) \tag{52}$$

And combining with Corollary 17 gives

$$W_1(\mathcal{L}(\mathbf{x}_t^1), \mathcal{L}(\mathbf{x}_t^2)) \le 2\varphi(R)^{-1}e^{-\tilde{a}(t-s)}W_\delta(\mathcal{L}(\mathbf{x}_s^1), \mathcal{L}(\mathbf{x}_s^2)) \tag{53}$$

Finally, utilizing the second inequality of (50) gives the desired result. ∎

### D.2 Proof of Lemma 2

In Lemma 12 of Appendix B, we showed that the Gibbs distribution, $\pi_{\beta\bar{f}}$, defined in (6) is invariant for the dynamics of $\mathbf{x}_t^C$. Thus, setting $\mathcal{L}(\mathbf{x}_t^1) = \mathcal{L}(\mathbf{x}_t^C)$ and $\mathcal{L}(\mathbf{x}_t^2) = \pi_{\beta\bar{f}}$ in Corollary 18 gives

$$W_1(\mathcal{L}(\mathbf{x}_t^C), \pi_{\beta\bar{f}}) \le 2\varphi(R)^{-1}e^{-\tilde{a}t}W_1(\mathcal{L}(\mathbf{x}_0^C), \pi_{\beta\bar{f}}). \tag{54}$$

Let $\mathbf{y}$ be distributed according to $\pi_{\beta\bar{f}}$. For any joint distribution over $(\mathbf{x}_0^C, \mathbf{y})$ whose marginals are $\mathcal{L}(\mathbf{x}_0^C)$ and $\pi_{\beta\bar{f}}$, we have that

$$
\begin{aligned}
W_1(\mathcal{L}(\mathbf{x}_0^C), \pi_{\beta\bar{f}}) &\le \mathbb{E}[\|\mathbf{x}_0^C - \mathbf{y}\|] \\
&\le \sqrt{\mathbb{E}[\|\mathbf{x}_0^C - \mathbf{y}\|^2]} \\
&\le \sqrt{\mathbb{E}[2\|\mathbf{x}_0^C\|^2 + 2\|\mathbf{y}\|^2]} \\
&= \sqrt{2\mathbb{E}[\|\mathbf{x}_0^C\|^2] + 2\mathbb{E}[\|\mathbf{y}\|^2]} \\
&\le \sqrt{2\varsigma + 2\left(\varsigma + \frac{1}{\mu}c_{10}\right)} \\
&\le \sqrt{\frac{2}{\mu}c_{10}} + 2\sqrt{\varsigma}.
\end{aligned}
\tag{55}
$$

The second to last inequality uses Lemma 14.

Combining (54), (55) shows that

$$W_1(\mathcal{L}(\mathbf{x}_t^C), \pi_{\beta\bar{f}}) \le 2\varphi(R)^{-1}e^{-\tilde{a}t}\left(\sqrt{\frac{2}{\mu}c_{10}} + 2\sqrt{\varsigma}\right).$$

Thus, the lemma is proved and the constants are given by:

$$a = \frac{2\xi}{\beta} \tag{56a}$$

$$c_1 = 2\varphi(R)^{-1}\sqrt{\frac{2}{\mu}c_{10}} \tag{56b}$$

$$c_2 = 4\varphi(R)^{-1} \tag{56c}$$

where $\xi$ is given in (41c). ∎

# E Proofs of averaging lemmas

**Proof of Lemma 4** Non-expansiveness of the projection and the definitions of $\mathbf{x}_t^{M,s}$ and $\mathbf{x}_t^{B,s}$ show that:

$$\|\mathbf{x}_{t+1}^{M,s} - \mathbf{x}_{t+1}^{B,s}\|^2$$

$$\leq \left\| \mathbf{x}_t^{M,s} - \mathbf{x}_t^{B,s} + \eta \left( \mathbb{E}[\nabla_x f(\mathbf{x}_t^{M,s}, \mathbf{z}_t) | \mathcal{F}_{t-s-1} \vee \mathcal{G}_t] - \mathbb{E}[\nabla_x f(\mathbf{x}_t^{M,s}, \mathbf{z}_t) | \mathcal{F}_{t-s} \vee \mathcal{G}_t] \right) \right\|^2$$

$$= \|\mathbf{x}_t^{M,s} - \mathbf{x}_t^{B,s}\|^2 + 2\eta \left( \mathbf{x}_t^{M,s} - \mathbf{x}_t^{B,s} \right)^\top$$

$$\left( \mathbb{E}[\nabla_x f(\mathbf{x}_t^{M,s}, \mathbf{z}_t) | \mathcal{F}_{t-s-1} \vee \mathcal{G}_t] - \mathbb{E}[\nabla_x f(\mathbf{x}_t^{M,s}, \mathbf{z}_t) | \mathcal{F}_{t-s} \vee \mathcal{G}_t] \right)$$

$$+ \eta^2 \left\| \mathbb{E}[\nabla_x f(\mathbf{x}_t^{M,s}, \mathbf{z}_t) | \mathcal{F}_{t-s-1} \vee \mathcal{G}_t] - \mathbb{E}[\nabla_x f(\mathbf{x}_t^{M,s}, \mathbf{z}_t) | \mathcal{F}_{t-s} \vee \mathcal{G}_t] \right\|^2. \tag{57}$$

We will show that the second term on the right of (57) has mean zero, and then we will bound the mean of the third term on the right of (57).

By construction, we have that $\mathbf{x}_t^{M,s}$ is $\mathcal{F}_{t-s-1} \vee \mathcal{G}_t$-measurable, while $\mathbf{x}_t^{B,s}$ is $\mathcal{F}_{t-s-2} \vee \mathcal{G}_t$-measurable. Thus, the only part of the second term on the right of (57) which is not $\mathcal{F}_{t-s-1} \vee \mathcal{G}_t$-measurable is $\mathbb{E}[\nabla_x f(\mathbf{x}_t^{M,s}, \mathbf{z}_t) | \mathcal{F}_{t-s} \vee \mathcal{G}_t]$. Therefore, the tower-property gives:

$$\mathbb{E}\left[ \left( \mathbf{x}_t^{M,s} - \mathbf{x}_t^{B,s} \right)^\top \left( \mathbb{E}[\nabla_x f(\mathbf{x}_t^{M,s}, \mathbf{z}_t) | \mathcal{F}_{t-s-1} \vee \mathcal{G}_t] - \mathbb{E}[\nabla_x f(\mathbf{x}_t^{M,s}, \mathbf{z}_t) | \mathcal{F}_{t-s} \vee \mathcal{G}_t] \right) \right]$$

$$= \mathbb{E}\left[ \left( \mathbf{x}_t^{M,s} - \mathbf{x}_t^{B,s} \right)^\top \left( \mathbb{E}[\nabla_x f(\mathbf{x}_t^{M,s}, \mathbf{z}_t) | \mathcal{F}_{t-s-1} \vee \mathcal{G}_t] \right.\right.$$

$$\left.\left. - \mathbb{E}\left[ \mathbb{E}[\nabla_x f(\mathbf{x}_t^{M,s}, \mathbf{z}_t) | \mathcal{F}_{t-s} \vee \mathcal{G}_t] \right) \middle| \mathcal{F}_{t-s-1} \vee \mathcal{G}_t \right] \right]$$

$$= 0.$$

Now we focus on bounding the mean of the third term on the right of (57). Recall that $\mathbf{x}_t^{M,s}$ is $\mathcal{F}_{t-s-1} \vee \mathcal{G}_t$-measurable. Furthermore, since $\mathcal{F}_{t-s}^+$ is independent of $\mathcal{F}_{t-s} \vee \mathcal{G}_t$, it must also be independent of $\mathcal{F}_{t-s-1} \vee \mathcal{G}_t$ because $\mathcal{F}_{t-s-1} \subset \mathcal{F}_{t-s}$. It follows that

$$\mathbb{E}[\nabla_x f(\mathbf{x}_t^{M,s}, \mathbb{E}[\mathbf{z}_t | \mathcal{F}_{t-s}^+]) | \mathcal{F}_{t-s} \vee \mathcal{G}_t] = \mathbb{E}[\nabla_x f(\mathbf{x}_t^{M,s}, \mathbb{E}[\mathbf{z}_t | \mathcal{F}_{t-s}^+]) | \mathcal{F}_{t-s-1} \vee \mathcal{G}_t].$$

Thus, adding and subtracting $\mathbb{E}\left[ \nabla_x f(\mathbf{x}_t^{M,s}, \mathbb{E}[\mathbf{z}_t | \mathcal{F}_{t-s}^+]) \middle| \mathcal{F}_{t-s} \vee \mathcal{G}_t \right]$ gives

$$\left\| \mathbb{E}[\nabla_x f(\mathbf{x}_t^{M,s}, \mathbf{z}_t) | \mathcal{F}_{t-s-1} \vee \mathcal{G}_t] - \mathbb{E}[\nabla_x f(\mathbf{x}_t^{M,s}, \mathbf{z}_t) | \mathcal{F}_{t-s} \vee \mathcal{G}_t] \right\|^2$$

$$\leq 2 \left\| \mathbb{E}\left[ \nabla_x f(\mathbf{x}_t^{M,s}, \mathbf{z}_t) - \nabla_x f(\mathbf{x}_t^{M,s}, \mathbb{E}[\mathbf{z}_t | \mathcal{F}_{t-s}^+]) \middle| \mathcal{F}_{t-s-1} \vee \mathcal{G}_t \right] \right\|^2$$

$$+ 2 \left\| \mathbb{E}\left[ \nabla_x f(\mathbf{x}_t^{M,s}, \mathbf{z}_t) - \nabla_x f(\mathbf{x}_t^{M,s}, \mathbb{E}[\mathbf{z}_t | \mathcal{F}_{t-s}^+]) \middle| \mathcal{F}_{t-s} \vee \mathcal{G}_t \right] \right\|^2. \tag{58}$$

To bound the second term on the right of (58), we have

$$\mathbb{E}\left[ \left\| \mathbb{E}\left[ \nabla_x f(\mathbf{x}_t^{M,s}, \mathbf{z}_t) - \nabla_x f(\mathbf{x}_t^{M,s}, \mathbb{E}[\mathbf{z}_t | \mathcal{F}_{t-s}^+]) \middle| \mathcal{F}_{t-s} \vee \mathcal{G}_t \right] \right\|^2 \right]$$

$$\overset{\text{Jensen}}{\leq} \mathbb{E}\left[ \left\| \nabla_x f(\mathbf{x}_t^{M,s}, \mathbf{z}_t) - \nabla_x f(\mathbf{x}_t^{M,s}, \mathbb{E}[\mathbf{z}_t | \mathcal{F}_{t-s}^+]) \right\|^2 \right]$$

$$\overset{\text{Lipschitz}}{\leq} \ell^2 \mathbb{E}\left[ \|\mathbf{z}_t - \mathbb{E}[\mathbf{z}_t | \mathcal{F}_{t-s}^+]\|^2 \right]$$

$$\leq \ell^2 \psi_2(s, \mathbf{z})^2.$$

Here $\psi_2(s, \mathbf{z})$ was defined in (3b).

The first term on the right of (58) is bounded by analogous calculations with $\mathcal{F}_{t-s-1}$ used in place of $\mathcal{F}_{t-s}$, and gives rise to the same bound of $\ell^2 \psi(s, \mathbf{z})^2$.

Plugging these bounds into (57) shows that

$$\mathbb{E}\left[\|\mathbf{x}_{t+1}^{M,s} - \mathbf{x}_{t+1}^{B,s}\|^2\right] \leq \mathbb{E}\left[\|\mathbf{x}_t^{M,s} - \mathbf{x}_t^{B,s}\|^2\right] + 4\eta^2 \ell^2 \psi_2(s, \mathbf{z})^2$$

Iterating (E) $t$ times and using the fact that $\mathbf{x}_0^{B,s} = \mathbf{x}_0^{M,s}$, shows that

$$\mathbb{E}\left[\|\mathbf{x}_t^{M,s} - \mathbf{x}_t^{B,s}\|^2\right] \leq 4\eta^2 t \ell^2 \psi_2(s, \mathbf{z})^2.$$

Using the fact that

$$\mathbb{E}[\|\mathbf{x}_t^{M,s} - \mathbf{x}_t^{B,s}\|] \leq \sqrt{\mathbb{E}\left[\|\mathbf{x}_t^{M,s} - \mathbf{x}_t^{B,s}\|^2\right]}$$

gives the result. ∎

**Proof of Lemma 5**  Non-expansiveness of the projection and the definitions of $\mathbf{x}_t^{B,s}$ and $\mathbf{x}_t^{M,s+1}$, shows that

$$\|\mathbf{x}_{t+1}^{B,s} - \mathbf{x}_{t+1}^{M,s+1}\|$$
$$\leq \left\|\mathbf{x}_t^{B,s} - \mathbf{x}_t^{M,s+1} + \eta\left(\mathbb{E}[\nabla_x f(\mathbf{x}_t^{M,s+1}, \mathbf{z}_t) | \mathcal{F}_{t-s-1} \vee \mathcal{G}_t] - \mathbb{E}[\nabla_x f(\mathbf{x}_t^{M,s}, \mathbf{z}_t) | \mathcal{F}_{t-s-1} \vee \mathcal{G}_t]\right)\right\|.$$

Let $\|\mathbf{x}\|_2 = \sqrt{\mathbb{E}[\|\mathbf{x}\|^2]}$ denote the 2-norm over random vectors. The triangle inequality then implies that

$$\left\|\mathbf{x}_t^{B,s} - \mathbf{x}_t^{M,s+1} + \eta\left(\mathbb{E}[\nabla_x f(\mathbf{x}_t^{M,s+1}, \mathbf{z}_t) | \mathcal{F}_{t-s-1} \vee \mathcal{G}_t] - \mathbb{E}[\nabla_x f(\mathbf{x}_t^{M,s}, \mathbf{z}_t) | \mathcal{F}_{t-s-1} \vee \mathcal{G}_t]\right)\right\|_2$$
$$\leq \left\|\mathbf{x}_t^{B,s} - \mathbf{x}_t^{M,s+1}\right\|_2 + \eta\left\|\mathbb{E}[\nabla_x f(\mathbf{x}_t^{M,s+1}, \mathbf{z}_t) - \nabla_x f(\mathbf{x}_t^{M,s}, \mathbf{z}_t) | \mathcal{F}_{t-s-1} \vee \mathcal{G}_t]\right\|_2.$$

For any random vector, $\mathbf{x}$, and any $\sigma$-algebra, $\mathcal{F}$, Jensen's inequality followed by the tower property implies that $\mathbb{E}[\|\mathbb{E}[\mathbf{x}|\mathcal{F}]\|^2] \leq \mathbb{E}[\|\mathbf{x}\|^2]$. Applying this fact to the second term on the right of (59) and then using the Lipschitz property shows that

$$\left\|\mathbb{E}[\nabla_x f(\mathbf{x}_t^{M,s+1}, \mathbf{z}_t) - \nabla_x f(\mathbf{x}_t^{M,s}, \mathbf{z}_t) | \mathcal{F}_{t-s-1} \vee \mathcal{G}_t]\right\|_2 \leq \ell \|\mathbf{x}_t^{M,s+1} - \mathbf{x}_t^{M,s}\|_2.$$

Plugging this bound into (59) then adding and subtracting $\mathbf{x}_t^{B,s}$ gives:

$$\|\mathbf{x}_{t+1}^{B,s} - \mathbf{x}_{t+1}^{M,s+1}\|_2$$
$$\leq \|\mathbf{x}_t^{B,s} - \mathbf{x}_t^{M,s+1}\|_2 + \eta\ell\|\mathbf{x}_t^{M,s+1} - \mathbf{x}_t^{M,s}\|_2$$
$$\leq \|\mathbf{x}_t^{B,s} - \mathbf{x}_t^{M,s+1}\|_2 + \eta\ell\|\mathbf{x}_t^{M,s+1} - \mathbf{x}_t^{B,s}\|_2 + \eta\ell\|\mathbf{x}_t^{B,s} - \mathbf{x}_t^{M,s}\|_2$$
$$= (1 + \eta\ell)\|\mathbf{x}_t^{B,s} - \mathbf{x}_t^{M,s+1}\|_2 + \eta\ell\|\mathbf{x}_t^{B,s} - \mathbf{x}_t^{M,s}\|_2. \tag{59}$$

Using the fact that $\mathbf{x}_0^{B,s} = \mathbf{x}_0^{M,s+1}$ and iterating this inequality shows that:

$$\|\mathbf{x}_t^{B,s} - \mathbf{x}_t^{M,s+1}\|_2$$
$$\leq \eta\ell \sum_{k=0}^{t-1}(1 + \eta\ell)^k \|\mathbf{x}_{t-k}^{B,s} - \mathbf{x}_{t-k}^{M,s}\|_2$$
$$\overset{\text{Lemma 4}}{\leq} \left(2\ell\psi_2(s, \mathbf{z})\eta\sqrt{t}\right)\eta\ell \sum_{k=0}^{t-1}(1 + \eta\ell)^k$$
$$= \left(2\ell\psi_2(s, \mathbf{z})\eta\sqrt{t}\right)\left((1 + \eta\ell)^t - 1\right)$$
$$\leq \left(2\ell\psi_2(s, \mathbf{z})\eta\sqrt{t}\right)\left(e^{\eta t \ell} - 1\right).$$

The final inequality follows by taking logarithms and using the fact that $\log(1 + \eta\ell) \leq \eta\ell$. ∎

# F    Discretization bounds

**Proof of Lemma 6**    Recall that $\mathbf{y}_t^D = \mathbf{y}_{\lfloor t \rfloor}^C$ and so for all $k \in \mathbb{N}$, $\mathbf{y}_k^D = \mathbf{y}_k^C$. By the construction of Skorokhod solutions to the process $\mathbf{x}_t^C$ and $\mathbf{x}_t^D$, and using Theorem 9, we have for all $k \in \mathbb{N}$

$$\left\| \mathbf{x}_k^C - \mathbf{x}_k^D \right\| \leq (c_9 + 1) \sup_{0 \leq s \leq k} \left\| \mathbf{y}_s^C - \mathbf{y}_{\lfloor s \rfloor}^C \right\|.$$

Since

$$\mathbf{y}_t^C = \mathbf{x}_0^C - \eta \int_0^t \nabla_x \bar{f}(\mathbf{x}_s^C) ds + \sqrt{\frac{2\eta}{\beta}} \mathbf{w}_t,$$

the triangle inequality implies that

$$\left\| \mathbf{x}_k^C - \mathbf{x}_k^D \right\| \leq (c_9 + 1)\eta \sup_{s \in [0,k]} \left\| \int_{\lfloor s \rfloor}^s \nabla_x \bar{f}(\mathbf{x}_\tau^C) d\tau \right\| + (c_9 + 1)\sqrt{\frac{2\eta}{\beta}} \sup_{s \in [0,k]} \left\| \mathbf{w}_s - \mathbf{w}_{\lfloor s \rfloor} \right\|.$$

$\mathbb{E}\left[ \sup_{s \in [0,k]} \left\| \mathbf{w}_s - \mathbf{w}_{\lfloor s \rfloor} \right\| \right]$ is upper bounded by $2n\sqrt{\log(4k)}$. See Lemma 9 in [27]. So, the remaining work is to bound the first term on the right.

Take the expectation of the first term, we have

$$\mathbb{E}\left[ \sup_{s \in [0,k]} \left\| \int_{\lfloor s \rfloor}^s \nabla_x \bar{f}(\mathbf{x}_\tau^C) d\tau \right\| \right]$$

$$= \mathbb{E}\left[ \max_{i=0,\cdots,k-1} \sup_{s \in [i,i+1]} \left\| \int_i^s \nabla_x \bar{f}(\mathbf{x}_\tau^C) d\tau \right\| \right]$$

$$\leq \mathbb{E}\left[ \left( \sum_{i=0}^{k-1} \left( \sup_{s \in [i,i+1]} \left\| \int_i^s \nabla_x \bar{f}(\mathbf{x}_\tau^C) d\tau \right\| \right)^2 \right)^{1/2} \right]$$

$$\overset{\text{Jensen}}{\leq} \left( \mathbb{E}\left[ \sum_{i=0}^{k-1} \left( \sup_{s \in [i,i+1]} \left\| \int_i^s \nabla_x \bar{f}(\mathbf{x}_\tau^C) d\tau \right\| \right)^2 \right] \right)^{1/2}$$

$$= \left( \sum_{i=0}^{k-1} \mathbb{E}\left[ \left( \sup_{s \in [i,i+1]} \left\| \int_i^s \nabla_x \bar{f}(\mathbf{x}_\tau^C) d\tau \right\| \right)^2 \right] \right)^{1/2}.$$

So we want to upper bound the supremum inside the expectation operation.

We can show for all $s \in [0, k]$,

$$\left\| \int_{\lfloor s \rfloor}^s \nabla_x \bar{f}(\mathbf{x}_\tau^C) d\tau \right\| \overset{\text{triangle inequality}}{\leq} \int_{\lfloor s \rfloor}^s \left\| \nabla_x \bar{f}(\mathbf{x}_\tau) \right\| d\tau$$

$$\leq \int_{\lfloor s \rfloor}^{\lfloor s \rfloor + 1} \left\| \nabla_x \bar{f}(\mathbf{x}_\tau^C) \right\| d\tau$$

$$\overset{\text{Jensen}}{\leq} \left( \int_{\lfloor s \rfloor}^{\lfloor s \rfloor + 1} \left\| \nabla_x \bar{f}(\mathbf{x}_\tau^C) \right\|^2 d\tau \right)^{1/2}.$$

Therefore,

$$\mathbb{E}\left[ \sup_{s \in [0,k]} \left\| \int_{\lfloor s \rfloor}^s \nabla_x \bar{f}(\mathbf{x}_\tau^C) d\tau \right\| \right] \leq \left( \sum_{i=0}^{k-1} \mathbb{E}\left[ \int_i^{i+1} \left\| \nabla_x \bar{f}(\mathbf{x}_\tau^C) \right\|^2 d\tau \right] \right)^{1/2}$$

$$\overset{\text{Fubini}}{=} \left( \sum_{i=0}^{k-1} \int_i^{i+1} \mathbb{E}\left[ \left\| \nabla_x \bar{f}(\mathbf{x}_\tau^C 1) \right\|^2 \right] d\tau \right)^{1/2}.$$

Here, we can see it suffices to bound $\mathbb{E}\left[\|\nabla_x \bar{f}(\mathbf{x}_t)\|^2\right]$.

We have assumed that $0 \in \mathcal{K}$, and so we have

$$
\begin{aligned}
\left\|\nabla_x \bar{f}(\mathbf{x}_t^C)\right\|^2 &= \left\|\nabla_x \bar{f}(\mathbf{x}_t^C) - \nabla_x \bar{f}(0) + \nabla_x \bar{f}(0)\right\|^2 \\
&\leq 2\left\|\nabla_x \bar{f}(\mathbf{x}_t^C) - \nabla_x \bar{f}(0)\right\|^2 + 2\left\|\nabla_x \bar{f}(0)\right\|^2 \\
&\leq 2\ell^2 \left\|\mathbf{x}_t^C\right\|^2 + 2\left\|\nabla_x \bar{f}(0)\right\|^2.
\end{aligned}
$$

Plugging in the bound from Lemma 14 shows that

$$
\begin{aligned}
\mathbb{E}\left[\|\nabla_x \bar{f}(\mathbf{x}_t^C)\|^2\right] &\leq \mathbb{E}\left[2\ell^2\|\mathbf{x}_t^C\|^2 + 2\|\nabla_x \bar{f}(0)\|^2\right] \\
&= 2\ell^2 \mathbb{E}\left[\|\mathbf{x}_t^C\|^2\right] + 2\|\nabla_x \bar{f}(0)\|^2 \\
&\leq 2\ell^2\left(\varsigma + \frac{1}{\mu}c_{10}\right) + 2\|\nabla_x \bar{f}(0)\|^2.
\end{aligned}
$$

Therefore, we have

$$
\begin{aligned}
\mathbb{E}\left[\sup_{s \in [0,k]}\left\|\int_{\lfloor s \rfloor}^s \nabla_x \bar{f}(\mathbf{x}_\tau^C) d\tau\right\|\right] &\leq \left(\sum_{i=0}^{k-1}\int_{J_i}^{i+1} \mathbb{E}\left[\|\nabla_x \bar{f}(\mathbf{x}_\tau^C)\|^2\right] d\tau\right)^{1/2} \\
&\leq \sqrt{2\ell^2\left(\varsigma + \frac{1}{\mu}c_{10}\right) + 2\|\nabla_x \bar{f}(0)\|^2}\sqrt{k} \\
&\leq \left(\sqrt{\frac{2}{\mu}\ell^2 c_{10} + 2\|\nabla_x \bar{f}(0)\|^2} + \sqrt{2\ell^2}\sqrt{\varsigma}\right)\sqrt{k}.
\end{aligned}
$$

Setting

$$
\begin{aligned}
c_5 &= (c_9 + 1)\sqrt{\frac{2}{\mu}\ell^2 c_{10} + 2\|\nabla_x \bar{f}(0)\|^2} \\
c_6 &= (c_9 + 1)\sqrt{2\ell^2} \\
c_7 &= (c_9 + 1)n\sqrt{\frac{8}{\beta}}
\end{aligned}
$$

where $c_{10}$ is defined in Lemma 14 and $c_9$ is defined in Theorem 9 and combining the bound on the second supreme term gives the desired result. ∎

**Proof of Lemma 7** The argument of bounding $\mathbf{x}_t^M$ and $\mathbf{x}_t^D$ closely follows the proof of Lemma 10 in [27]. Recall that $\mathbf{x}_t^M$ is a discretized process and $\mathbf{x}_t^M = \mathbf{x}_{\lfloor t \rfloor}^M$. We also have $\mathbf{x}_t^M = \mathcal{S}(\mathcal{D}(\mathbf{y}_t^M))$, where $\mathbf{y}_t^M$ is defined by

$$
\mathbf{y}_t^M = \mathbf{x}_0^M - \eta \int_0^t \nabla \bar{f}(\mathbf{x}_{\lfloor s \rfloor}^M) ds + \sqrt{\frac{2\eta}{\beta}}\mathbf{w}_t.
$$

The intermediate process $\mathbf{x}_t^D$ satisfies $\mathbf{x}_t^D = \mathcal{S}(\mathcal{D}(\mathbf{y}_t^C))$, where

$$
\mathbf{y}_t^C = \mathbf{x}_0^C - \eta \int_0^t \nabla \bar{f}(\mathbf{x}_s^C) ds + \sqrt{\frac{2\eta}{\beta}}\mathbf{w}_t.
$$

So in particular,

$$
\begin{aligned}
\mathbf{x}_{k+1}^M &= \Pi_{\mathcal{K}}\left(\mathbf{x}_k^M + \mathbf{y}_{k+1}^M - \mathbf{y}_k^M\right) \\
&= \Pi_{\mathcal{K}}\left(\mathbf{x}_k^M - \eta\nabla \bar{f}(\mathbf{x}_k^M) + \sqrt{\frac{2\eta}{\beta}}(\mathbf{w}_{k+1} - \mathbf{w}_k)\right) \\
\mathbf{x}_{k+1}^D &= \Pi_{\mathcal{K}}\left(\mathbf{x}_k^D + \mathbf{y}_{k+1}^C - \mathbf{y}_k^C\right) \\
&= \Pi_{\mathcal{K}}\left(\mathbf{x}_k^D - \eta\int_k^{k+1}\nabla \bar{f}(\mathbf{x}_s^C) ds + \sqrt{\frac{2\eta}{\beta}}(\mathbf{w}_{k+1} - \mathbf{w}_k)\right).
\end{aligned}
$$

Define a difference process

$$\boldsymbol{\rho}_t = \left(\mathbf{x}_t^M + \mathbf{y}_t^M - \mathbf{y}_{\lfloor t \rfloor}^M\right) - \left(\mathbf{x}_t^D + \mathbf{y}_t^C - \mathbf{y}_{\lfloor t \rfloor}^C\right).$$

Note that at integers $k \in \mathbb{N}$, $\boldsymbol{\rho}_k = \mathbf{x}_k^M - \mathbf{x}_k^D$ and for $t \in [k, k+1)$, we have

$$\boldsymbol{\rho}_t = \left(\mathbf{x}_k^M - \mathbf{y}_k^M - \mathbf{x}_k^D + \mathbf{y}_k^D\right) + \mathbf{y}_t^M - \mathbf{y}_t^C.$$

It follows that

$$d\boldsymbol{\rho}_t = d(\mathbf{y}_t^M - \mathbf{y}_t^C) = \eta\left(\nabla\bar{f}(\mathbf{x}_t^C) - \nabla\bar{f}(\mathbf{x}_t^M)\right)$$

By construction, $\boldsymbol{\rho}_t$ is a continuous bounded variation process on the interval $[k, k+1)$. Thus, when $\boldsymbol{\rho}_t \neq 0$, we can calculate $d\|\boldsymbol{\rho}_t\|$ using the chain rule.

$$
\begin{aligned}
d\|\boldsymbol{\rho}_t\| &\stackrel{\text{chain rule}}{=} \left(\frac{\boldsymbol{\rho}_t}{\|\boldsymbol{\rho}_t\|}\right)^{\top} d\boldsymbol{\rho}_t \\
&= \left(\frac{\boldsymbol{\rho}_t}{\|\boldsymbol{\rho}_t\|}\right)^{\top} \eta\left(\nabla\bar{f}(\mathbf{x}_t^C) - \nabla\bar{f}(\mathbf{x}_t^M)\right) dt \\
&\stackrel{\text{Cauchy-Schwarz}}{\leq} \eta\left\|\nabla\bar{f}(\mathbf{x}_t^C) - \nabla\bar{f}(\mathbf{x}_t^M)\right\| dt \\
&\stackrel{\text{Lipschitz}}{\leq} \eta\ell\left\|\mathbf{x}_t^C - \mathbf{x}_t^M\right\| dt \\
&= \eta\ell\left\|\mathbf{x}_t^C - \mathbf{x}_t^D + \mathbf{x}_t^D - \mathbf{x}_t^M\right\| dt \\
&\stackrel{\text{triangle}}{\leq} \eta\ell\left(\left\|\mathbf{x}_t^C - \mathbf{x}_t^D\right\| + \left\|\mathbf{x}_t^D - \mathbf{x}_t^M\right\|\right) dt.
\end{aligned}
$$

To include the case that $\boldsymbol{\rho}_t = 0$, we use the Lemma 19 from [27]. The analysis is as below:

For $t \in [k, k+1)$,

$$
\begin{aligned}
\|\boldsymbol{\rho}_t\| &= \|\boldsymbol{\rho}_k\| + \int_k^t d\|\boldsymbol{\rho}_t\| \\
&= \|\boldsymbol{\rho}_k\| + \lim_{\epsilon\downarrow 0}\int_k^t \mathbb{1}\left(\|\boldsymbol{\rho}_s\| \geq \epsilon\right) d\|\boldsymbol{\rho}_s\| \\
&\leq \|\boldsymbol{\rho}_k\| + \lim_{\epsilon\downarrow 0}\int_k^t \mathbb{1}\left(\|\boldsymbol{\rho}_s\| \geq \epsilon\right)\eta\ell\left(\left\|\mathbf{x}_s^C - \mathbf{x}_s^D\right\| + \left\|\mathbf{x}_s^D - \mathbf{x}_s^M\right\|\right) dt \\
&= (1 + \eta\ell)\|\boldsymbol{\rho}_k\| + \eta\ell\int_k^t \left(\left\|\mathbf{x}_s^C - \mathbf{x}_s^D\right\|\right) ds.
\end{aligned}
$$

The second equality follows from Lemma 19 from [27]. The last equality holds because that $\boldsymbol{\rho}_k = \mathbf{x}_s^M - \mathbf{x}_s^D, \forall s \in [k, k+1)$.

Non-expansiveness of the convex projection implies that

$$\|\boldsymbol{\rho}_k\| = \left\|\mathbf{x}_s^M - \mathbf{x}_s^D\right\| \leq \lim_{t\uparrow k}\|\boldsymbol{\rho}_t\|. \tag{60}$$

Letting $t = k + 1$ gives

$$\left\|\boldsymbol{\rho}_{k+1}\right\| \leq (1 + \eta\ell)\|\boldsymbol{\rho}_k\| + \eta\ell\int_k^{k+1}\left\|\mathbf{x}_s^C - \mathbf{x}_s^D\right\| ds.$$

Iterating this inequality, and using the assumption that $\mathbf{x}_0^M = \mathbf{x}_0^D$ gives

$$\|\boldsymbol{\rho}_k\| \leq \sum_{i=0}^{k-1}\eta\ell(1 + \eta\ell)^{k-i-1}\int_i^{i+1}\left\|\mathbf{x}_s^C - \mathbf{x}_s^D\right\| ds.$$

Taking expectation, and using Lemma 6 gives

$$\mathbb{E}\left[\|\boldsymbol{\rho}_k\|\right] \leq \sum_{i=0}^{k-1} \eta\ell(1+\eta\ell)^{k-i-1} \int_i^{i+1} \left( (c_5 + c_6\sqrt{\varsigma})\eta\sqrt{s} + c_7\sqrt{\eta\log(4s)} \right) ds$$

$$\leq \eta\ell\left( (c_5 + c_6\sqrt{\varsigma})\eta\sqrt{k} + c_7\sqrt{\eta\log(4k)} \right) \sum_{i=0}^{k-1}(1+\eta\ell)^{k-i-1}$$

$$\leq \left( (c_5 + c_6\sqrt{\varsigma})\eta\sqrt{k} + c_7\sqrt{\eta\log(4k)} \right) \left( (1+\eta\ell)^k - 1 \right)$$

$$\leq \left( (c_5 + c_6\sqrt{\varsigma})\eta\sqrt{k} + c_7\sqrt{\eta\log(4k)} \right) \left( e^{\eta\ell k} - 1 \right).$$

The last inequality is based on the fact that $(1+\eta\ell)^k \leq e^{\eta\ell k}$ for all $\eta\ell > 0$.

Recall that for all $k \in \mathbb{N}$, $\boldsymbol{\rho}_k = \mathbf{x}_k^M - \mathbf{x}_k^D$, which gives the desired result. ∎

## G   Conclusion of the proof of Lemma 3

This subsection uses a "switching" trick to derive a bound on $W_1(\mathcal{L}(\mathbf{x}_k^A), \mathcal{L}(\mathbf{x}_k^C))$ that is uniform in time. The essential idea is to utilize a family of processes that switch from the dynamics of $\mathbf{x}_k^A$ to the dynamics of $\mathbf{x}_k^C$, and utilize contractivity of the law of $\mathbf{x}_k^C$ to derive the uniform bounds. A similar methodology was utilized in [10].

For $s \geq 0$, let $\mathbf{x}_{s,t}^{A,C}$ be the process such that $\mathbf{x}_{s,t}^{A,C} = \mathbf{x}_t^A = \mathbf{x}_{\lfloor t \rfloor}^A$ for $t \leq s$ and for $t \geq s$, $\mathbf{x}_{s,t}^{A,C}$ follows:

$$d\mathbf{x}_{s,t}^{A,C} = -\eta\nabla_x\bar{f}(\mathbf{x}_{s,t}^{A,C})dt + \sqrt{\frac{2\eta}{\beta}}d\mathbf{w}_t - \mathbf{v}_{s,t}^{A,C}d\boldsymbol{\mu}_s^{A,C}(t).$$

In other words, $\mathbf{x}_{s,t}^{A,C}$ follows the algorithm for $t \leq s$, and then switches to the dynamics of the continuous-time approximation from (8) at $t = s$.

Now let $0 \leq s \leq \hat{s} \leq t$ where $s, \hat{s} \in \mathbb{N}$, then Corollary 18 from Appendix D shows that

$$W_1(\mathcal{L}(\mathbf{x}_{s,t}^{A,C}), \mathcal{L}(\mathbf{x}_{\hat{s},t}^{A,C})) \leq 2\varphi(R)^{-1}e^{-\tilde{a}(t-\hat{s})}W_1(\mathcal{L}(\mathbf{x}_{s,\hat{s}}^{A,C}), \mathcal{L}(\mathbf{x}_{\hat{s},\hat{s}}^{A,C})). \tag{61}$$

By starting the analysis of the processes $\mathbf{x}^A$ and $\mathbf{x}^C$ at time $s$, rather than time $0$, Lemma 8 implies the following bound:

$$W_1(\mathcal{L}(\mathbf{x}_{s,\hat{s}}^{A,C}), \mathcal{L}(\mathbf{x}_{\hat{s},\hat{s}}^{A,C})) = W_1(\mathcal{L}(\mathbf{x}_{s,\hat{s}}^{A,C}), \mathcal{L}(\mathbf{x}_{\hat{s}}^A))$$

$$\leq \left( \left( c_8 + c_6\sqrt{\mathbb{E}[\|\mathbf{x}_s^A\|^2]} \right) \eta\sqrt{\hat{s}-s} + c_7\sqrt{\eta\log(4(\hat{s}-s))} \right) e^{\eta\ell(\hat{s}-s)}$$

$$\leq \left( (c_8 + c_6\sqrt{\varsigma + c_{11}})\eta\sqrt{\hat{s}-s} + c_7\sqrt{\eta\log(4(\hat{s}-s))} \right) e^{\eta\ell(\hat{s}-s)}. \tag{62}$$

The second inequality is based on Lemma 15.

Let $H = \lfloor 1/\eta \rfloor$ and $t \in [\hat{k}H, (\hat{k}+1)H)$ where $\hat{k} \in \mathbb{N}$, we have $\mathbf{x}_{0,t}^{A,C} = \mathbf{x}_t^C$ and $\mathbf{x}_{(\hat{k}+1)H,t}^{A,C} = \mathbf{x}_t^A = \mathbf{x}_{\lfloor t \rfloor}^A$. Then, the triangle inequality implies that

$$W_1(\mathcal{L}(\mathbf{x}_t^A), \mathcal{L}(\mathbf{x}_t^C)) \leq \sum_{i=0}^{\hat{k}} W_1(\mathcal{L}(\mathbf{x}_{iH,t}^{A,C}), \mathcal{L}(\mathbf{x}_{(i+1)H,t}^{A,C})).$$

For $i < \hat{k}$, setting $s = iH$, $\hat{s} = (i+1)H$ in (61) gives that

$$W_1(\mathcal{L}(\mathbf{x}_{iH,t}^{A,C}), \mathcal{L}(\mathbf{x}_{(i+1)H,t}^{A,C})) \leq 2\varphi(R)^{-1}e^{-\tilde{a}(t-(i+1)H)}W_1(\mathcal{L}(\mathbf{x}_{iH,(i+1)H}^{A,C}), \mathcal{L}(\mathbf{x}_{(i+1)H,(i+1)H}^{A,C}))$$

$$\leq 2\varphi(R)^{-1}e^{-\eta a(t-(i+1)H)}g(H)$$

$$\leq 2\varphi(R)^{-1}e^{-a(\hat{k}-i-1)/2)}g(\eta^{-1})$$

where

$$g(r) = \left( \left( c_8 + c_6 \sqrt{\varsigma + c_{11}} \right) \eta \sqrt{r} + c_7 \sqrt{\eta \log(4r)} \right) e^{\eta \ell r}. \tag{63}$$

The last inequality uses the facts that $1/2 \le \eta H \le 1$ along with monotonicity of $g$. The lower bound of $\eta H$ arises because $H \ge \eta^{-1} - 1$ and so $\eta H \ge 1 - \eta \ge 1/2$, since $\eta \le 1/2$. Thus, the first $\hat{k}$ terms are bounded by:

$$\sum_{i=0}^{\hat{k}-1} W_1(\mathcal{L}(\mathbf{x}_{iH,t}^{A,C}), \mathcal{L}(\mathbf{x}_{(i+1)H,t}^{A,C})) \le \sum_{i=0}^{\hat{k}-1} 2\varphi(R)^{-1} e^{-a(\hat{k}-i-1)/2} g(\eta^{-1})$$

$$\le 2\varphi(R)^{-1} \frac{g(\eta^{-1})}{1 - e^{-a/2}}$$

For $i = \hat{k}$,

$$W_1(\mathcal{L}(\mathbf{x}_{iH,t}^{A,C}), \mathcal{L}(\mathbf{x}_{(i+1)H,t}^{A,C})) = W_1(\mathcal{L}(\mathbf{x}_{\hat{k}H,t}^{A,C}), \mathcal{L}(\mathbf{x}_t^{A}))$$

$$\le g(t - \hat{k}H) \le g(\eta^{-1})$$

By triangle inequality, adding all the $\hat{k} + 1$ terms gives

$$W_1(\mathcal{L}(\mathbf{x}_t^A), \mathcal{L}(\mathbf{x}_t^C))$$

$$\le g(\eta^{-1}) \left( 1 + \frac{2\varphi(R)^{-1}}{1 - e^{-a/2}} \right)$$

$$\le \left( \left( c_8 + c_6 \sqrt{\varsigma + c_{11}} \right) \eta \sqrt{\eta^{-1}} + c_7 \sqrt{\eta \log(4\eta^{-1})} \right) e^{\ell} \left( 1 + \frac{2\varphi(R)^{-1}}{1 - e^{-a/2}} \right). \tag{64}$$

For $\eta^{-1} \ge 4$, we have $\log(4\eta^{-1}) \le 2\log(\eta^{-1})$, and also $\log \eta^{-1} > 1$. Thus, if $\eta \le 1/4$, then (64) can be further upper bounded by

$$W_1(\mathcal{L}(\mathbf{x}_t^A), \mathcal{L}(\mathbf{x}_t^C))$$

$$\le \left( \left( c_8 + c_6 \sqrt{\varsigma + c_{11}} \right) \eta \sqrt{\eta^{-1} \log(\eta^{-1})} + c_7 \sqrt{2\eta \log(\eta^{-1})} \right) e^{\ell} \left( 1 + \frac{2\varphi(R)^{-1}}{1 - e^{-a/2}} \right)$$

$$= \left( c_8 + c_6 \sqrt{\varsigma + c_{11}} + \sqrt{2} c_7 \right) e^{\ell} \left( 1 + \frac{2\varphi(R)^{-1}}{1 - e^{-a/2}} \right) \sqrt{\eta \log(\eta^{-1})}$$

$$\le \left( c_8 + c_6 \sqrt{c_{11}} + \sqrt{2} c_7 + c_6 \sqrt{\varsigma} \right) e^{\ell} \left( 1 + \frac{2\varphi(R)^{-1}}{1 - e^{-a/2}} \right) \sqrt{\eta \log(\eta^{-1})}.$$

So setting

$$c_3 = \left( c_8 + c_6 \sqrt{c_{11}} + \sqrt{2} c_7 \right) e^{\ell} \left( 1 + \frac{2\varphi(R)^{-1}}{1 - e^{-a/2}} \right)$$

$$c_4 = c_6 e^{\ell} \left( 1 + \frac{2\varphi(R)^{-1}}{1 - e^{-a/2}} \right)$$

completes the proof. ∎

## H  Bounding the constants

In this section, we summarize all the constants in Table 1. The second column of the table points to the place where these values are defined or computed. Then we show the simplified bounds of the main constants $c_1, c_2, c_3, c_4, a$ in Theorem 1 explicitly and also discuss their dependencies on state dimension $n$ and parameter $\beta$.

**Proposition 19.** *The constants $c_2$ and $c_4$ grow linearly with $n$. The constants $c_1$ and $c_3$ have $O(\sqrt{n})$ and $O(n)$ dependencies respectively. So overall, the dimension dependency of convergence guarantee is $O(n)$. Constants $c_1, c_2, c_3, c_4$ all grow exponentially with respect to $\frac{\beta \ell R^2}{2}$. And for all $\beta > 0$,*
$a \ge \frac{2}{\frac{\beta R^2}{2} + \frac{16}{\mu}} e^{-\frac{\beta \ell R^2}{4}}.$

Table 1: List of constants

| Constant | Definition |
|---|---|
| $a = \frac{2\xi}{\beta}$ | |
| $c_1 = 2\varphi(R)^{-1}\sqrt{\frac{2}{\mu}c_{10}}$ | |
| $c_2 = 4\varphi(R)^{-1}$ | Appendix D.2 (Proof of Lemma 2) |
| $c_3 = \left(c_8 + c_6\sqrt{c_{11}} + \sqrt{2}c_7\right)e^\ell\left(1 + \frac{2\varphi(R)^{-1}}{1 - e^{-a/2}}\right)$ | |
| $c_4 = c_6 e^\ell\left(1 + \frac{2\varphi(R)^{-1}}{1 - e^{-a/2}}\right)$ | Appendix G (Proof of Lemma 3) |
| $c_5 = (c_9 + 1)\sqrt{\frac{2}{\mu}\ell^2 c_{10} + 2\|\nabla_x \bar{f}(0)\|^2}$ | |
| $c_6 = (c_9 + 1)\sqrt{2\ell^2}$ | |
| $c_7 = (c_9 + 1)n\sqrt{\frac{8}{\beta}}$ | Appendix F (Proof of Lemma 6) |
| $c_8 = c_5 + 2\ell\Psi_2(\mathbf{z})$ | Section 3.3 (Proof of Lemma 8) |
| $c_9 = 6\left(\frac{1}{\alpha}\right)^{\operatorname{rank}(A)/2}$ | Appendix A.3 (Proof of Lemma 11) |
| $c_{10} = (\ell + \mu)R^2 + R\|\nabla_x \bar{f}(0)\| + \frac{n}{\beta}$ | Appendix C.1 (Proof of Lemma 13) |
| $c_{11} = \frac{4}{\mu}\left(\frac{n}{\beta} + (\ell + \mu)R^2 + (2 + R)\|\nabla_x \bar{f}(0)\| + \left(8\ell^2 + \frac{1}{\mu}\right)\ell^2\mathcal{M}_2(\mathbf{z})\right)$ | Appendix C.2 (Proof of Lemma 15) |

**Proof of Proposition 19**  Recall that $a = 2\xi/\beta$, and from (41c) we have that from

$$\xi^{-1} = \int_0^{R_1} \Phi(s)\varphi(s)^{-1}ds.$$

So, to get a lower bound on $\xi$, we need an upper bound on the right side. Recalling the definitions of the various functions for our scenario gives:

$$h(s) = \frac{\ell\beta\min\{s^2, R^2\}}{8}$$
$$\varphi(s) = e^{-h(s)}$$
$$\Phi(s) = \int_0^s \varphi(r)dr.$$

It follows that $\Phi(s) \leq s$ and $\varphi(s)^{-1} = e^{h(s)} \leq e^{\frac{\ell\beta R^2}{8}}$. Thus, we have that

$$\xi^{-1} \leq \frac{1}{2}R_1^2 e^{\frac{\ell\beta R^2}{8}}.$$

Now, note that in Corollary 17 that we have set

$$R_1 = \frac{R}{2} + \frac{1}{2}\sqrt{R^2 + \frac{32}{\mu\beta}e^{\frac{\beta\ell R^2}{8}}}.$$

So, a bit of crude upper bounding gives:

$$\xi^{-1} \le \frac{1}{2} R_1^2 e^{\frac{\ell \beta R^2}{8}}$$

$$\le \frac{1}{2} \left( R^2 + \frac{32}{\mu\beta} e^{\frac{\beta \ell R^2}{8}} \right) e^{\frac{\beta \ell R^2}{8}}$$

$$\le \left( \frac{R^2}{2} + \frac{16}{\mu\beta} \right) e^{\frac{\beta \ell R^2}{4}}$$

The final bound on $a$ becomes:

$$a = 2\xi/\beta \ge \frac{2}{\frac{\beta R^2}{2} + \frac{16}{\mu}} e^{-\frac{\beta \ell R^2}{4}}$$

The rest of focuses on bounding the other constants as $\beta$ grows large. For all sufficiently large $\beta$, we have that

$$\frac{\frac{\beta R^2}{2} + \frac{16}{\mu}}{2} \le e^{\frac{\beta \ell R^2}{4}}$$

so that

$$a \ge e^{-\frac{\beta \ell R^2}{2}}. \tag{65}$$

We have the following inequality for all sufficiently large $\beta$:

$$\frac{1}{1 - e^{-a/2}} \le \max\left\{ \frac{4}{a}, \frac{1}{1 - e^{-1}} \right\}$$

$$\le \max\left\{ 4e^{\frac{\beta \ell R^2}{2}}, \frac{1}{1 - e^{-1}} \right\}$$

$$= 4e^{\frac{\beta \ell R^2}{2}}.$$

The first inequality uses the fact that for all $y > 0$, $\frac{1}{1-e^{-y}} \le \max\left\{ \frac{2}{y}, \frac{1}{1-e^{-1}} \right\}$, which is shown in [27].

So

$$1 + \frac{2\varphi(R)^{-1}}{1 - e^{-a/2}} \le 1 + 4e^{\frac{\beta \ell R^2}{2}}. \tag{66}$$

Now we bound the growth of the other constants for large $\beta$. So, without loss of generality, assume $\beta \ge 1$. Then, plugging the definition of $\xi$ and $\varphi$ and (66) gives

$$c_1 = 2e^{\frac{\beta \ell R^2}{8}} \sqrt{\frac{2}{\mu} \left( (\ell + \mu)R^2 + R\|\nabla_x \bar{f}(0)\| + \frac{n}{\beta} \right)}$$

$$\le 2e^{\frac{\beta \ell R^2}{8}} \sqrt{\frac{2}{\mu} \left( (\ell + \mu)R^2 + R\|\nabla_x \bar{f}(0)\| + n \right)}$$

$$c_2 = 4e^{\frac{\beta \ell R^2}{8}}$$

$$c_3 = \left( c_8 + c_6\sqrt{c_{11}} + \sqrt{2}c_7 \right) e^{\ell} \left( 1 + \frac{2\varphi(R)^{-1}}{1 - e^{-a/2}} \right)$$

$$\le \left( c_8 + c_6\sqrt{c_{11}} + \sqrt{2}c_7 \right) e^{\ell} \left( 1 + 4e^{\frac{\beta \ell R^2}{2}} \right)$$

$$\le r(\sqrt{n})e^{\ell} \left( 1 + 4e^{\frac{\beta \ell R^2}{2}} \right)$$

$$c_4 = \left( 6(\frac{1}{\alpha})^{\text{rank}(A)/2} + 1 \right) \sqrt{2\ell^2} e^{\ell} \left( 1 + \frac{2\varphi(R)^{-1}}{1 - e^{-a/2}} \right)$$

$$\le \left( 6(\frac{1}{\alpha})^{\text{rank}(A)/2} + 1 \right) \sqrt{2\ell^2} e^{\ell} \left( 1 + 4e^{\frac{\beta \ell R^2}{2}} 1 \right).$$

For constant $c_3$, $r(\sqrt{n})$ is a monotonically increasing function of order $\sqrt{n}$, (independent of $\eta$ and $\beta$). The upper bound of $c_3$ is derived by direct observation of the corresponding constants.

We can see neither $c_2$ nor $c_4$ depends on the state dimension, so the two constants grow linearly with $n$. The constant $c_1$ are $O(\sqrt{n})$ and $c_3$ are $O(n)$. As for the dependencies on $\beta$, we can see that all four constants are $O(e^{\frac{\beta \ell R^2}{2}})$.

■

# I   Near-optimality of Gibbs distributions

In this appendix, we prove Proposition 22 which shows that $\mathbf{x}_k$ can be near-optimal. The proof closely follows [27] and [35]. The main difference is that in our case we have to deal with the unbounded polyhedral constraint, while in [35] there is no constraint and in [27] the constraint is compact.

Firstly, we need a preliminary result shown as below.

**Lemma 20.** *Assume* $\mathbf{x}$ *is drawn according to* $\pi_{\beta \bar{f}}$. *There exists a positive constant* $c_{12}$ *such that the following bounds hold:*

$$\mathbb{E}[\bar{f}(\mathbf{x})] \leq \min_{x \in \mathcal{K}} \bar{f}(x) + \frac{n}{\beta} \left( 2 \max\{0, \log \varsigma\} + c_{12} \right)$$

*where* $c_{12} = \log n + 2 \log(1 + \frac{1}{\mu} c_{10}) + \frac{1}{6} \log 3 + \log 2\sqrt{\pi} - \log r_{\min}$ *and* $r_{\min}$ *is a positive constant.*

**Proof of Lemma 20**   Recall that the probability measure $\pi_{\beta \bar{f}}(A)$ is defined by $\pi_{\beta \bar{f}(A)}(A) = \frac{\int_{A \cap \mathcal{K}} e^{-\beta \bar{f}(x)} dx}{\int_{\mathcal{K}} e^{-\beta \bar{f}(y)} dy}$.

Let $\Lambda = \int_{\mathcal{K}} e^{-\beta \bar{f}(y)} dy$ and $p(x) = \frac{e^{-\beta \bar{f}(x)}}{\Lambda}$. So $\log p(x) = -\beta \bar{f}(x) - \log \Lambda$, which implies that $\bar{f}(x) = -\frac{1}{\beta} \log p(x) - \frac{1}{\beta} \log \Lambda$. Then we have

$$\mathbb{E}_{\pi_{\beta \bar{f}}}[\bar{f}(\mathbf{x})] = \int_{\mathcal{K}} \bar{f}(x) p(x) dx$$

$$= -\frac{1}{\beta} \int_{\mathcal{K}} p(x) \log p(x) dx - \frac{1}{\beta} \log \Lambda. \tag{67}$$

We can bound the first term by maximizing the differential entropy.

Let $h(x) = -\int_{\mathcal{K}} p(x) \log p(x) dx$. Using the fact that the differential entropy of a distribution with finite moments is upper-bounded by that of a Gaussian density with the same second moment (see Theorem 8.6.5 in [15]), we have

$$h(x) \leq \frac{n}{2} \log(2\pi e \sigma^2) \leq \frac{n}{2} \log(2\pi e (\varsigma + \frac{1}{\mu} c_{10})), \tag{68}$$

where $\sigma^2 = \mathbb{E}_{\pi_{\beta \bar{f}}}[\|\mathbf{x}\|^2]$ and the second inequality uses Lemma 14.

We aim to derive the upper bound of the second term of (67).

First we show that there is a vector $x^\star \in \mathcal{K}$ which minimizes $\bar{f}$ over $\mathcal{K}$. In other words, an optimal solution exists. The bound (25) from the proof of Lemma 12 implies that $\bar{f}(x) \geq \bar{f}(0) + 1$ for all sufficiently large $x$. This implies that there is a compact ball, $B$ such that if $x_n \in \mathcal{K}$ is a sequence such that $\lim_{n \to \infty} \bar{f}(x_n) = \inf_{x \in \mathcal{K}} \bar{f}(x)$, then $x_n$ must be in $B \cap \mathcal{K}$ for all sufficiently large $n$. Then since $\bar{f}$ is continuous and $B \cap \mathcal{K}$ is compact, there must be a limit point $x^\star \in B \cap \mathcal{K}$ which minimizes $\bar{f}$.

Let $x^* \in \mathcal{K}$ be a minimizer. The normalizing constant can be expressed as:

$$\log \Lambda = \log \int_{\mathcal{K}} e^{-\beta \bar{f}(x)} dx$$

$$= \log e^{-\beta \bar{f}(x^*)} \int_{\mathcal{K}} e^{\beta (\bar{f}(x^*) - \bar{f}(x))} dx$$

$$= -\beta \bar{f}(x^*) + \log \int_{\mathcal{K}} e^{\beta (\bar{f}(x^*) - \bar{f}(x))} dx$$

So, to derive our desired upper bound on $-\log \Lambda$, it suffices to derive a lower bound on

$$\int_{\mathcal{K}} e^{\beta(\bar{f}(x^*) - \bar{f}(x))} dx. \tag{69}$$

We have

$$\bar{f}(x) - \bar{f}(x^*) = \int_0^1 \nabla \bar{f}(x^* + t(x - x^*))^\top (x - x^*) dt.$$

Let $y = x^* + t(x - x^*)$, $t \in [0, 1]$, then

$$\begin{aligned}
\|\nabla \bar{f}(y)\| &= \|\nabla \bar{f}(y) - \nabla \bar{f}(x^*) + \nabla \bar{f}(x^*) - \nabla \bar{f}(0) + \nabla \bar{f}(0)\| \\
&\leq \ell \|y - x^*\| + \ell \|x^*\| + \|\nabla \bar{f}(0)\| \\
&\leq \ell \|x - x^*\| t + \ell \|x^*\| + \|\nabla \bar{f}(0)\|.
\end{aligned}$$

We can show $\|x^*\|$ is upper bounded by $\max\{R, \frac{\|\nabla \bar{f}(0)\|}{\mu}\}$.

We have to find the bound for the case $\|x^*\| > R$.

The convexity outside a ball assumption gives

$$\left(\nabla \bar{f}(x^*) - \nabla \bar{f}(0)\right)^\top x^* \geq \mu \|x^*\|^2. \tag{70}$$

The optimality of $x^*$ gives $-\nabla \bar{f}(x^*) \in N_{\mathcal{K}}(x^*)$, which is to say for all $y \in \mathcal{K}$, $-\nabla \bar{f}(x^*)^\top (y - x^*) \leq 0$. Since $0 \in \mathcal{K}$, $\nabla \bar{f}(x^*)^\top x^* \leq 0$ holds. Applying the Cauchy-Schwartz inequality to the left side of (70) gives

$$\|\nabla \bar{f}(0)\| \|x^*\| \geq \mu \|x^*\|^2.$$

This implies that $\|x^*\| \leq \frac{\|\nabla \bar{f}(0)\|}{\mu}$. So we can conclude that $\|x^*\| \leq \max\{R, \frac{\|\nabla \bar{f}(0)\|}{\mu}\} = c_{13}$.

Therefore,

$$\begin{aligned}
\bar{f}(x) - \bar{f}(x^*) &\leq \int_0^1 \|\nabla \bar{f}(x^* + t(x - x^*))\| \|x - x^*\| dt \\
&\leq \frac{\ell}{2} \|x - x^*\|^2 + \left(\ell \|x^*\| + \|\nabla \bar{f}(0)\|\right) \|x - x^*\| \\
&\leq \frac{\ell}{2} \|x - x^*\|^2 + \left(\ell c_{13} + \|\nabla \bar{f}(0)\|\right) \|x - x^*\|.
\end{aligned}$$

To lower-bound the integral from (69), we restrict our attention to the points $x$ such that the integrand is at least $1/2$. For these values, we have the following implications:

$$\begin{aligned}
&e^{\beta(\bar{f}(x^*) - \bar{f}(x))} \geq 1/2 \\
\iff &\beta\left(\bar{f}(x^*) - \bar{f}(x)\right) \geq -\log 2 \\
\impliedby &-\frac{\ell}{2} \|x - x^*\|^2 - \left(\ell c_{13} + \|\nabla \bar{f}(0)\|\right) \|x - x^*\| \geq -\frac{1}{\beta} \log 2.
\end{aligned}$$

So solving the corresponding quadratic equation and taking the positive root gives an upper bound of $\|x - x^*\|$:

$$\|x - x^*\| \leq -\frac{1}{\ell} \left(\ell c_{13} + \|\nabla \bar{f}(0)\|\right) + \frac{1}{\ell} \sqrt{\left(\ell c_{13} + \|\nabla \bar{f}(0)\|\right)^2 + 2\ell \frac{1}{\beta} \log 2}.$$

So let $\epsilon = -\frac{1}{\ell} \left(\ell c_{13} + \|\nabla \bar{f}(0)\|\right) + \frac{1}{\ell} \sqrt{\left(\ell c_{13} + \|\nabla \bar{f}(0)\|\right)^2 + 2\ell \frac{1}{\beta} \log 2}$ and let $\mathcal{B}_{x^*}(\epsilon)$ be the ball of radius $\epsilon$ centered at $x^*$. Then we want to find a ball $\mathcal{S}$ such that

$$\int_{\mathcal{K}} e^{\beta(\bar{f}(x^*) - \bar{f}(x))} dx \geq \frac{1}{2} \text{vol}(\mathcal{K} \cap \mathcal{B}_{x^*}(\epsilon)) \geq \frac{1}{2} \text{vol}(\mathcal{S}).$$

To find the desired ball $\mathcal{S}$, we consider the problem of finding the largest ball inscribed within $\mathcal{K} \cap \mathcal{B}_{x^\star}(\epsilon)$. This is a Chebyshev centering problem, and can be formulated as the following convex optimization problem.

$$\max_{r,y} \quad r \tag{71a}$$

$$\text{subject to} \quad Ay \leq b - r\mathbf{1} \tag{71b}$$

$$\|x^* - y\| + r \leq \epsilon \tag{71c}$$

where $r$ and $y$ denotes the radius and the center of the Chebyshev ball respectively. The particular form arises because the rows of $A$ are unit vectors, and so the ball of radius $r$ around $y$ is inscribed in $\mathcal{K}$ if and only if (71b) holds, while this ball is contained in $\mathcal{B}_{x^\star}(\epsilon)$ if and only if (71c) holds.

We rewrite this optimization problem as:

$$\min_{r,y} \quad -r + I_S\left(x^*, \begin{bmatrix} r \\ y \end{bmatrix}\right) \tag{72a}$$

$$\text{subject to} \quad Ay \leq b - r\mathbf{1} \tag{72b}$$

where $S = \{(x^*, \begin{bmatrix} r \\ y \end{bmatrix}) | \|x^* - y\| + r < \epsilon\}$.

Here, $I_S$ is defined by

$$I_S(x, \begin{bmatrix} r \\ y \end{bmatrix}) = \begin{cases} +\infty & \text{if } (x, \begin{bmatrix} r \\ y \end{bmatrix}) \notin S \\ 0 & \text{otherwise.} \end{cases} \tag{73}$$

Let $g(x^\star)$ denote the optimal value of (72). We will show that there is a positive constant $r_{\min} > 0$ such that $-g(x) \geq r_{\min}$ for all $x \in \mathcal{K}$. As a result, for any $x^*$ the corresponding Chebyshev centering solutions has radius at least $r_{\min}$.

Let $F(x^*, \begin{bmatrix} r \\ y \end{bmatrix}) = -r + I_s(x^*, \begin{bmatrix} r \\ y \end{bmatrix})$. We can see that $F$ is convex in $(x^*, \begin{bmatrix} r \\ y \end{bmatrix})$ and dom $F = S$.

Let $C = \{\begin{bmatrix} r \\ y \end{bmatrix} | [\mathbf{1} \ A]\begin{bmatrix} r \\ y \end{bmatrix} \leq b\}$. Then the optimal value of (72) can be expressed as $g(x) = \inf_{\begin{bmatrix} r \\ y \end{bmatrix} \in C} F(x, \begin{bmatrix} r \\ y \end{bmatrix})$ and dom $g = \{x | \exists \begin{bmatrix} r \\ y \end{bmatrix} \in C \text{ s.t. } (x, \begin{bmatrix} r \\ y \end{bmatrix}) \in S\}$.

The results of Section 3.2.5 of [6] imply that if $F$ is convex, $S$ is convex, and $g(x) > -\infty$ for all $x$, then $g$ is also convex.

If $(x, \begin{bmatrix} r \\ y \end{bmatrix}) \in$ dom $F$, then

$$\|x - y\| + r \leq \epsilon \implies r \leq \epsilon - \|x - y\|$$
$$\implies -r \geq -\epsilon + \|x - y\| > -\infty.$$

In particular, if there exist $y, r$ such that $(x, \begin{bmatrix} r \\ y \end{bmatrix}) \in$ dom $F$, then $\inf_{\begin{bmatrix} r \\ y \end{bmatrix} \in C} F(x, \begin{bmatrix} r \\ y \end{bmatrix}) \geq -\epsilon$.

There are two cases:

- If there exists $\begin{bmatrix} r \\ y \end{bmatrix} \in C$ such that $(x, \begin{bmatrix} r \\ y \end{bmatrix}) \in$ dom $F$, then $\inf_{\begin{bmatrix} r \\ y \end{bmatrix} \in C} F(x, \begin{bmatrix} r \\ y \end{bmatrix})$ is finite and bounded below.

- If there does not exist $\begin{bmatrix} r \\ y \end{bmatrix} \in C$ such that $(x, \begin{bmatrix} r \\ y \end{bmatrix}) \in$ dom$F$, then for all $\begin{bmatrix} r \\ y \end{bmatrix} \in C$, $F(x, \begin{bmatrix} r \\ y \end{bmatrix}) = +\infty$. So $g(x) = \inf_{\begin{bmatrix} r \\ y \end{bmatrix} \in C} F(x, \begin{bmatrix} r \\ y \end{bmatrix}) = +\infty > -\infty$.

Hereby, we can conclude that for all $x$, $g(x) > -\infty$, so $g(x)$ is convex.

So, to found a lower bound on the inscribed radius, we want to maximize $g(x)$ over $\mathcal{K}$. Specifically, we analyze the following optimization problem

$$\max_{x \in \mathcal{K}} \quad g(x) \tag{74a}$$

which corresponds to maximizing a convex function over a convex set.

Note that $\mathcal{K} \subset \text{dom}(g)$. In particular, if $x \in \mathcal{K}$, then $(x, \begin{bmatrix} 0 \\ x \end{bmatrix}) \in S$, which implies that $g(x) \leq 0$. Thus, $g(x) \leq 0$ for all $x \in \mathcal{K}$. Therefore, using Theorem 32.2 [38], given $\mathcal{K}$ is closed convex by our assumption and $g(x)$ is bounded above gives

$$\sup\{g(x) | x \in \mathcal{K}\} = \sup\{g(x) | x \in E\}$$

where $E$ is a subset of $\mathcal{K}$ consisting of the extreme points of $\mathcal{K} \cap L^\perp$, where $L$ is the linearity space of $C$ and $L = \{x | Ax = 0\} = \mathcal{N}(A)$.

Now, we will show that $E$ is a finite set.

Let

$$A = [U_1 \quad U_2] \begin{bmatrix} \Sigma & 0 \\ 0 & 0 \end{bmatrix} \begin{bmatrix} V_1^\top \\ V_2^\top \end{bmatrix}.$$

Then $\mathcal{N}(A) = L = \mathcal{R}(V_2)$ and $L^\perp = \mathcal{R}(V_1)$, and

$$K \cap L^\perp = \{V_1 Z_1 | AV_1 Z_1 \leq b\}.$$

This is a polyhedral with no lines so has a finite set of extreme points, i.e. $E$ is finite. In particular, they are contained in a compact subset of $\mathcal{K}$. Then it is shown in the proof of Proposition 16 of [27] that the Chebyshev centering problem has a positive global lower bound, when restricted to a compact convex set with 0 in its interior. Denote this value by $r_{\min}$.

Thus, we have that $\text{vol}(\mathcal{S}) \geq \frac{\pi^{n/2}}{\Gamma(n/2+1)} r_{\min}^n$, using the fact that a ball of radius $\rho$ has volumn given by $\frac{\pi^{n/2}}{\Gamma(n/2+1)} \rho^n$

Then, utilizing an upper bound of Gamma function recorded in [36] shown as below:

$$\Gamma(x+1) < \sqrt{\pi} \left(\frac{x}{e}\right)^x \left(8x^3 + 4x^2 + x + \frac{1}{30}\right)^{1/6}, \quad x \geq 0. \tag{75}$$

Setting $x = \frac{n}{2}$ in (75) gives:

$$\Gamma(\frac{n}{2} + 1) < \sqrt{\pi} \left(\frac{n}{2e}\right)^{\frac{n}{2}} \left(n^3 + n^2 + \frac{n}{2} + \frac{1}{30}\right)^{1/6}. \tag{76}$$

Therefore, we can find the lower bound of $\log \frac{1}{2}\text{vol}(S)$:

$$\log \frac{1}{2}\text{vol}(S) = \log \frac{\pi^{n/2}}{\Gamma(n/2+1)} r_{\min}^n - \log 2$$

$$> \frac{n}{2} \log \pi + n \log r_{\min} - \log \left\{ \sqrt{\pi} \left(\frac{n}{2e}\right)^{\frac{n}{2}} \left(n^3 + n^2 + \frac{n}{2} + \frac{1}{30}\right)^{1/6} \right\} - \log 2$$

$$= -\frac{1}{2} \log \pi + n \log r_{\min} + \frac{n}{2} \log(2\pi e) - \frac{n}{2} \log n - \frac{1}{6} \log \left(n^3 + n^2 + \frac{n}{2} + \frac{1}{30}\right) - \log 2$$

$$> n \log r_{\min} + \frac{n}{2} \log(2\pi e) - \frac{n}{2} \log n - \frac{1}{6} \log \left(3n^3\right) - \log(2\sqrt{\pi}) \tag{77}$$

The last inequality holds because $n \geq 1$.

Plugging (77) and (68) in (67) gives

$$\mathbb{E}_{\pi_{\beta \bar{f}}}[\bar{f}(x)] < \min f(x) + \frac{n}{2\beta} \log(2\pi e(\varsigma + \frac{1}{\mu} c_{10}))$$

$$- \frac{1}{\beta} \left( n \log r_{\min} + \frac{n}{2} \log(2\pi e) - \frac{n}{2} \log n - \frac{1}{2} \log n - \frac{1}{6} \log 3 - \log 2\sqrt{\pi} \right)$$

$$= \min f(x) + \frac{n}{2\beta} \log(\varsigma + \frac{1}{\mu} c_{10}) - \frac{1}{\beta} \left( n \log r_{\min} - \frac{n}{2} \log n - \frac{1}{2} \log n \right)$$

$$+ \frac{1}{\beta}(\frac{1}{6} \log 3 + \log 2\sqrt{\pi})$$

$$\leq \min f(x) + \frac{n}{\beta} \left( 2 \log(\varsigma + \frac{1}{\mu} c_{10}) + \frac{1}{6} \log 3 + \log 2\sqrt{\pi} - \log r_{\min} + \log n \right).$$

where last inequality holds because $n \geq 1$.

The final form of the bound holds because for any $c > 0$

$$\log\left(\varsigma + c\right) \leq \log\left(\max\{\varsigma, 1\} + c\right)$$

$$= \log\max\{\varsigma, 1\} + \log\left(1 + \frac{c}{\max\{\varsigma, 1\}}\right)$$

$$\leq \max\{\log\varsigma, 0\} + \log\left(1 + c\right).$$

∎

Now we cover the case of compact sets for comparison with [27].

**Proposition 21.** *Assume that $\mathcal{K}$ has diameter $D$ and $0 \in \mathcal{K}$ and let $c_{14} = \ell D + \|\nabla\bar{f}(0)\|$. Then for all $k \geq 0$, the iterates of the algorithm satisfy*

$$\mathbb{E}[\bar{f}(\mathbf{x}_k^A)] \leq \min_{x \in \mathcal{K}} f(x) + c_{14}W(\mathcal{L}(\mathbf{x}_k^A), \pi_{\beta\bar{f}}) + \frac{n}{\beta}\left(\max\{\log\varsigma, 0\} + c_{12}\right) \tag{78}$$

*In particular, there are constants $c_{15}$ and $c_{16}$ such that, for all sufficiently small $\epsilon$, if*

$$\beta = \frac{2n(\max\{\log\varsigma, 0\} + c_{15})}{\epsilon} \tag{79a}$$

$$T = e^{c_{16}/\epsilon} \tag{79b}$$

*then*

$$\mathbb{E}[\bar{f}(\mathbf{x}_T^A)] \leq \min_{x \in \mathcal{K}} \bar{f}(x) + \epsilon. \tag{80}$$

**Proof of Proposition 21** First, we show that $\bar{f}(x)$ is Lipschitz with Lipschitz constant $c_{14}$. Indeed,

$$\|\nabla\bar{f}(x)\| \leq \|\nabla\bar{f}(x) - \nabla\bar{f}(0)\| + \|\nabla\bar{f}(0)\| \leq \ell D + \|\nabla\bar{f}(0)\|.$$

So, if $x$ and $y$ are in $\mathcal{K}$, we have

$$|\bar{f}(x) - \bar{f}(y)| = \left|\int_0^1 \nabla\bar{f}(y + t(x - y))^\top (x - y)dt\right|$$

$$\leq c_1\|x - y\|.$$

Then (78) follows by Kantorovich duality combined with Lemma 20.

Now, using our bound from Theorem 1 gives that for $T \geq 4$:

$$\mathbb{E}[\bar{f}(\mathbf{x}_T^A)] \leq \min_{x \in \mathcal{K}} \bar{f}(x) + \frac{n}{\beta}\left(\max\{\log\varsigma, 0\} + c_{12}\right) + c_{14}\left(c_1 + c_2\sqrt{\varsigma} + \frac{c_3 + c_4\sqrt{\varsigma}}{(2a)^{1/2}}\right)T^{-1/2}\log T$$

Now, note that $c_{12}$ is monotonically decreasing in $\beta$. In particular, for $\beta \geq 1$

$$c_{10} = (\ell + \mu)R^2 + R\|\nabla_x\bar{f}(0)\| + \frac{n}{\beta} \leq (\ell + \mu)R^2 + R\|\nabla_x\bar{f}(0)\| + n,$$

so that

$$c_{12} = \log n + 2\log(1 + \frac{1}{\mu}c_{10}) + \frac{1}{6}\log 3 + \log 2\sqrt{\pi} - \log r_{\min}$$

$$\leq \log n + 2\log\left(1 + \frac{(\ell + \mu)R^2 + R\|\nabla_x\bar{f}(0)\| + n}{\mu}\right) + \frac{1}{6}\log 3 + \log 2\sqrt{\pi} - \log r_{\min}$$

$$=: c_{15}$$

It follows that for $\beta \geq 1$ we have the bound

$$\mathbb{E}[\bar{f}(\mathbf{x}_T^A)] \leq \min_{x \in \mathcal{K}} \bar{f}(x) + \frac{n}{\beta}\left(\max\{\log\varsigma, 0\} + c_{15}\right) + c_{14}\left(c_1 + c_2\sqrt{\varsigma} + \frac{c_3 + c_4\sqrt{\varsigma}}{(2a)^{1/2}}\right)T^{-1/2}\log T$$

Now, picking $\beta$ as in (79) gives

$$\frac{n}{\beta}\left(\max\{\log\varsigma, 0\} + c_{15}\right) = \epsilon/2.$$

Proposition 19 implies that there is some constant, $c$ (independent of $\eta$ and $\beta$) such that $c_1, c_2, c_3, c_4 \leq ce^{\frac{\beta\ell R^2}{2}}$. Furthermore, for all $\beta$ sufficiently large, we have from (65) that

$$\frac{1}{\sqrt{a}} \leq e^{\frac{\beta\ell R^2}{4}}. \tag{81}$$

Thus, for all $\beta$ sufficiently large we have that

$$c_{14}\left(c_1 + c_2\sqrt{\varsigma} + \frac{c_3 + c_4\sqrt{\varsigma}}{(2a)^{1/2}}\right) \leq e^{\beta\ell R^2}.$$

Thus, for our choice of $\beta$ (which is large for sufficiently small $\epsilon$), we have that

$$\mathbb{E}[\bar{f}(\mathbf{x}_T^A)] \leq \min_{x\in\mathcal{K}} \bar{f}(x) + \frac{\epsilon}{2} + e^{\beta\ell R^2}T^{-1/2}\log T$$

For simple notation, let $\alpha$ be such that

$$\beta\ell R^2 = \frac{\alpha}{\epsilon}.$$

In this case, $\alpha = 2n\left(\max\{\log\varsigma, 0\} + c_{15}\right)\ell R^2$.

We will choose $T = e^{\gamma/\epsilon}$ and choose $\gamma$ to ensure that

$$e^{\beta\ell R^2}T^{-1/2}\log T = \exp\left(\frac{1}{\epsilon}\left(\alpha - \frac{\gamma}{2}\right)\right)\frac{\gamma}{\epsilon} \leq \frac{\epsilon}{2}.$$

The desired inequality holds if and only if:

$$\exp\left(\frac{1}{\epsilon}\left(\alpha - \frac{\gamma}{2}\right)\right)\frac{2\gamma}{\epsilon^2} \leq 1$$

Note that if $\gamma/2 > \alpha$, then the left side is maximized over $(0, \infty)$ at $\epsilon = \frac{\frac{\gamma}{2} - \alpha}{2}$. Thus, a sufficient condition for this inequality to hold is:

$$\frac{8\gamma e^{-2}}{\left(\frac{\gamma}{2} - \alpha\right)^2} \leq 1.$$

A clean sufficient condition is $T = e^{c_{16}/\epsilon}$, where

$$c_{16} := \gamma = 4\alpha + 32e^{-2} = 8n\left(\max\{\log\varsigma, 0\} + c_{15}\right)\ell R^2 + 32e^{-2}.$$

∎

Now we extend the analysis to the non-compact case.

**Proposition 22.** *Let $\mathbf{x}_k$ be the iterates of the algorithms and assume $\eta \leq \frac{\mu}{3\ell^2}$ and $\mathbb{E}[\|\mathbf{x}_0\|^{2q}] < \infty$ for all $q > 1$. For all $q > 1$, there exist positive constants $c_{18}, c_{19}$ such that for all integers $k \geq 0$, the following bound holds:*

$$\mathbb{E}[\bar{f}(\mathbf{x}_k)] \leq \min_{x\in\mathcal{K}} \bar{f}(x) + c_{18}W_1(\mathcal{L}(\mathbf{x}_k), \pi_{\beta\bar{f}}) + c_{19}W_1(\mathcal{L}(\mathbf{x}_k), \pi_{\beta\bar{f}})^{\frac{2-2q}{1-2q}} + \frac{n}{\beta}\left(\max\{\log\varsigma, 0\} + c_{12}\right) \tag{82}$$

*where*

$$c_{18} = \|\nabla\bar{f}(0)\| + \ell\frac{2\left(\|\nabla\bar{f}(0)\| + \sqrt{\|\nabla\bar{f}(0)\|} + \sqrt{\mu\bar{f}(0)}\right)}{\mu}$$

$$c_{19} = \left(\frac{2\ell}{\sqrt{\mu}} + \left(\mathbb{E}[\|\mathbf{x}_0\|^{2q}] + c_{22}\right)\frac{\ell^q 2^{q-1}}{(q-1)}\right)\left(\frac{\ell}{\sqrt{\mu}(q-1)\left(\mathbb{E}[\|\mathbf{x}_0\|^{2q}] + c_{22}\right)\frac{\ell^q 2^{q-1}}{(q-1)}}\right)^{\frac{2-2q}{-2q+1}}$$

*and $c_{22}$ depends on $q$, the statistics of $\mathbf{z}$, the parameters $\mu$, $\ell$ and $\nabla\bar{f}(0)$ and decreases monotonically with respect to $\beta$.*

*Furthermore, there is a constant $c_{20}$ such if $\epsilon$ is sufficiently small, $\beta$ is chosen as in (79), and $T = e^{c_{20}/\epsilon}$, then*

$$\mathbb{E}[\bar{f}(\mathbf{x}_T)] \leq \min_{x\in\mathcal{K}} \bar{f}(x) + \epsilon.$$

**Proof of Proposition 22** Let $\mathbf{x}$ be drawn according to $\pi_{\beta\bar{f}}$. Then Lemma 20 implies:

$$\mathbb{E}[\bar{f}(\mathbf{x}_k)] = \mathbb{E}[\bar{f}(\mathbf{x})] + \mathbb{E}[\bar{f}(\mathbf{x}_k) - \bar{f}(\mathbf{x})]$$

$$\leq \min_{x \in \mathcal{K}} \bar{f}(x) + \mathbb{E}[\bar{f}(\mathbf{x}_k) - \bar{f}(\mathbf{x})] + \frac{n}{\beta}\left(\max\{\log\varsigma, 0\} + c_{12}\right) \tag{83}$$

So, it now suffices to bound $\mathbb{E}[\bar{f}(\mathbf{x}_k) - \bar{f}(\mathbf{x})]$. Ideally, we would bound this term via Kantorovich duality. The problem is that $\bar{f}$ may not be globally Lipschitz. So, we must approximate it with a Lipschitz function, and then bound the gap induced by this approximation.

Namely, fix a constant $m > \bar{f}(0)$ with $m$ to be chosen later. Set $g(x) = \min\{\bar{f}(x), m\}$. The inequality from (25) implies that if $\|x\| \geq \hat{R} := \frac{2\left(\|\nabla\bar{f}(0)\| + \sqrt{\|\nabla\bar{f}(0)\| + \mu(m - \bar{f}(0))}\right)}{\mu}$, then $\bar{f}(x) \geq m$. We claim that $g$ is globally Lipschitz.

For $\|x\| \leq \hat{R}$, we have that
$$\|\nabla\bar{f}(x)\| \leq \|\nabla\bar{f}(0)\| + \ell\hat{R} =: u.$$
We will show that $g$ is $u$-Lipschitz.

In the case that $f(y) \geq m$ and $f(x) \geq m$, we have $|g(x) - g(y)| = 0$, so the property holds.

Now say that $\bar{f}(x) < m$ and $\bar{f}(y) < m$. Then we must have $\|x\| \leq \hat{R}$ and $\|y\| \leq \hat{R}$. Then for all $t \in [0, 1]$, we have $\|(1-t)x + ty\| \leq \hat{R}$. It follows that

$$g(x) - g(y) = \bar{f}(x) - \bar{f}(y)$$

$$= \int_0^1 \nabla\bar{f}(x + t(y - x))^\top (y - x)dt$$

$$\leq u\|x - y\|.$$

Finally, consider the case that $\bar{f}(x) \geq m$ and $\bar{f}(y) < m$. Then there is some $\theta \in [0, 1]$ such that $\bar{f}(y + \theta(x - y)) = m$. Furthermore

$$|g(x) - g(y)| = m - \bar{f}(y)$$

$$= \bar{f}(y + \theta(x - y)) - \bar{f}(y)$$

$$= \int_0^\theta \nabla\bar{f}(y + t(x - y))^\top (x - y)dt$$

$$\leq u\|x - y\|.$$

It follows that $g$ is $u$-Lipschitz.

Now noting that $g(x) \leq \bar{f}(x)$ for all $x$ gives

$$\mathbb{E}[\bar{f}(\mathbf{x}_k) - \bar{f}(\mathbf{x})] \leq \mathbb{E}[\bar{f}(\mathbf{x}_k) - g(\mathbf{x})]$$

$$= \mathbb{E}[g(\mathbf{x}_k) - g(\mathbf{x})] + \mathbb{E}[\mathbb{1}(\bar{f}(\mathbf{x}_k) > m)(\bar{f}(\mathbf{x}_k) - m)]$$

$$\leq uW_1(\mathcal{L}(\mathbf{x}_k), \pi_{\beta\bar{f}}) + \mathbb{E}[\mathbb{1}(\bar{f}(\mathbf{x}_k) > m)(\bar{f}(\mathbf{x}_k) - m)]. \tag{84}$$

The final inequality uses Kantorovich duality. Now, it remains to bound $\mathbb{E}[\mathbb{1}(\bar{f}(\mathbf{x}_k) > m)(\bar{f}(\mathbf{x}_k) - m)]$.

Note that if $\mathbf{y}$ is a non-negative random variable, a standard identity gives that $\mathbb{E}[\mathbf{y}] = \int_0^\infty \mathbb{P}(\mathbf{y} > \epsilon)d\epsilon$. Thus, we have

$$\mathbb{E}[\mathbb{1}(\bar{f}(\mathbf{x}_k) > m)(\bar{f}(\mathbf{x}_k) - m)] = \int_0^\infty \mathbb{P}(\bar{f}(\mathbf{x}_k) - m > \epsilon)d\epsilon.$$

For all $x \in \mathcal{K}$, we have

$$\bar{f}(x) = \bar{f}(0) - \nabla\bar{f}(0)^\top x + \int_0^1 (\nabla\bar{f}(tx) - \nabla\bar{f}(0))^\top x dt$$

$$\leq \bar{f}(0) + \|\nabla\bar{f}(0)\|\|x\| + \frac{1}{2}\ell\|x\|^2$$

$$\leq \bar{f}(0) + \frac{\|\nabla\bar{f}(0)\|^2}{2\ell} + \ell\|x\|^2.$$

So,

$$\bar{f}(x) - m > \epsilon \implies \bar{f}(0) + \frac{\|\nabla \bar{f}(0)\|^2}{2\ell} + \ell\|x\|^2 > m + \epsilon$$

$$\iff \|x\|^2 > \frac{m + \epsilon - \left(\bar{f}(0) + \frac{\|\nabla \bar{f}(0)\|^2}{2\ell}\right)}{\ell}.$$

Now assume that $m/2 > \bar{f}(0) + \frac{\|\nabla \bar{f}(0)\|^2}{2\ell}$. Then the right side implies $\|x\|^2 \geq \frac{\frac{m}{2}+\epsilon}{\ell}$. It follows that for any $q > 1$, we have, via Markov's inequality and direct computation:

$$\mathbb{E}[\mathbb{1}(\bar{f}(\mathbf{x}_k) > m)(\bar{f}(\mathbf{x}_k) - m)] \leq \int_0^\infty \mathbb{P}\left(\|\mathbf{x}_k\|^2 > \frac{\frac{m}{2}+\epsilon}{\ell}\right) d\epsilon$$

$$= \int_0^\infty \mathbb{P}\left(\|\mathbf{x}_k\|^{2q} > \left(\frac{\frac{m}{2}+\epsilon}{\ell}\right)^q\right) d\epsilon$$

$$\leq \mathbb{E}[\|\mathbf{x}_k\|^{2q}]\int_0^\infty \left(\frac{\frac{m}{2}+\epsilon}{\ell}\right)^{-q} d\epsilon$$

$$= \mathbb{E}[\|\mathbf{x}_k\|^{2q}]\frac{\ell^q 2^{q-1}}{(q-1)m^{q-1}}.$$

Plugging this expression into (84) and using the definition of $u$ gives

$$\mathbb{E}[\bar{f}(\mathbf{x}_k) - \bar{f}(\mathbf{x})]$$

$$\leq \left(\|\nabla \bar{f}(0)\| + \ell\frac{2\left(\|\nabla \bar{f}(0)\| + \sqrt{\|\nabla \bar{f}(0)\| + \mu(m - \bar{f}(0))}\right)}{\mu}\right) W_1(\mathcal{L}(\mathbf{x}_k), \pi_{\beta\bar{f}})$$

$$+ \mathbb{E}[\|\mathbf{x}_k\|^{2q}]\frac{\ell^q 2^{q-1}}{(q-1)m^{q-1}}. \tag{85}$$

We want to derive the bound of $\mathbb{E}[\|\mathbf{x}_k\|^{2q}]$.

We have

$$\|\mathbf{x}_{k+1}\|^{2q} \leq \|\mathbf{x}_k - \eta\nabla f(\mathbf{x}_k, \mathbf{z}_k) + \sqrt{\frac{2\eta}{\beta}}\hat{\mathbf{w}}_k\|^2.$$

For notational simplicity, let $\mathbf{y} = \frac{\mathbf{x}_k - \eta\nabla f(\mathbf{x}_k, \mathbf{z}_k)}{\sqrt{2\eta/\beta}}$ and $\mathbf{w} = \hat{\mathbf{w}}_k$, then the above inequality can be expressed as

$$\|\mathbf{x}_{k+1}\|^{2q} \leq \left(\frac{2\eta}{\beta}\right)^q \|\mathbf{y} + \mathbf{w}\|^{2q}$$

$$= \left(\frac{2\eta}{\beta}\right)^q \left(\|\mathbf{y}\|^2 + \|\mathbf{w}\|^2 + 2\mathbf{y}^\top\mathbf{w}\right)^q$$

$$= \left(\frac{2\eta}{\beta}\right)^q \sum_{k=0}^q \binom{q}{k}(2\mathbf{y}^\top\mathbf{w})^{q-k}\left(\|\mathbf{y}\|^2 + \|\mathbf{w}\|^2\right)^k$$

$$= \left(\frac{2\eta}{\beta}\right)^q \sum_{k=0}^q \binom{q}{k}(2\mathbf{y}^\top\mathbf{w})^{q-k}\sum_{i=0}^k \binom{k}{i}\left(\|\mathbf{y}\|^{2i}\|\mathbf{w}\|^{2(k-i)}\right). \tag{86}$$

The last two equalities use the binomial theorem. Here, we construct an orthogonal matrix $U = \begin{bmatrix} \frac{1}{\|\mathbf{y}\|}\mathbf{y}^\top\mathbf{w} \\ s \end{bmatrix}$ such that we can linearly transform the Gaussian noise $\mathbf{w}$ into $\mathbf{v} = U\mathbf{w} = \begin{bmatrix} \mathbf{v}_1 \\ \mathbf{v}_2 \end{bmatrix}$, where $\mathbf{v}_1 = \frac{1}{\|\mathbf{y}\|}\mathbf{y}^\top\mathbf{w}$ and $\mathbf{v}_2 = s\mathbf{w}$. And the orthogonality of the matrix $U$ gives $\mathbf{v}_1 \perp \mathbf{v}_2$ and thus $\mathbf{v}_1^2 + \mathbf{v}_2^\top\mathbf{v}_2$ follows a chi-squared distribution with $n$ degrees of freedom. Furthermore, we have $\|\mathbf{w}\|^2 = \mathbf{v}_1^2 + \mathbf{v}_2^\top\mathbf{v}_2$ and $\mathbf{y}^\top\mathbf{w} = \|\mathbf{y}\|\mathbf{v}_1$.

Therefore, with the change of variables, (86) can be expressed as

$$\|\mathbf{x}_{k+1}\|^{2q} \leq \left(\frac{2\eta}{\beta}\right)^q \sum_{k=0}^{q} \binom{q}{k} (2\|\mathbf{y}\|\mathbf{v}_1)^{q-k} \sum_{i=0}^{k} \binom{k}{i} \left(\|\mathbf{y}\|^{2i}(\mathbf{v}_1^2 + \mathbf{v}_2^\top \mathbf{v}_2)^{(k-i)}\right).$$

Taking the expectation of the above inequality gives

$$\mathbb{E}[\|\mathbf{x}_{k+1}\|^{2q}] \leq \left(\frac{2\eta}{\beta}\right)^q \mathbb{E}\left[\sum_{k=0}^{q} \binom{q}{k}(2\|\mathbf{y}\|\mathbf{v}_1)^{q-k}\sum_{i=0}^{k}\binom{k}{i}\left(\|\mathbf{y}\|^{2i}(\mathbf{v}_1^2 + \mathbf{v}_2^\top \mathbf{v}_2)^{(k-i)}\right)\right] \quad (87)$$

$$\leq \mathbb{E}\left[\|\mathbf{x}_k - \eta\nabla f(\mathbf{x}_k, \mathbf{z}_k)\|^{2q}\right] + \eta\mathbb{E}\left[p(\|\mathbf{x}_k - \eta\nabla f(\mathbf{x}_k, \mathbf{z}_k)\|^2)\right] \quad (88)$$

where $p(\|\mathbf{x}_k - \eta\nabla f(\mathbf{x}_k, \mathbf{z}_k)\|^2)$ is a polynomial in $\|\mathbf{x}_k - \eta\nabla f(\mathbf{x}_k, \mathbf{z}_k)\|^2$ with order strictly lower than $q$ and the coefficients of $\mathbb{E}[p(\|\mathbf{x}_k - \eta\nabla f(\mathbf{x}_k, \mathbf{z}_k)\|^2)]$ depend on the moments of the chi-squared distributions and $q$. (Additionally, note that the coefficients of $p$ can be taken to be monotonically decreasing with respect to $\beta$.) And the reason the polynomial only have even order terms in $\|\mathbf{x}_k - \eta\nabla f(\mathbf{x}_k, \mathbf{z}_k)\|$ is that in (87), when $q - k$ is odd, the expectation is zero since $\mathbf{v}_1 \sim \mathcal{N}(0, 1)$ whose odd order moments are all zero.

Then we firstly aim to bound $\mathbb{E}[\|\mathbf{x}_k - \eta\nabla\bar{f}(\mathbf{x}_k)\|^{2q}]$.

We have

$$\|\mathbf{x}_k - \eta\nabla f(\mathbf{x}_k, \mathbf{z}_k)\|^{2q} = \left(\|\mathbf{x}_k\|^2 - 2\eta\mathbf{x}_k^\top\nabla f(\mathbf{x}_k, \mathbf{z}_k) + \eta^2\|\nabla f(\mathbf{x}_k, \mathbf{z}_k)\|^2\right)^q. \quad (89)$$

We examine the second term:

$$\mathbf{x}_k^\top\nabla f(\mathbf{x}_k, \mathbf{z}_k) = \mathbf{x}_k^\top(\nabla\bar{f}(\mathbf{x}_k) - \nabla\bar{f}(0)) + \mathbf{x}_k^\top(\nabla\bar{f}(0) - \nabla\bar{f}(\mathbf{x}_k) + \nabla f(\mathbf{x}_k, \mathbf{z}_k))$$

$$\geq \mu\|\mathbf{x}_k\|^2 - (\ell + \mu)R^2 + \mathbf{x}_k^\top\left(\nabla\bar{f}(0) + \mathbb{E}_{\hat{\mathbf{z}}}[\nabla f(\mathbf{x}_k, \mathbf{z}_k) - \nabla f(\mathbf{x}_k, \hat{\mathbf{z}}_k)]\right)$$

where the first term is bounded by the assumption of the strong convexity outside a ball and the detailed statement is shown below:

If $\|x\| \geq R$, then $x^\top(\nabla\bar{f}(x) - \nabla\bar{f}(0)) \geq \mu\|x\|^2$.

If $\|x\| \leq R$, then $x^\top(\nabla\bar{f}(x) - \nabla\bar{f}(0)) \geq -\ell\|x\|^2 \geq -\ell R^2$.

Therefore, we have for all $x \in \mathcal{K}$, $x^\top(\nabla\bar{f}(x) - \nabla\bar{f}(0)) \geq \mu\|x\|^2 - (\ell + \mu)R^2$.

Note here and below $\hat{\mathbf{z}}$ and $\mathbf{z}$ are IID.

Taking expectation of (89) gives

$$\mathbb{E}\left[\|\mathbf{x}_k - \eta\nabla f(\mathbf{x}_k, \mathbf{z}_k)\|^{2q}\right]$$

$$= \mathbb{E}\left[\left(\|\mathbf{x}_k\|^2 - 2\eta\mathbf{x}_k^\top\nabla f(\mathbf{x}_k, \mathbf{z}_k) + \eta^2\|\nabla f(\mathbf{x}_k, \mathbf{z}_k)\|^2\right)^q\right]$$

$$\leq \mathbb{E}\left[\left((1 - 2\mu\eta)\|\mathbf{x}_k\|^2 + 2\eta(\ell + \mu)R^2\right.\right.$$

$$\left.\left. - 2\eta\mathbf{x}_k^\top(\nabla\bar{f}(0) + \mathbb{E}_{\hat{\mathbf{z}}}[\nabla f(\mathbf{x}_k, \mathbf{z}_k) - \nabla f(\mathbf{x}_k, \hat{\mathbf{z}}_k)]) + \eta^2\|\nabla f(\mathbf{x}_k, \mathbf{z}_k)\|^2\right)^q\right]$$

$$\leq \mathbb{E}\left[\left((1 - 2\mu\eta)\|\mathbf{x}_k\|^2 + 2\eta(\ell + \mu)R^2\right.\right.$$

$$\left.\left. - 2\eta\mathbf{x}_k^\top(\nabla\bar{f}(0) + \nabla f(\mathbf{x}_k, \mathbf{z}_k) - \nabla f(\mathbf{x}_k, \hat{\mathbf{z}}_k)) + \eta^2\|\nabla f(\mathbf{x}_k, \mathbf{z}_k)\|^2\right)^q\right]$$

$$\leq \mathbb{E}\left[\left((1 - 2\mu\eta)\|\mathbf{x}_k\|^2 + 2\eta(\ell + \mu)R^2\right.\right.$$

$$\left.\left. + 2\eta\|\mathbf{x}_k\|(\|\nabla\bar{f}(0)\| + \ell\|\mathbf{z}_k - \hat{\mathbf{z}}_k\|) + \eta^2\|\nabla f(\mathbf{x}_k, \mathbf{z}_k)\|^2\right)^q\right]. \quad (90)$$

The second inequality uses Jensen's inequality, and the last inequality uses Cauchy-Schwartz inequality together with $\ell$-Lipschitzness of $\nabla f(x, z)$ in $z$.

Now we examine the last term of (90).

Firstly, we have

$$\|\nabla f(x, z)\| = \|\nabla f(x, z) - \mathbb{E}_{\hat{\mathbf{z}}}[\nabla f(x, \hat{\mathbf{z}})] + \mathbb{E}_{\hat{\mathbf{z}}}[\nabla f(x, \hat{\mathbf{z}})]\|$$

$$\leq \|\mathbb{E}_{\hat{\mathbf{z}}}[\nabla f(x, z) - \nabla f(x, \hat{\mathbf{z}})]\| + \|\bar{f}(x)\|$$

$$\leq \ell\mathbb{E}_{\hat{\mathbf{z}}}[\|z - \hat{\mathbf{z}}\|] + \|\nabla\bar{f}(0)\| + \ell\|x\|.$$

So

$$\|\nabla f(x, z)\|^2 \le 3 \left( \ell^2 \left( \mathbb{E}_{\hat{\mathbf{z}}}[\|z - \hat{\mathbf{z}}\|] \right)^2 + \|\nabla \bar{f}(0)\|^2 + \ell^2 \|x\|^2 \right). \tag{91}$$

Then, we can group the square terms in (90) together and simplify it:

$$(1 - 2\mu\eta)\|x\|^2 + \eta^2 3\ell^2 \|x\|^2 \le (1 - \eta\mu)\|x\|^2$$
$$\Longleftrightarrow 1 - 2\mu\eta + \eta^2 3\ell^2 \le 1 - \eta\mu$$
$$\Longleftrightarrow \eta \le \frac{\mu}{3\ell^2}.$$

So, if $\eta \le \frac{\mu}{3\ell^2}$, plugging (91) into (90) gives

$$\mathbb{E}\left[\|\mathbf{x}_k - \eta\nabla f(\mathbf{x}_k, \mathbf{z}_k)\|^{2q}\right] \le \mathbb{E}\left[\left((1 - \mu\eta)\|\mathbf{x}_k\|^2 + 2\eta(\ell + \mu)R^2\right.\right.$$
$$\left.\left. + 2\eta\|\mathbf{x}_k\|(\|\nabla\bar{f}(0)\| + \ell\|\mathbf{z}_k - \hat{\mathbf{z}}_k\|) + \eta^2 3\left(\ell^2\left(\mathbb{E}_{\hat{\mathbf{z}}}[\|\mathbf{z}_k - \hat{\mathbf{z}}_k\|]\right)^2 + \|\nabla\bar{f}(0)\|^2\right)\right)^q\right]. \tag{92}$$

We want to further group the first and third terms above together.

For all $\epsilon \ge 0$, $2ab = 2(\epsilon a)(\frac{1}{\epsilon}b) \le (\epsilon a)^2 + (\frac{1}{\epsilon}b)^2$. Let $a = \|\mathbf{x}_k\|$, $b = \|\nabla\bar{f}(0)\| + \ell\|\mathbf{z}_k - \hat{\mathbf{z}}_k\|$, then we can see the third term of the right side of (92) can be upper bounded by a summation of two parts. The first part can be grouped with the first term of the right side of (92):

$$(1 - \mu\eta)\|\mathbf{x}_k\|^2 + \eta\epsilon^2\|\mathbf{x}_k\|^2 \le (1 - \frac{\mu\eta}{2})\|\mathbf{x}_k\|^2$$
$$\Longleftrightarrow 1 - \mu\eta + \eta\epsilon^2 \le 1 - \frac{\mu\eta}{2}$$
$$\Longleftrightarrow \epsilon \le \sqrt{\frac{\mu}{2}}.$$

So let $\epsilon = \sqrt{\frac{\mu}{2}}$, we have

$$\mathbb{E}\left[\|\mathbf{x}_k - \eta\nabla f(\mathbf{x}_k, \mathbf{z}_k)\|^{2q}\right] \le \mathbb{E}\left[\left((1 - \frac{\mu\eta}{2})\|\mathbf{x}_k\|^2 + 2\eta(\ell + \mu)R^2\right.\right.$$
$$\left.\left. + \frac{2\eta}{\mu}(\|\nabla\bar{f}(0)\| + \ell\|\mathbf{z}_k - \hat{\mathbf{z}}_k\|)^2 + \eta^2 3\left(\ell^2\left(\mathbb{E}_{\hat{\mathbf{z}}}[\|\mathbf{z}_k - \hat{\mathbf{z}}_k\|]\right)^2 + \|\nabla\bar{f}(0)\|^2\right)\right)^q\right]$$
$$\le \mathbb{E}\left[\left((1 - \frac{\mu\eta}{2})\|\mathbf{x}_k\|^2 + 2\eta(\ell + \mu)R^2\right.\right.$$
$$\left.\left. + \frac{2\eta}{\mu}(\|\nabla\bar{f}(0)\| + \ell\|\mathbf{z}_k - \hat{\mathbf{z}}_k\|)^2 + \eta^2 3\left(\ell^2\left(\|\mathbf{z}_k - \hat{\mathbf{z}}_k\|\right)^2 + \|\nabla\bar{f}(0)\|^2\right)\right)^q\right] \tag{93}$$
$$= (1 - \frac{\mu\eta}{2})^q \mathbb{E}[\|\mathbf{x}_k\|^{2q}] + \eta\mathbb{E}[p_2(\|\mathbf{x}_k\|^2, \|\mathbf{z}_k - \hat{\mathbf{z}}_k\|)]$$
$$\le (1 - \frac{\mu\eta}{2})\mathbb{E}[\|\mathbf{x}_k\|^{2q}] + \eta\mathbb{E}[p_2(\|\mathbf{x}_k\|^2, \|\mathbf{z}_k - \hat{\mathbf{z}}_k\|)]. \tag{94}$$

The inequality (93) uses Jensen's inequality twice. The polynomial $p_2(\|\mathbf{x}_k\|^2, \|\mathbf{z}_k - \hat{\mathbf{z}}_k\|)$ is with order strictly lower than $q$ in $\|\mathbf{x}_k\|^2$ and with the highest order of $2q$ in $\|\mathbf{z}_k - \hat{\mathbf{z}}_k\|$.

Similarly, we can obtain for all $i < q$,

$$\mathbb{E}\left[\|\mathbf{x}_k - \eta\nabla f(\mathbf{x}_k, \mathbf{z}_k)\|^{2i}\right] \le \mathbb{E}\left[\left((1 - \frac{\mu\eta}{2})\|\mathbf{x}_k\|^2 + 2\eta(\ell + \mu)R^2\right.\right.$$
$$\left.\left. + \frac{2\eta}{\mu}(\|\nabla\bar{f}(0)\| + \ell\|\mathbf{z}_k - \hat{\mathbf{z}}_k\|)^2 + \eta^2 3\left(\ell^2\left(\|\mathbf{z}_k - \hat{\mathbf{z}}_k\|\right)^2 + \|\nabla\bar{f}(0)\|^2\right)\right)^i\right].$$

This implies that $\mathbb{E}[p(\|\mathbf{x}_k - \eta\nabla f(\mathbf{x}_k, \mathbf{z}_k)\|^2)]$ can be upper bounded by $\mathbb{E}[p_1(\|\mathbf{x}_k\|^2, \|\mathbf{z}_k - \hat{\mathbf{z}}_k\|)]$ where $p_1(\|\mathbf{x}_k\|^2, \|\mathbf{z}_k - \hat{\mathbf{z}}_k\|)$ is a polynomial with the order strictly lower than q in $\|\mathbf{x}_k\|^2$ and the highest order of $2q - 2$ in $\|\mathbf{z}_k - \hat{\mathbf{z}}_k\|$.

So (88) can be further upper bounded as below:

$$\mathbb{E}[\|\mathbf{x}_{k+1}\|^{2q}] \leq (1 - \frac{\mu\eta}{2})\mathbb{E}[\|\mathbf{x}_k\|^{2q}] + \eta\mathbb{E}[p_2(\|\mathbf{x}_k\|^2, \|\mathbf{z}_k - \hat{\mathbf{z}}_k\|)] + \eta\mathbb{E}[p_1(\|\mathbf{x}_k\|^2, \|\mathbf{z}_k - \hat{\mathbf{z}}_k\|)]$$

$$= (1 - \frac{\mu\eta}{2})\mathbb{E}[\|\mathbf{x}_k\|^{2q}] + \eta\mathbb{E}[p_3(\|\mathbf{x}_k\|^2, \|\mathbf{z}_k - \hat{\mathbf{z}}_k\|)]$$

$$\leq (1 - \frac{\mu\eta}{4})\mathbb{E}[\|\mathbf{x}_k\|^{2q}] + \eta\mathbb{E}[-\frac{\mu}{4}\|\mathbf{x}_k\|^{2q} + p_3(\|\mathbf{x}_k\|^2, \|\mathbf{z}_k - \hat{\mathbf{z}}_k\|)]$$

$$\leq (1 - \frac{\mu\eta}{4})\mathbb{E}[\|\mathbf{x}_k\|^{2q}] + \eta\mathbb{E}[\frac{\mu}{4}\left(-\|\mathbf{x}_k\|^{2q} + \tilde{p}(\|\mathbf{x}_k\|^2, \|\mathbf{z}_k - \hat{\mathbf{z}}_k\|)\right)] \qquad (95)$$

To get the upper bound of the second term of (95), we examine the following polynomial with $x \geq 0$

$$-x^q + \sum_{i=0}^{q-1} a_{q,i}x^i,$$

where the $a_{q,i}$'s depend on the value of $q$, the statistics of the external random variables $\mathbf{z}$ and some other parameters including $\ell$, $\mu$ and $\|\nabla\bar{f}(0)\|$ and $a_{q,i}$'s decrease monotonically with respect to $\beta$.

To find the upper bound of such a polynomial, we consider two cases

- Assume $0 \leq x \leq 1$, then $-x^q + \sum_{i=0}^{q-1} a_{q,i}x^i \leq \sum_{i=0}^{q-1} |a_{q,i}|$;
- Assume $x > 1$, then $-x^q + \sum_{i=0}^{q-1} a_{q,i}x^i \leq \left(\sum_{i=0}^{q-1} |a_{q,i}|\right)\left(\sum_{i=0}^{q-1} |a_{q,i}| + 1\right)^{q-1}$.

Combining the two cases gives that for all $x \geq 0$,

$$-x^q + \sum_{i=0}^{q-1} a_{q,i}x^i \leq \left(\sum_{i=0}^{q-1} |a_{q,i}|\right)\left(\sum_{i=0}^{q-1} |a_{q,i}| + 1\right)^{q-1}.$$

The first case is a direct result of dropping the negative term and using Cauchy-Schwartz inequality. The second case is obtained by firstly showing the sufficient condition of the polynomial being non-positive. The detail is shown below:

$$-x^q + \sum_{i=0}^{q-1} a_{q,i}x^i \leq 0 \iff -1 + \sum_{i=0}^{q-1} \frac{a_{q,i}}{x^{q-i}} \leq 0$$

$$\impliedby -1 + \sum_{i=0}^{q-1} \frac{|a_{q,i}|}{x} \leq 0 \qquad (96)$$

$$\iff -1 + \frac{1}{x}\sum_{i=0}^{q-1} |a_{q,i}| \leq 0$$

$$\iff x \geq \max\{\sum_{i=0}^{q-1} |a_{q,i}|, 1\} \qquad (97)$$

$$\impliedby x \geq \sum_{i=0}^{q-1} |a_{q,i}| + 1$$

Both (96) and (97) use the assumption that $x > 1$.

Besides, for $1 < x \leq \sum_{i=0}^{q-1} |a_{q,i}| + 1$,

$$-x^q + \sum_{i=0}^{q-1} a_{q,i}x^i \leq \sum_{i=0}^{q-1} |a_{q,i}|x^i$$

$$\leq \sum_{i=0}^{q-1} |a_{q,i}|x_{max}^{q-1}$$

$$= \sum_{i=0}^{q-1} |a_{q,i}|\left(\sum_{i=0}^{q-1} |a_{q,i}| + 1\right)^{q-1}.$$

Therefore, we can conclude that

$$\mathbb{E}[-\frac{\mu}{4}\|\mathbf{x}_k\|^{2q} + \tilde{p}(\|\mathbf{x}_k\|^2, \|\mathbf{z}_k - \hat{\mathbf{z}}_k\|)] \leq \mathbb{E}\left[\frac{\mu}{4}\sum_{i=0}^{q-1}|a_{q,i}|\left(\sum_{i=0}^{q-1}|a_{q,i}|+1\right)^{q-1}\right].$$

The L-mixing property ensures that the right side of the inequality is bounded. Then, we achieve the upper bound of equation (95).

$$\mathbb{E}[\|\mathbf{x}_{k+1}\|^{2q}] \leq (1-\frac{\mu\eta}{4})\mathbb{E}[\|\mathbf{x}_k\|^{2q}] + \eta\mathbb{E}\left[\frac{\mu}{4}\sum_{i=0}^{q-1}|a_{q,i}|\left(\sum_{i=0}^{q-1}|a_{q,i}|+1\right)^{q-1}\right]$$

Iterating the inequality above and letting $\tilde{a}_q = \mathbb{E}\left[\frac{\mu}{4}\sum_{i=0}^{q-1}|a_{q,i}|\left(\sum_{i=0}^{q-1}|a_{q,i}|+1\right)^{q-1}\right]$ give

$$\mathbb{E}[\|\mathbf{x}_k\|^q] \leq \left(1-\frac{\mu\eta}{4}\right)^k \mathbb{E}[\|\mathbf{x}_0\|^{2q}] + \eta\tilde{a}_q\sum_{i=0}^{k-1}(1-\frac{\mu\eta}{4})^i$$

$$\leq \mathbb{E}[\|\mathbf{x}_0\|^{2q}] + \eta\tilde{a}_q\frac{1-\left(1-\frac{\mu\eta}{4}\right)^k}{1-\left(1-\frac{\mu\eta}{4}\right)}$$

$$\leq \mathbb{E}[\|\mathbf{x}_0\|^{2q}] + \frac{4}{\mu}\tilde{a}_q\left(1-\left(1-\frac{\mu\eta}{4}\right)^k\right)$$

$$\leq \mathbb{E}[\|\mathbf{x}_0\|^{2q}] + \frac{4}{\mu}\tilde{a}_q.$$

Now as long as $\mathbb{E}[\|\mathbf{x}_0\|^{2q}] < \infty$ and $\eta < 1$, we have

$$\mathbb{E}[\|\mathbf{x}_k\|^{2q}] \leq \mathbb{E}[\|\mathbf{x}_0\|^{2q}] + c_{22},$$

where $c_{22} = \frac{4}{\mu}\tilde{a}_q$. More specifically, $c_{22}$ depends on $q$, the statistics of $\mathbf{z}$, the parameters $\mu$, $\ell$ and $\nabla\bar{f}(0)$.

Plugging the above result into (85) gives

$$\mathbb{E}[\bar{f}(\mathbf{x}_k) - \bar{f}(\mathbf{x})]$$
$$\leq \left(\|\nabla\bar{f}(0)\| + \ell\frac{2\left(\|\nabla\bar{f}(0)\| + \sqrt{\|\nabla\bar{f}(0)\| + \mu(m - \bar{f}(0))}\right)}{\mu}\right)W_1(\mathcal{L}(\mathbf{x}_k), \pi_{\beta\bar{f}})$$
$$+ \left(\mathbb{E}[\|\mathbf{x}_0\|^{2q}] + c_{22}\right)\frac{\ell^q 2^{q-1}}{(q-1)m^{q-1}}.$$

The remaining work is to optimize the right side of the above inequality with respect to $m$ so that we can make a choice of the value of $m$ mentioned earlier in the proof.

Let

$$g(m) = \left(\|\nabla\bar{f}(0)\| + \ell\frac{2\left(\|\nabla\bar{f}(0)\| + \sqrt{\|\nabla\bar{f}(0)\|} + \sqrt{\mu m} + \sqrt{\mu\bar{f}(0)}\right)}{\mu}\right)W_1(\mathcal{L}(\mathbf{x}_k), \pi_{\beta\bar{f}})$$
$$+ \left(\mathbb{E}[\|\mathbf{x}_0\|^{2q}] + c_{22}\right)\frac{\ell^q 2^{q-1}}{(q-1)m^{q-1}}.$$

We can see that $g(m)$ is an upper bound of the right side of (85).

Setting $g'(m) = 0$ leads to $m^* = \left(\frac{\ell W_1}{\sqrt{\mu}(q-1)C}\right)^{\frac{-2}{-2q+1}}$, where $C = \left(\mathbb{E}[\|\mathbf{x}_0\|^{2q}] + c_{22}\right)\frac{\ell^q 2^{q-1}}{(q-1)}$ for notation simplicity. So

$$
\max_{m \geq 0} g(m) = g(m^*)
$$

$$
\leq \left(\|\nabla \bar{f}(0)\| + \ell\frac{2\left(\|\nabla \tilde{f}(0)\| + \sqrt{\|\nabla \tilde{f}(0)\|} + \sqrt{\mu \tilde{f}(0)}\right)}{\mu}\right) W_1(\mathcal{L}(\mathbf{x}_k), \pi_{\beta \bar{f}})
$$

$$
+ \frac{2\ell}{\sqrt{\mu}}\left(\frac{\ell W_1(\mathcal{L}(\mathbf{x}_k), \pi_{\beta \bar{f}})}{\sqrt{\mu}(q-1)C}\right)^{\frac{2-2q}{-2q+1}} + C\left(\frac{\ell W_1(\mathcal{L}(\mathbf{x}_k), \pi_{\beta \bar{f}})}{\sqrt{\mu}(q-1)C}\right)^{\frac{2-2q}{-2q+1}}
$$

$$
= \left(\|\nabla \bar{f}(0)\| + \ell\frac{2\left(\|\nabla \tilde{f}(0)\| + \sqrt{\|\nabla \tilde{f}(0)\|} + \sqrt{\mu \tilde{f}(0)}\right)}{\mu}\right) W_1(\mathcal{L}(\mathbf{x}_k), \pi_{\beta \bar{f}})
$$

$$
+ \left(\frac{2\ell}{\sqrt{\mu}} + C\right)\left(\frac{\ell}{\sqrt{\mu}(q-1)C}\right)^{\frac{2-2q}{-2q+1}} W_1(\mathcal{L}(\mathbf{x}_k), \pi_{\beta \bar{f}})^{\frac{2-2q}{-2q+1}}
$$

Setting

$$
c_{18} = \|\nabla \bar{f}(0)\| + \ell\frac{2\left(\|\nabla \tilde{f}(0)\| + \sqrt{\|\nabla \tilde{f}(0)\|} + \sqrt{\mu \tilde{f}(0)}\right)}{\mu}
$$

$$
c_{19} = \left(\frac{2\ell}{\sqrt{\mu}} + \left(\mathbb{E}[\|\mathbf{x}_0\|^{2q}] + c_{22}\right)\frac{\ell^q 2^{q-1}}{(q-1)}\right)\left(\frac{\ell}{\sqrt{\mu}(q-1)\left(\mathbb{E}[\|\mathbf{x}_0\|^{2q}] + c_{22}\right)\frac{\ell^q 2^{q-1}}{(q-1)}}\right)^{\frac{2-2q}{-2q+1}}.
$$

and plugging this bound into (83) give the suboptimality bound from (82).

In particular, if $q = 4$, $\beta \geq 1$ and $W_1(\mathcal{L}(\mathbf{x}_k), \pi_{\beta \bar{f}}) \leq 1$ we get a bound of the form:

$$
\mathbb{E}[\bar{f}(\mathbf{x}_k)] \leq \min_{x \in \mathcal{K}} \bar{f}(x) + cW_1(\mathcal{L}(\mathbf{x}_k), \pi_{\beta \bar{f}})^{\frac{2}{3}} + \frac{n}{\beta}\left(\max\{\log \varsigma, 0\} + c_{12}\right)
$$

for some constant $c$ independent of $\beta$.

Indeed, $c_{22}$ decreases monotonically with respect to $\beta$, and thus so does $c_{19}$. So, assuming $\beta \geq 1$, we can take $c \geq c_{18} + c_{19}$ to be a fixed value independent of $\beta$.

Setting $\beta$ as in (79) gives

$$
\mathbb{E}[\bar{f}(\mathbf{x}_k)] \leq \min_{x \in \mathcal{K}} \bar{f}(x) + cW_1(\mathcal{L}(\mathbf{x}_k), \pi_{\beta \bar{f}})^{\frac{2}{3}} + \frac{\epsilon}{2}
$$

Then, arguing as in the proof of Proposition 21, for sufficiently large $\beta$ and $T \geq 4$, we have that

$$
cW_1(\mathcal{L}(\mathbf{x}_k), \pi_{\beta \bar{f}})^{\frac{2}{3}} \leq c\left(c_1 + c_2\sqrt{\varsigma} + \frac{c_3 + c_4\sqrt{\varsigma}}{(2a)^{1/2}}\right)^{2/3} T^{-1/3}\log T
$$

$$
\leq e^{\frac{2\beta \ell R^2}{3}} T^{-1/3}\log T
$$

Then setting, $\alpha = 2n\left(\max\{\log \varsigma, 0\} + c_{15}\right)\ell R^2$, $\beta$ from (79), and $T = e^{\gamma/\epsilon}$ gives

$$
cW_1(\mathcal{L}(\mathbf{x}_k), \pi_{\beta \bar{f}})^{\frac{2}{3}} \leq \exp\left(\frac{1}{\epsilon}\left(\frac{2\alpha}{3} - \frac{\gamma}{3}\right)\right)\frac{\gamma}{\epsilon}
$$

So, we seek a sufficient condition for

$$
\exp\left(\frac{1}{\epsilon}\left(\frac{2\alpha}{3} - \frac{\gamma}{3}\right)\right)\frac{\gamma}{\epsilon} \leq \frac{\epsilon}{2} \iff \exp\left(\frac{1}{\epsilon}\left(\frac{2\alpha}{3} - \frac{\gamma}{3}\right)\right)\frac{2\gamma}{\epsilon^2} \leq 1.
$$

Then, similar to the compact case, we have that when $\gamma > 2\alpha$, the left side is maximized over $(0, \infty)$ at $\epsilon = \frac{\gamma - 2\alpha}{6}$. Plugging in the maximizer gives the sufficient condition:

$$
\frac{72e^{-2}\gamma}{(\gamma - 2\alpha)^2} \leq 1
$$

This is satisfied in particular at
$$c_{20} = \gamma = 8\alpha + 72e^{-2}.$$

∎