# OpenReview forum: "Constrained Langevin Algorithms with L-mixing External Random Variables"
_NeurIPS.cc/2022/Conference — NeurIPS 2022 Accept_

### Official Review · Reviewer_RTV8 · 2022-06-17

**Rating:** 6
**Confidence:** 4
**Soundness:** 3 good
**Presentation:** 3 good
**Contribution:** 3 good

**Summary:**

This paper gives W_1 convergence guarantees for the constrained Langevin algorithm with stochastic gradients over a polyhedral constraint set. The assumption on the target function is that it is Lipschitz everywhere and strongly convex outside of a ball; moreover, the stochastic gradients are not i.i.d., but rather are obtained from an L-mixing process (which quantifies the amount of dependence in the noise). Similar problems have been studied before on more general constraint sets, but in the case of a polyhedral constraint set this paper obtains the sharpest bound thus far. The main challenge in this work is the mathematical difficulty associated with studying SDEs reflected at the boundary.

**Questions:**

None.

**Limitations:**

In the paper, quantitative dependence on problem parameters such as the dimension is not made explicit until the appendix. This is an issue because several parameters are quite large, e.g., there is an exponential dependence on the radius R of the ball without strong convexity, and an exponential dependence on the rank of the matrix specifying the polytope constraints. For the former, this is a standard feature of works which make the “strongly convex outside ball” assumption, but nevertheless such exponential dependence should be discussed clearly in the main text (especially since the abstract claims to handle non-convex functions, but realistically these bounds are quite poor for very non-convex functions). In particular, the contraction factor a is one of the most important parameters and an explicit bound on a should appear in the main text.

**Strengths And Weaknesses:**

This paper makes a good contribution to a challenging problem. Technically, the analysis is quite similar to that of the prior work [24], which dampens the novelty. However, I believe that this is outweighed by the virtue of analyzing the projected Langevin algorithm, for which quite little is known. I also found the writing to be clear.

Another notable weakness is the lack of theoretical comparisons with other approaches, such as proximal LMC [5]. I understand that the settings considered in prior works may differ from the one considered here, but nevertheless it would be helpful to include a discussion on whether techniques in those other works can be expected to apply to this setting, especially since the other approaches avoid the technicalities involved in the Skorokhod problem.

Specific Comments:
- Line 107, typo in the Lipschitz gradient assumption
- Line 691 in supplement, typo: or -> of

---

> ### Author Response · Authors · 2022-08-02
> **Response to Reviewer RTV8**
>
> Thanks for the valuable comments. Below, we will respond to the weaknesses, limitations you pointed out.
>
> (1) Response to Weaknesses:
>
> -  Though the analysis looks similar to the prior work [24], we have a more general assumption on the external random variables which are assumed to be the L-mixing processes. The class of L-mixing processes is time-correlated and covers many applications in system identification and time-series analysis. In [24], IID random variables are considered, which are a special case of L-mixing processes. Besides, in [24] the constraint set is closed and bounded and during the proof, the diameter of a ball covering the constraint set is utilized. Instead, our work studies the closed polyhedral constraint, which is not necessarily bounded. A tighter bound of the discretized process and continuous-time process is obtained through a constructive proof of an earlier result from [14].
>
> -  This is a good point, that it may be possible to avoid the technicalities of Skorokhod problems by changing the algorithm. We will append the following sentence to the conclusion:
>
>      ``Future work will examine whether the projection step, and thus Skorkhod problems, can be circumvented by utilizing different algorithms, such as those based on proximal LMC [5].''
>
>
> (3) Response to Limitations:
>
> We will put explicit bounds for the main constants, $c_1,\ldots,c_4$ in the main paper. As you mention, some constants will have a factor of the form $e^{\beta \ell R^2/8}$, which can be large, especially if $\beta$ or $R$ is large. This type of scaling is likely unavoidable, since the Langevin algorithm gives near-optimal solutions to a general class of non-convex optimization problems. So, NP-hardness would imply that the method would scale badly in some instances. In this case, the scaling shows up in the constant factors.

---

> > ### Comment · Reviewer_RTV8 · 2022-08-06
> > **Response**
> >
> > Thank you for the response. I stand by my original evaluation.

---

### Official Review · Reviewer_6kMS · 2022-07-09

**Rating:** 4
**Confidence:** 4
**Soundness:** 3 good
**Presentation:** 3 good
**Contribution:** 3 good

**Summary:**

This paper studies the problem of non-asymptotic convergence guarantees in 1-Wasserstein distance for projected Langevin Monte Carlo with (unbounded) polyhedral constraints, non-convex losses and L-mixing data. The convergence rate $O(T^{-1/2}\log T)$, is faster than the previous works on constrained Langevin algorithms in the literature.

**Questions:**

(1) On page 2 and page 3, I suggest the author(s) to define 1-Wasserstein distance and the polyhedral just in case some of the less theoretical readers are not familiar with these concepts.

(2)  The authors should discuss or make some comments about the assumption in Section 2.4. You assumed that $\bar{f}(x)$ is $\mu$-strongly convex outside a ball of radius $R>0$. This can be confusing to some readers who are not familiar with such assumptions because on the surface it gives the readers the impression that you are assuming almost strong-convexity whereas in the abstract you are claiming that you are doing non-convex learning. Please provide some references and comments on whether this is a common assumption, and whether it is equivalent for example to dissipativity condition in the literature.

(3) You mentioned that most of the novel work in the paper focuses on deriving Lemma 3 and you provided Section 3.3 for a proof overview for Lemma 3. But I think what would really be nice is to provide some high-level general discussions in plain English either before Section 3.3. or at the beginning of Section 3.3 about the novelty of your techniques to obtain Lemma 3 as well as the high-level descriptions of the steps needed to achieve Lemma 3.

(4) Journal names etc. should be capitalized in the reference. For example, the journal name in [4] should be capitalized. Also in [6] and [9], langevin should be Langevin.

**Limitations:**

Indeed, the author(s) included a section near the end of the paper discussing the limitations of their work.

**Strengths And Weaknesses:**

This is a solid theoretical paper and the analysis seems rigorous. This paper obtains the problem of non-asymptotic convergence guarantees in 1-Wasserstein distance for projected Langevin Monte Carlo with (unbounded) polyhedral constraints, non-convex losses and L-mixing data, with the convergence rate $O(T^{-1/2}\log T)$ that is faster than the previous works on constrained Langevin algorithms in the literature. My understanding is that the paper has made the following contributions (1) the convergence rate is faster than the previous works for constrained Langevin algorithms (2) the continuous-time convergence rate in 1-Wasserstein distance extends some existing literature to allow reflected boundary (3) novel discretization error bound where the constraint can be unbounded.

There are several weaknesses of the paper that makes me question the technical novelty and the significance of the contributions of the paper.

(1) Even though the results are new, similar analysis for non-convex learning for projected Langevin algorithms has already been appeared in [24] for example. The author(s) should do a better job at explaining what are the technical novelties and innovations that allow them to improve the convergence rate and allow the result to work on unbounded constraint set.

(2) In the model, there is a scaling parameter $\beta$, that is known as the inverse temperature. When you are doing sampling, you can simply let $\beta=1$. But when you need to use Langevin algorithms to solve a non-convex optimization problem, you need to have large $\beta$ in order for the Gibbs distribution to concentrate around the global minimizer of the target; see e.g. [30]. The authors at least should comment on the order of dependence of the constants on $\beta$ and whether it is possible to allow $\beta$ to be large to be used for non-convex optimization. Moreover, Theorem 1 is a result that can be used for sampling for non-log-concave target distribution, but can you use the ideas from [30] to obtain results for empirical risk minimization/population risk minimization? It is not clear to me because your result is in 1-Wasserstein and the constraint set is unbounded. In some sense, it is not clear to me your 1-Wasserstein result can lead to any non-convex optimization guarantees on an unbounded domain.

(3)  In Lemma 7, Lemma 8, the error bound has the term $e^{\eta\ell k}$, where $\ell$ is the parameter of the $\ell$-smoothness of the target, $\eta$ is the stepsize and $k$ is the number of iterates. The authors should comment on how tight/good these bounds are. I am a bit worried that these are too loose. For example, to make Langevin algorithm work, you need the continuous time dynamics to be close to the invariant distribution, also known as the Gibbs distribution, which requires $\eta k$ to be large. But if $\eta k$ is large, then the error bounds in Lemma 7 and Lemma 8 are exponentially bound. The discretization error bound in [30] does not have exponential dependence on $\eta k$ and it would certainly be great if the author(s) can improve upon this.

---

> ### Author Response · Authors · 2022-08-02
> **Response to Reviewer 6kMs, part I**
>
> Thanks for valuable comments. Below, we will respond to the weaknesses, limitations you pointed out and answer the questions you brought up.
>
> (1) Response to Weakness One:
>
> Though the analysis may look similar to that in [24], the problem we solve is specifically constrained in a polyhedral domain and with L-mixing random variables. Such a constraint set removes the boundedness assumptions, and the random variables under consideration are not IID. In [24], the constraint is convex compact, and the external random variables are IID. The author of [24] uses Lemma 2.2 of [34] (Tanaka, 1979) to bound the distance between the continuous-time process and the discretized process, which is rather loose. However, we are able to get a tight bound between the continuous-time process and discretized process with the constructive proof of the earlier result from [14] without boundedness assumption. The proof of Theorem 9 is one of the major novelties of this paper. The discussion of such a technical novelty appears in Section 4. Besides, instead of dealing with commonly considered IID random noise, we consider a class of L-mixing process as the external random variables, which is typical in system identification and time-series analysis. L-mixing random variables generalize the IID random variables and cover most of the stable Markov models. The consideration of such a kind of time-correlated random variables is another novelty of this paper. We present the technical novelties in the introduction part and also highlight them in the main paper.
>
> (2) Response to Weakness Two:
>
> We will add the sub-optimality proof of Gibbs algorithms in the revision. This will show the the dependency of parameter $\beta$ explicitly.
>
> (3) Response to Weakness Three:
>
> - It is true that Lemma 7 and Lemma 8 have undesired exponential error bounds shown as $e^{\eta \ell k}$. However, Lemma 7 and Lemma 8 are used to prove Lemma 3. And the proof of Lemma 3 shows how to completely eliminate the exponential dependency and get a time-independent bound of $O(\eta^{-1/2} (\log \eta^{-1})^{1/2})$.
>
> - The work [30] only considers the IID external random variables, while our work deals with time-correlated external random variables (L-mixing processes) and the exponential dependency appears in the process of averaging out the external random variables. Hence, the result of the discretization error is not comparable.
>
> (4) Response to Question One:
>
> We will add the definition of 1-Wasserstein distance in the revision. The definition of polyhedral set is defined in Section 4, which we believe is easier for the reader to keep track of the technical details due to the page limit.

---

> ### Author Response · Authors · 2022-08-02
> **Response to Reviewer 6kMS, part II**
>
> (5) Response to Question Two:
>
> Strong convexity outside a ball is an assumption to ensure non-explosiveness of the solutions to the Stochastic Differential Equations. In other words, strong convexity is one of the sufficient conditions for the stochastic stability. In literature [dalalyan2019user,durmus2019high,chatterji2018theory,baker2019control] on unconstrained Langevin algorithms, convergence analysis in Wasserstein distance is conducted under the assumption of strong convexity over the whole $\mathbb{R}^n$ domain. The paper [majka2020nonasymptotic] uses the term contractivity at infinity (which is the same as our assumption) to cover a wider class of SDEs such as equations with drifts given by double-well potentials to replace the global log-concavity assumption.
>
> The dissipativity condition in [chau2019stochastic] and [raginsky2017non] is not equivalent to the strong convexity outside a ball assumption. Instead, the dissipativity condition is weaker. Here we present some algebraic manipulation to show that the relationship between these two sufficient conditions for stochastic stability.
>
> Assume $\bar f(x)$ is $\mu$-strongly convex outside a ball of radius $R>0$, i.e. $(x_1 - x_2)^\top(\nabla \bar f(x_1) - \nabla \bar f(x_2)) \ge \mu \|x_1 - x_2\|^2$ for all $x_1, x_2 \in \mathcal{K}$ such that $\|x_1 - x_2\| \ge R$. Since $0 \in \mathcal{K}$, we can let $x_2 = 0$ in the assumption.
> Therefore, we have $x_1^\top(\nabla \bar f(x_1) - \nabla \bar f(0)) \ge \mu \|x_1 \|^2$ for all $x_1 >R$. This is to say $x_1^\top \nabla \bar f(x_1)\ge \mu \|x_1 \|^2 + x_1^\top \nabla \bar f(0) $ for all $\|x_1\| >R$. To ensure this inequality to hold, it suffices to have $x_1^\top \nabla \bar f(x_1)\ge \mu \|x_1 \|^2 + \|x_1\|\| \nabla \bar f(0)\| $ for all $\|x_1\| >R$ by C-S inequality. Note that $\|x_1\|\| \nabla \bar f(0)\| $ is non-negative. Compared with the disspativity condition in \cite{chau2019stochastic} and \cite{raginsky2017non} (There exists $a,b > 0 $ such that for all $x \in \mathbb{R}^n$, $x^\top  \bar f(x) \ge a \|x\|^2 - b $). We can see that strong convexity outside a ball implies the disspativity condition if it is also constrained outside a ball.
>
>
> Besides, the strong convexity outside a ball assumption can be enforced using weight decay regularization, which is common in machine learning problems. More details can be seen in the appendix of [raginsky2017non].
>
> Overall, strong convexity is a fairly standard assumption for the existing convergence analysis of Langevin algorithms in Wasserstein distance. We limit the strong convexity outside a ball so that within such a ball, the objective function can be non-convex. This is why we identify our work as non-convex learning.
>
> (6) Response to Question Three:
>
> We will add the following paragraph to the beginning of Section 3.3:
>
> This subsection describes the main ideas in the proof of Lemma 3. The results highlighted here, and proved in the appendix, cover the main novel aspects of the current work. The first novelty, captured in Lemmas 4 and 5, is a new way to bound stochastic gradient Langevin schemes with L-mixing data from a Langevin method with the data variables averaged out. The key idea is a method for examining a collection of partially averaged processes. The second novelty is a tight quantitative bound on the deviation of discretized Langevin algorithms from their continuous-time counterparts when constrained to a polyhedron. This result is based on a new quantitative bound on Skorokhod solutions over polyhedra.

---

### Official Review · Reviewer_xdj5 · 2022-07-27

**Rating:** 4
**Confidence:** 4
**Soundness:** 4 excellent
**Presentation:** 2 fair
**Contribution:** 2 fair

**Summary:**

Authors study the problem of constrained sampling guarantees for the projected Langevin algorithm in a non-asymptotic sense in Wasserstein 1 metric.

**Questions:**

Q1: What is the dimension dependence of this convergence guarantee?

Q2: What is the motivation for sampling from the Gibbs potential \bar{f}? Is this related to a Bayesian posterior sampling problem or any practical sampling problem? Is it only for obtaining non-convex optimization guarantees?

Q3: What is the benefit of using Eberle's result instead of a functional inequality based approach such as LSI + Holley-Stroock as in Vempala&Wibisono if the dimension dependence is already exponential?

**Limitations:**

See the section "Strengths and Weakness".

**Strengths And Weaknesses:**

- Strengths:
+The paper is well written and the proofs look correct as far as I can tell.
+ Improves the previously known rates under milder conditions.

- Weakness:
+ Major concern 1: Lack of motivation in the solved problem.
+ Major concern 2: it is not clear how the dimension and other problem parameters interact with the convergence rate. Since the authors are relying on Eberle's reflection coupling, the dimension dependency will be exponential, e.g. exp{LR^2}.
+ Insufficient recent sampling literature review: Unconstrained sampling with Langevin algorithm is a well studied problem, especially under quadratic growth conditions that the current paper is making. Authors cite a few papers from 2017 referring to them as recent, yet there has been a lot of progress since then.

- Minor concerns:
I listed a few typos that I noticed below.
+ l20 Deep Neural networks
+ l107 missing \nabla
+l133 the interpolation process is not properly defined, so is the Brownian motion (and its filtration).
+l85 before ever mentioning about f, we are introduced its gradient.

---

> ### Author Response · Authors · 2022-08-02
> **Response to Reviewer xdj5, part I**
>
> (1) Response to Major Concern 1:
>
> Here we summarize the contribution of this paper, which also implies the motivation of solving this specific problem.
>
> Our work derives a tight convergence rate of Langevin algorithms with L-mixing process under polyhedral constraints. Polyhedral constraint is a common constraint of optimization problems, and include boxes, orthants, and simplex constraints. Though polyhedral constraint as a special type of the convex constraint seems less general compared with the compact convex constraint in [lamperski2021projected] and the non-convex constraint in [sato2022convergence], we remove the boundedness assumption of the constraint set.
>
> Specifying L-mixing data variables in our analysis is an extension to the existing works which either have no external random variables or have IID external random variables. Though the work [chau2019stochastic] also considers the L-mixing random variables, but there is no constraint in that case. We want to emphasize that L-mixing random variables generalize the IID assumption in [raginsky2017non,lamperski2021projected], in particular it covers many stable Markov models. More specifically, all uniformly ergodic Markov chain can be proved to be L-mixing. Thus, the convergence rate of Langevin algorithms with L-mixing data variables we obtain gives theoretical guarantee to a broad set of real-world problems. Moreover, our result matches the result of the work [chau2019stochastic], which is the first work considering the L-mixing random variables but without constraints at all. Though with polyhedral constraints, our work matches the best known convergence rate in the unconstrained Langevin algorithms for non-convex problems [chau2019stochastic] and tighter than the work only considering IID random variables [lamperski2021projected].
>
> Overall, our work fills in the gap of the theoretical analysis of constrained Langevin algorithms with dependent data streams and provides convergence guarantees of Langevin algorithms with arbitrary polyhedral constraints and L-mixing data variables.
>
> (2) Response to Major Concern 2:
>
> You are correct that we have this exponential term $e^{\beta \ell R^2/8}$, but it is not dependent on the dimension. Instead, such an exponential bound depends on the parameter $\beta$ and $R$. We will clarify how the constants depends on $\beta$ and scale with $beta$. Besides, we will make a more explicit expression for the main constants $c_1, \ldots, c_4$ in the revision such that the dimension and parameter dependencies will be shown clearly.
>
> (3) Response to "Insufficient sampling literature":
>
> Based on this suggestion, we will update the literature review in the revision. Below is a review summary including the recent unconstrained sampling results.
>
> More recent relevant work are given in the list below:
> - Difan Zou, Pan Xu, and Quanquan Gu. Faster convergence of stochastic gradient langevin
>  dynamics for non-log-concave sampling. In Uncertainty in Artificial Intelligence, pages 1152–1162. PMLR, 2021.
> - Ruilin Li, Hongyuan Zha, and Molei Tao. Sqrt (d) dimension dependence of langevin monte
>   carlo. arXiv preprint arXiv:2109.03839, 2021.
> - Murat A Erdogdu, Rasa Hosseinzadeh, and Shunshi Zhang. Convergence of langevin monte
>   carlo in chi-squared and rényi divergence. In International Conference on Artificial Intelligence
>   and Statistics, pages 8151–8175. PMLR, 2022.
> - Krishna Balasubramanian, Sinho Chewi, Murat A Erdogdu, Adil Salim, and Shunshi Zhang.
>   Towards a theory of non-log-concave sampling: first-order stationarity guarantees for langevin
>   monte carlo. In Conference on Learning Theory, pages 2896–2923. PMLR, 2022.
> - Dao Nguyen, Xin Dang, and Yixin Chen. Unadjusted langevin algorithm for non-convex weakly
>   smooth potentials. arXiv preprint arXiv:2101.06369, 2021.
> - Joseph Lehec. The langevin monte carlo algorithm in the non-smooth log-concave case. arXiv
>   preprint arXiv:2101.10695, 2021.
> - Sinho Chewi, Murat A Erdogdu, Mufan Bill Li, Ruoqi Shen, and Matthew Zhang. Analysis of
>   langevin monte carlo from poincar\’e to log-sobolev. arXiv preprint arXiv:2112.12662, 2021.
>
> We will have more discussions on these works in the revision.
>
> (4) Response to Q1:
>
> The dimensional dependency is $\mathcal{O}(n)$ based on the list of constants in the appendix. The detail of getting such a dimensional dependency is shown below:
>
> In Theorem 1, $c_1, c_2, c_3, c_4$ show up. Using the constant list, we can see $c_2$ and $c_4$ has no dimensional dependency, $c_1$ has $\mathcal{O}(\sqrt{n})$) dependency and $c_3$ has $\mathcal{O}(\sqrt{n}) + \mathcal{O}(n)$ dependency. So overall, the dimensional dependency is $\mathcal{O}(n)$.

---

> ### Author Response · Authors · 2022-08-02
> **Response to Reviewer xdj5, part II**
>
> (5) Response to Q2:
>
> - Sampling from Gibbs distribution, i.e. Langevin algorithms is used for high-dimensional and large-scale sampling applications. Gibbs distribution is an invariant distribution of the Langevin dynamics. And Langevin dynamics can converge to arbitrary probability distributions, so we can sample from a large amount of distribution via Langevin algorithms.
>
> - It is related to Bayesian posterior sampling. Langevin algorithms can be used to sample from the posterior in Baysian settings. When setting $\beta = 1$, Langevin algorithms can be used for posterior sampling, which is the basic idea on Bayesian learning. More details about Langevin algorithms for Bayesian learning is introduced in [welling2011bayesian].
>
> - It is not only for obtaining non-convex optimization guarantees. Instead, Theorem 1 indicates the distance between a target distribution and the distribution of the samples from the algorithms. Some extra work will be added to the revision which shows the near-optimality of the algorithms iterates.
>
> (6) Response to Q3:
>
> LSI method in Vempala and Wibisono utilizes the KL divergence, which is infinite when the initial condition is deterministic. Therefore, methods using LSIs typically require the initial density to be supported everywhere. However, Wasserstein distance is well defined with deterministic initial condition. So we choose to use Eberle's result which uses Wasserstein distance. The detailed reasoning is below:
>
> Let $q(x)$ be the stationary distribution, and  $q(x) =  \frac{1}{\sqrt{2 \pi}} e^{-x^2/2}$, which is a standard Gaussian distribution.
>
> Let the initial distribution $p(x)$ be Gaussian with mean 0 and variance $\sigma^2$. So, $p(x) =  \frac{1}{\sqrt{2 \pi}\sigma} e^{-x^2/(2\sigma^2)}$. As $\sigma \rightarrow 0$, $p(x)$ approaches to a deterministic distribution which concentrates at 0 with probability 1.
>
> The KL divergence can be computed as $\int p(x) \log \frac{p(x)}{q(x)} dx = \frac{\sigma^2}{2}- \frac{1}{2} - \log \sigma$ which goes to $\infty$ as $\sigma \rightarrow 0$.
>
> In contrast, $W_1$ remains bounded since both $p(x)$ and $q(x)$ have bounded variance.
> In particular, if $P$ and $Q$ are the respective measures, then
> \begin{align*}
> W_1(P,Q) \le \int \|x\| p(x) dx + \int \|x\| q(x) dx \le \sigma +1.
> \end{align*}
>
> In particular, when $P$ corresponds to the Dirac delta centered at 0 (i.e. we have $x=1$ with probability 1), we have $W_1(P,Q) = \int \|x\| q(x) dx \le 1$.
>
> So we can see that $W_1$ is more flexible which allows the deterministic initialization. This is why we use Eberle's result.
>
> To clarify this point, we will make some comments on work using LSIs in the revision.

---

### Official Review · Reviewer_JNhe · 2022-07-27

**Rating:** 5
**Confidence:** 3
**Soundness:** 3 good
**Presentation:** 3 good
**Contribution:** 3 good

**Summary:**

The paper aims at deriving new bounds for langevin algorithms in some specific cases, showing that the bound of convergence it attains is better than previously obtained in the literature.
It focuses on obtaining new 1-Wasserstein distance for non-convex losses with L-mixing data variables and polyhedral constraints, showing that the rate of convergence is faster.

**Questions:**

My main question to the authors is can you find a practical case where the bound they obtain begins to be optimal, especially with respect to the rest of the literature ? Which would validate and motivate to my mind the impressive theoretical work you show.

**Limitations:**

The limitations are we’ll described by the authors  and are mainly the specificity of the results.

**Strengths And Weaknesses:**

The paper is technically well  written and presents a novel bound for Langevin algorithms. The math is advanced and the proof technically difficult.
Furthermore, the paper is well organized and the derivation is clear.
I checked some of the proofs which were very involved and the resulting paper is impressive theoretically.

The main weaknesses of the paper are to my mind thelimitation to the specific case with L-mixing data variables and polyhedral constraints, which is  Unfortunately not supported by a practical example to show the relevance of the approach proposed by the authors.

---

> ### Author Response · Authors · 2022-08-02
> **Response to Reviewer JNhe**
>
> Thanks for the valuable comments. Below, we will respond to the weaknesses, limitations you pointed out and answer the questions you brought up.
>
> (1) Response to Weaknesses and Limitations:
>
> We want to clarify that specifying L-mixing data variables is not a limitation of our work, but an extension to the existing works. Most of the existing literature do not consider the external random variables at all [dalalyan2012sparse,dalalyan2017theoretical,durmus2017nonasymptotic,ma2019sampling,majka2018non, wang2020fast]. There are some works considering the external randomness, but only restricted to IID random variables [raginsky2017non,lamperski2021projected]. Though the work [chau2019stochastic] also considers the L-mixing random variables, there is no constraint in that case. We want to emphasize that L-mixing random variables generalize the IID assumption, in particular it covers most of the stable Markov model. More specifically, all uniformly ergodic Markov chain can be proved to be L-mixing.  Thus, the convergence rate of Langevin algorithms with L-mixing data variables we obtain gives theoretical guarantees to a broad set of real-world sampling and optimization problems.
>
> We are clear about the limitation of the constraint assumption. Though polyhedral constraint as a special type of convex constraint seems less general compared with the compact convex constraint in [lamperski2021projected] and the non-convex constraint in [sato2022convergence], we remove the boundedness assumption of the constraint set. In that sense, our work is dealing with a weaker assumption in terms of boundedness. Not to mention that a lot of works do not consider constraints at all [raginsky2017non,xu2018global,erdogdu2018global,cheng2018sharp,chen2020stationary]. Therefore, though the constraint is specified as a polyhedron in our case, we still make progress compared with the existing work. Besides, the polyhedral constraint is very common in applications with box and simplex constraints, so the convergence analysis of Langevin algorithms with polyhedral constraints is of practical value.
>
> A specific example matches our assumption is the system identification of the autoregressive model. We can estimate the output via a neural network and define the loss function as the mean square error, which is stated in [chau2019stochastic]. This is a typical non-convex learning problem and if we clip the parameters of the neural network, the constraint is polyhedral. Besides, the regularization term in the loss function makes the strong convexity outside a ball assumption holds.
>
> (2) Response to Questions:
>
> We present a practical case above. To the best of our knowledge, our convergence rate matches the best known rate of the unconstrained Langevin algorithms for non-convex problems [chau2019stochastic], but arbitrary polyhedral constraints are enforced in our case. In other words, our work provides theoretical guarantees of the convergence of the constrained Langevin algorithms with L-mixing random variables, which covers the case of IID random variables and thus is more general.

---

### Meta-Review · Area_Chair_KUwv · 2022-08-26

**Recommendation:** Accept
**Confidence:** Certain

**Metareview:**

After going through all the reviews, rebuttals, and discussions in detail I am recommending a borderline acceptance for the paper. More precisely, the technical contribution of the paper is significant, even though there have been some concerns raised regarding the motivation/applicability of the setup. However, I do believe that the merits of the paper outweigh its limitations. I recommend the authors to implement all the suggested changes.

**Award:**

No

---

### Decision · Program_Chairs · 2022-09-14

Accept